# Large Eddy Simulation of the IEA 15-MW Wind Turbine Using a Two-Way Coupled Fluid-Structure Interaction Model

Claudio Bernardi<sup>1</sup>, Stefania Cherubini<sup>1</sup>, Felice Manganelli<sup>1</sup>, Giacomo Della Posta<sup>2</sup>, Stefano
 Leonardi<sup>3</sup>, and Pietro De Palma<sup>1</sup>

<sup>1</sup>Department of Mechanics, Mathematics and Management, Polytechnic University of Bari,
 70126, Bari, Italy (claudio.bernardi@poliba.it, stefania.cherubini@poliba.it,
 f.manganelli@phd.poliba.it, pietro.depalma@poliba.it)

 <sup>2</sup>Department of Mechanical and Aerospace Engineering, Sapienza University of Rome, Rome, RM, 00184, Italy (giacomo.dellaposta@uniroma1.it)

<sup>3</sup>Department of Mechanical Engineering, University of Texas at Dallas, Richardson, TX,
 75080, USA (stefano.leonardi@utdallas.edu)

12 **Correspondence:** Claudio Bernardi (claudio.bernardi@poliba.it)

## 13 Abstract

The aim of the work is studying the aeroelastic response of the 15 MW NREL-IEA large-14 scale wind turbine using a high-fidelity fluid-structure interaction solver that combines 15 large-eddy simulation with a modal computational structural dynamics solver through 16 a two-way coupling. The fluid solver employs the actuator line model to simulate the 17 interaction between the turbine blades and the fluid and the immersed boundary method 18 to model the presence of the tower and nacelle. The results are compared with those ob-19 tained by the OpenFAST software, which is a well-known numerical tool for engineering 20 predictions. A series of simulations have been performed with and without the pres-21 ence of the tower and nacelle to better understand the effects of these components on 22 flow structures and structural deformations. The largest discrepancies among the solvers 23 24 have been observed in correspondence with the blade passage in front of the tower, which induces an abrupt alteration in the local incidence angle of the flow. Moreover, by com-25 paring the outcomes of different structural approximations, it has been established that 26 taking into account the torsional degree of freedom considerably affects the deforma-27 tions, aerodynamic loads and power coefficient. Whereas, the nonlinearity of the solver 28 29 appears to have a weak effect on the same quantities.

## 30 Keywords

<sup>31</sup> Aeroelasticity, Large Eddy Simulation, Actuator Line Model, Fluid-Structure Interaction, Computa-

<sup>32</sup> tional Fluid Dynamics, Computational Structural Dynamics, Blade Element Momentum, IEA-15MW

33 Wind Turbine.

## 34 1 Introduction

Wind energy has become a crucial component of the global transition toward renewable energy sources. 35 The increasing demand for clean energy has led to the development of large-scale wind turbines, such 36 as the IEA 15-MW offshore wind turbine developed within IEA Wind Task 37 (Gaertner et al., 2020). 37 This turbine, with a rotor diameter of 240 meters and blades measuring 117 meters in length, rep-38 resents a new frontier in wind energy technology (Gaertner et al., 2020), and research is currently 39 pointing towards even larger rotors, reaching 22-MW of power production (Zahle et al., 2024). The 40 increasing scale and flexibility of such newly designed turbines present significant engineering chal-41 lenges, particularly in predicting their aeroelastic response (Burton et al., 2011; Zheng et al., 2023). 42 43 As turbines grow in size, their structural components, especially the blades, are subject to complex aerodynamic forces that cause deformations, which in turn affect the aerodynamic loads. Understand-44 ing these interactions is essential to improve the performance, reliability, and longevity of large-scale 45 wind turbines (Manwell et al., 2010). In the worst cases, aeroelastic instabilities such as edgewise 46 instability and flutter might even lead to blade damage, as reported for the Lunderskov Mobelfabrik 47 19 m wind turbine blades (Moeller, 1997), with devastating effects on the turbine performance. 48 Aeroelasticity, the study of the interaction between aerodynamic, inertial and elastic forces due to the 49 deformation of the structure, is critical in the design and analysis of modern wind turbines. Aeroe-50 lastic phenomena such as dynamic stall, flutter, and fatigue can have significant effects on turbine 51 performance, particularly as the blade length increases (Hansen, 2007). These blades experience vary-52 ing aerodynamic forces along their span, which can lead to substantial deformations. When blades 53

<sup>54</sup> deform, they alter the local flow field, which in turn modifies the aerodynamic loads acting on them. <sup>55</sup> This feedback loop between aerodynamic forces and structural deformation makes it very difficult

to predict modern large-scale turbine performance under real-world operating conditions (Vermeer

<sup>57</sup> et al., 2003; Wang et al., 2016). Accurate evaluation of these interactions is key for ensuring turbine

<sup>58</sup> efficiency and structural integrity, especially in offshore environments where wind conditions are more <sup>59</sup> severe (Bayati et al., 2017).

The numerical modeling of the blades in most of the numerical aeroelastic codes used nowadays (Schep-60 ers et al., 2021) is accomplished by the blade element momentum (BEM) model, due to its robustness 61 and low computational cost. However, BEM has several limitations, due to the strong assumptions 62 made on the impinging flow, requiring models of dynamic stall, dynamic inflow, yaw and tilt flows, 63 and corrections of the aerofoil data for taking into account three-dimensional effects and tip losses. 64 Unfortunately, more computationally expensive models, such as the free-wake panel and the actuator 65 disc methods, are not able to predict the dynamic loading much more accurately. Therefore, the 66 application of computational fluid dynamics (CFD) to full-scale turbines is the most promising way 67 to drop those assumptions and describe the complex aerodynamics of the flow field more accurately 68

(Sørensen, 2011).

69 However, coupling three-dimensional CFD simulations with computational structural dynamics (CSD) 70 solvers taking into account the deformation of the blade is not trivial. Three-dimensional structural 71 finite-element models are in fact able to fully describe the complex shape of a wind turbine blade but, 72 although accurate, these models are computationally expensive and hard to implement, leading to 73 only a few examples of coupling with CFD codes (Bazilevs et al., 2011; Yu and Kwon, 2014). Since 74 wind turbine blades are slender structures, their structural modeling can be more easily achieved using 75 beam models, where the blade is approximated as a series of one-dimensional beam elements, each 76 characterised by a given cross-sectional stiffness and mass per unit length. One-dimensional beam 77 models can be either modal, since natural frequencies and mode shapes of a turbine are directly re-78 lated to the natural frequencies of its blades, or they can rely on the geometrically exact beam theory 79 including non-linear effects (Sabale and Gopal, 2019). 80 Due to their ability to provide a rapid evaluation of the turbine performance, numerical tools based on 81 the lifting-line approach equipped with aeroelastic modules based on one-dimensional beam models, 82 are currently widespread (Schepers et al., 2021). A notable example is OpenFAST, a numerical code 83 developed at NREL (Jonkman, Jonkman) and widely used for aeroelastic simulations, which employs 84 BEM theory for aerodynamic modeling and various structural solvers, such as ElastoDyn (Damiani 85 et al., 2015) and BeamDyn (Wang et al., 2016), for structural deformation analysis. However, it is 86 still not clear whether the predictions of such lifting-line aeroelastic codes are sufficiently accurate for 87 large-scale turbines, in which the effect of shear and inflow turbulence can lead to complex inflows and 88 turbine aerodynamic responses. Comparing the predictions of OpenFAST with those of a Large-Eddy 89 Simulation (LES) equipped with a structural one-dimensional beam model has shown that, for an 90 NREL 5MW wind turbine, the passage in front of the tower leads to large deformations which are 91 largely underestimated by OpenFAST (Bernardi et al., 2023). 92 93 Concerning rotors of even larger size, such as the IEA 15-MW reference turbine, it is not yet known whether these discrepancies in the predictions of lifting-line codes with respect to CFD are even more 94 consistent. Using the unsteady Reynolds-Averaged Navier-Stokes (URANS) equations coupled with 95 an aeroelastic module, as reported by Pagamonci et al. (2023), has shown that neglecting the flexibility 96 of the blades in numerical simulations leads to an underestimation of the rotor thrust of approximately 97 2.5% for the IEA 15-MW turbine, which is not observed for the smaller NREL 5MW rotor. More-98 over, this work also concluded that the deformation of long, slender blades may act as a filter for the 99 high-frequency fluctuations arising from the flow field, proving that taking into account the blades' 100 aeroelasticity in the design process of these machines is key for the future upscaling of turbine rotors. 101 Furthermore, Trigaux et al. (2024) observed how the use of high-fidelity aerodynamic models is crucial 102 to predict the aeroelastic effects of large rotors. These results suggest the need to investigate this issue 103 resorting to LES, which is capable of describing the dynamics of the flow more accurately. 104 In this context, the present work aims at studying the aeroelastic response of a large-scale 15-MW 105 wind turbine by means of LES, assessing the effect of the flexibility of the blades on the wake dynam-106 ics. The results are compared with those obtained by more simple and less computationally expensive 107

models, such as the OpenFAST code. Computations are performed by an in-house LES code using 108

the immersed boundary method to model the tower and nacelle and the Actuator Line Model (ALM) 109

for blade modeling, coupled with a structural modal solver, originally developed by Della Posta et al. 110

#### 111 (2022).

- <sup>112</sup> The discussion of the results highlights the role of the tower and nacelle in the dynamics of the aerody-
- namical forces, thrust and power coefficients, as well as in the distribution of turbulent kinetic energy
- <sup>114</sup> within the wake, which could have an impact on the aerodynamic loads of downstream turbines in <sup>115</sup> wind farms. Moreover, the effect of the torsional degree of freedom has been investigated by comparing
- the outcomes of different structural approximations.
- <sup>117</sup> The work is structured as follows. In section 2, the aerodynamic and structural solvers of both CFD-
- <sup>118</sup> CSD and OpenFAST codes are described in detail. In section 3, the numerical setup is presented. In
- <sup>119</sup> section 4, relevant results are discussed, and conclusions are drawn in section 5.

### 120 2 Methodologies

#### 121 2.1 CFD-CSD solver

#### 122 2.1.1 Flow solver

The simulations of the flow around the wind turbine are carried out through Large-Eddy Simulations (LESs) of the incompressible, filtered, 3D Navier-Stokes equations, employing our in-house UTD-WF solver (Santoni et al., 2020). The code implements a second-order accurate centered finite difference scheme for the spatial discretization on a staggered Cartesian grid. A hybrid low-storage third-orderaccurate Runge-Kutta (RK) scheme is used for time integration of the non-linear terms (Orlandi, 2012), while the linear terms are treated implicitly using a Crank-Nicolson scheme. The filtered governing equations are:

$$\frac{\partial u_i}{\partial t} + \frac{\partial u_i u_j}{\partial r_i} = -\frac{\partial p}{\partial r_i} + \frac{1}{Re} \frac{\partial^2 u_i}{\partial r_i \partial r_i} - \frac{\partial \tau_{ij}}{\partial r_i} + \tilde{f}_i,\tag{1}$$

$$\frac{\partial u_i}{\partial x_i} = 0, \tag{2}$$

where  $i, j \in \{1, 2, 3\}$  represent, in a Cartesian reference frame, the components along the stream-130 wise, wall-normal, and spanwise directions, respectively. The Reynolds number  $Re = U_{\infty}D/\nu$  is 131 defined by the undisturbed inlet velocity  $U_{\infty}$ , the turbine diameter D, and the kinematic viscosity of 132 the fluid  $\nu$ . These quantities are used as reference values to make the equations non-dimensional. To 133 134 solve the filtered equations, a Subgrid-Scale (SGS) stress model is needed. The latter describes the interaction between the large resolved and the sub-grid unresolved scales, as described by Pino Martín 135 et al. (2000) and Santoni et al. (2017). Here, we employ the Smagorinsky model with constant 136  $C_s = 0.09$  as discussed by Martinez-Tossas et al. (2018). 137

The effect of the blades on the flow is modeled by the Actuator Line Model (ALM) (Troldborg, 2009), by adding a forcing term to the Navier-Stokes equations, representing the force per unit volume exerted by the rotor on the fluid. By approximating the rotor blades as rigid straight lines discretized into segments, it is possible to estimate the lift and drag forces per unit length on a 2D plane as follows:

$$F_l = \frac{1}{2}\rho u_{rel}^2 C_l(\alpha) cF, \qquad F_d = \frac{1}{2}\rho u_{rel}^2 C_d(\alpha) cF, \tag{3}$$

where  $\rho$  is the air density, c is the local chord,  $u_{rel}$  is the relative incoming velocity,  $\alpha$  is the angle of attack, and F represents the Prandtl tip loss correction factor (Shen et al., 2005). The forces are then projected on the flow employing a 2D Gaussian kernel, which spreads the lift and drag force vector,  $f^{aero}$ , in cylinders surrounding the actuator line,

$$\tilde{\boldsymbol{f}} = -\boldsymbol{f}^{aero} \frac{1}{\epsilon^2 \pi} exp \left[ -\left(\frac{r_\eta}{\epsilon}\right)^2 \right],\tag{4}$$

where  $r_{\eta}$  is the radial distance of a generic point of the cylinder from the actuator line and  $\epsilon$  is the spreading parameter, where  $\epsilon/\Delta \ge 2$ , with  $\Delta = \sqrt{\Delta x^2 + \Delta y^2 + \Delta z^2}$ , following Troldborg (2009). The tower and nacelle are modeled using the Immersed Boundary Method (IBM) following the approach described by Orlandi and Leonardi (2006).

### 151 2.1.2 Structural solver

From an aerodynamic standpoint, the rotor blades represent the most flexible components within a wind turbine. Several studies demonstrated that their modal properties have a significant impact on the dynamics of the entire structure (Damgaard et al., 2013; Dong et al., 2018). Moreover, an analysis of the isolated blades is also sufficient to accurately estimate the aeroelastic properties of the entire structure, including the flutter speed (Abdel Hafeez and El-Badawy, 2018). Additionally, the tower and shaft exhibit minimal deflection due to their stiffness. In light of the above considerations, the aeroelastic model is constructed to encompass solely the structure of the blades.

The structural model used in the present study was previously presented by Della Posta et al. (2022, 159 2023). In order to model the working conditions, the blades are assumed to be rotating beams rigidly 160 clamped at the hub (cantilever beams), under the assumption of small deformations with respect 161 to a relative frame of reference (FOR). The direction of the pitching axis is denoted by  $X_1$ . This 162 coincides with the neutral axis of the blade, defined as passing through the quarter of the chord. 163 The direction of the out-of-plane flapwise motion is indicated by  $X_2$  and is oriented in the positive 164 streamwise direction. The in-plane edgewise direction of  $X_3$  is defined such that the FOR is oriented 165 as a right-handed coordinate system (Figure 1). 166

<sup>167</sup> Under the assumption of linearity, the elastic generalised displacement d, which includes translational <sup>168</sup>  $d_i$  and rotational  $\theta_i$  degrees of freedom (DoFs), is decomposed along the coordinate  $X_1$  on the neutral <sup>169</sup> axis as:

$$\boldsymbol{d}(X_{1},t) = \sum_{m=1}^{M} q_{m}(t) \boldsymbol{\psi}^{m}(X_{1}), \qquad (5)$$

where  $\psi^m(X_1)$  is the m-th elastic mode shape from the modal analysis of the structure,  $q_m$  is the corresponding modal coordinate, and M is the number of modes used. The general inertial coupling is included in a modal basis by means of the methodology introduced by Reschke (2005) and further developed for the case of wind energy by Della Posta et al. (2022). In particular, the two-way coupling

Figure 1: Sketch of the frames of reference used for the CFD and for the CSD simulations.

<sup>174</sup> algorithm between rigid-body and structural dynamics does not take into account a modification of <sup>175</sup> the rotor inertia caused by the deformation of the blades. Hence, the structural dynamics of the <sup>176</sup> structure can be described by the following equation:

$$\boldsymbol{M}\ddot{\boldsymbol{q}} + [\boldsymbol{D} + \boldsymbol{D}^{Co}(\boldsymbol{\Omega})]\dot{\boldsymbol{q}} + [\boldsymbol{K} + \boldsymbol{K}^{c}(\boldsymbol{\Omega}) + \boldsymbol{K}^{Eu}(\dot{\boldsymbol{\Omega}})]\boldsymbol{q} = \boldsymbol{e} + \boldsymbol{e}^{c}(\boldsymbol{\Omega}) + \boldsymbol{e}^{Eu}(\dot{\boldsymbol{\Omega}}), \tag{6}$$

where M, D and K denote the modal structural mass, damping, and stiffness matrices, respec-177 178 tively, and e are the external loads expressed in modal basis. The remaining terms are inherently related to the various contributions to the acceleration in a moving FOR. Terms with the superscript 179 Co, c and Eu are related to the Coriolis, centrifugal, and Euler accelerations, respectively. The dis-180 crete evaluation of the additional inertial terms in Equation (6) is expressed as a function only of the 181 information known from the structural finite-element method (FEM) model and from the correspond-182 ing mode shapes, according to Saltari et al. (2017). For the modal analysis, we use a finite element 183 model of the blade based on complete beam elements with 6 DoFs, with Euler-Bernoulli behavior 184 for bending in directions  $X_2$  and  $X_3$ , and linear shape functions for axial and torsional deformations. 185 The generalized- $\alpha$  method (Chung and Hulbert, 1993) is employed to advance the structural dynamic 186 equation in time, which is unconditionally stable for linear problems, and second-order accurate. We 187 assume a lumped-mass representation, and we take into account the local offset of the centers of mass 188 with respect to  $X_1$ . Finally, the structural matrices are assembled considering the local twist. Details 189 about the modal analysis are provided in Appendix A. 190