# Peer review of "Large Eddy Simulation of the IEA 15-MW Wind Turbine Using a Two-Way Coupled Fluid-Structure Interaction Model"

_Wind Energy Science, 2025_

## Referee Comment (RC1)

—- Main Positive Feedback —-

- The manuscript demonstrates strong scientific motivation and engages with a highly complex topic in a structured way. The authors succeed in breaking down the dynamic interplay of multiple driving factors into separate aspects and analyzing them in considerable detail.
- The separation into CFD-CSD/OV and CFD-CSD/TN configurations is particularly valuable, as it provides insights into the impact of the torsional degree of freedom that would otherwise remain hidden in real-world measurements.
- The figures and plots are generally clear, well-designed, and effective in illustrating the results. The effort to validate the structural model against five alternative models is commendable and shows attention to methodological robustness.
- The simulations also provide several interesting findings:
  - The 15 MW turbine exhibits slower wake recovery due to tower effects, in contrast to the behavior of the 5 MW turbine
  - The results highlight the major influence of the fidelity of both aerodynamic and aeroelastic solvers on the predictions. In particular, the study suggests that CFD-based approaches may be crucial for accurately capturing tower effects, blade dynamic response, and wake recovery.
  - The torsional degree of freedom has a clear impact on aerodynamic quantities, for example leading to a decrease in the power coefficient and affecting blade deformations.
    The presence of the tower produces peaks in $C_p$ and $C_t$ that could have important implications for fatigue loading.

—- Detailed Positive Feedback —-
- L. 530–532: *"The isolated low-frequency peaks found in BeamDyn and ElastoDyn suggest that these solvers tend to over-simplify the aerodynamic fluctuations associated with phenomena such as wind shear and tower shadowing."* – This is a valuable observation and aligns well with what one would theoretically expect.
- L. 544–548: *"In agreement with previous studies, the results thus suggest that including the torsional degree of freedom in the structural solver is crucial for accurately describing the amplitude and dynamical behaviour of the aerodynamic quantities. Moreover, it is observed that duly taking into account the torsional degree of freedom reduces the value of Cp."* – It is very good to see your work confirming both previous studies and qualitative trends that are expected on physical grounds.
- Figure 7 (page 13): The direct comparison between CFD-CSD/OV and CFD-CSD/T is highly informative in illustrating the impact of torsion on pitching moment and aerodynamic forces. Furthermore, the alignment between the corresponding CFD-CSD and OpenFAST variations (L. 540–542: *"Concerning the forces on the blade and the incidence angle, one can observe a rather good match between the CFD-CSD/OV solver and ElastoDyn, as well as between the CFD-CSD/T model and the BeamDyn solver"*) represents a strong qualitative finding.
- Figure 8 (page 14): This figure clearly highlights the oscillation at the tower position and is very effective in conveying the underlying physics.
- L. 568–570: *"Future work will explore the effect of turbulent fluctuations at the inlet to better investigate the impact of the atmospheric boundary layer on the aerodynamic*

*forces, loads and deformations of the present turbine."* – This is a logical and well-justified next step following the present analysis with laminar shear inflow.

—- Main Critique Points —-

Documenting the software and its validation status.

- As a reader, I found it a bit hard to trace how the different pieces of your in-house UTD-WF framework were developed across prior work, and what has been validated under which conditions. This is important to know that the in-house code you use - i.e., the foundation for all your work and insights - is credible and correct in the first place. Adding a short provenance/validation paragraph would strengthen the manuscript. For the protocol, this is what I understand from the literature:
    - Santoni et al. (2015) first introduced UTD-WF (they didn't call it "UTD-WF" yet, but rather incompressible LES + ALM, with IBM for tower/nacelle) and reported limited wind-tunnel validation at NTNU for a single model turbine (mean velocity & TKE), establishing basic fidelity of the ALM+IBM setup.
    - Santoni et al. (2017) then reproduced the NTNU "Blind Test" and compared simulations to Krogstad et al.'s measurements, quantifying the impact of tower and nacelle—further supporting the IBM+ALM approach.
    - At wind-farm scale, Santoni et al. (2020) used an actuator-disk representation within UTD-WF and a mesoscale–microscale coupling, considering both momentum-only and momentum+TKE variants.
    - Della Posta et al. (2022) introduced the two-way FSI coupling (ALM in UTD-WF + a modal structural model; ALM/IV/IVT options). While that paper mainly focused on methodology and inter-model comparisons, it did not include a dedicated experimental validation.
    - Finally, Della Posta et al. (2023) integrated a Beddoes–Leishman unsteady-aerodynamics model in the LES-FSI framework and examined uniform, laminar-shear, and turbulent ABL inflows; comparisons are discussed against reference datasets (including HAWC2-based results reported by Heinz, 2013).
- Clarifying which subsets constitute formal validation would help readers map prior evidence to your present setup. In that light, two clarifications would make your contribution easier to interpret:
    1. Scope of prior validation vs. present cases: Prior validations include uniform and ABL inflows for a 5 MW turbine; your study addresses the IEA-15 MW case for a sheared laminar inflow configuration. Since larger rotors can amplify aeroelastic effects, it would be helpful to note explicitly that the present application extends the validated setting and to discuss any implications or arising uncertainties.
    2. Citations for solver provenance: Where you cite the solver's origin, consider also referencing Santoni et al. (2015) as the first UTD-WF publication, prior to Santoni et al. (2020). Further mentioning the following papers and validations that were added to Santoni et al. (2015) would help readers that are unfamiliar with the code's history. Just so they know that the inhouse code you use is validated within a certain range.

Comparing BEM with CFD for aeroelastic simulations

- The conclusions drawn about the relative suitability of CFD-based versus BEM-based aeroelastic simulations are not entirely straightforward, because the setups being compared differ in more than one respect. In particular:
    - Case A (CFD-CSD) couples a finite-volume LES solver with ALM to a *linear* structural model (A.1: OV; A.2: TN).
    - Case B (OpenFAST with BeamDyn) uses BEM aerodynamics with a *nonlinear* structural solver.
    - Case C (OpenFAST with AeroDyn/ElastoDyn) combines BEM aerodynamics with a *linear* structural solver.
- Because both the aerodynamic *and* structural solvers vary simultaneously, it is difficult to isolate whether observed differences stem from the flow solver or from the aeroelastic solver (or from their specific implementations). Any causal conclusions should therefore be presented with appropriate caution.
- A related point concerns the comparison with Bernardi et al. (2023). That study focused on a different turbine (NREL 5 MW), under uniform inflow, and reports blade load distributions that do not match those of the present paper. Drawing direct inferences from those results to the IEA 15 MW case with sheared inflow is thus challenging, as multiple differences may drive the discrepancies. In addition, the current manuscript discusses the 5 MW results extensively without showing them, requiring the reader to switch between papers to follow the discussion.
- To strengthen this part, I would suggest either limiting the cross-paper comparison, or alternatively presenting a direct comparison in the present manuscript — for instance, by reproducing a nondimensionalized plot from Santoni et al. (2017):

[Figure]

**FIGURE 6** Streamwise velocity averaged in time and over the rotor disk, $\langle U_{RA} \rangle$: $\lambda = 3$ with (——) and without tower and nacelle (----); $\lambda = 6$ with (— · —) and without (------) tower and nacelle. The vertical dashed lines denotes the position of the rotor [Colour figure can be viewed at wileyonlinelibrary.com]

alongside your own:

[Figure]

Figure 4: Rotor-averaged velocity along the streamwise direction normalized by the undisturbed velocity at the rotor height, namely, $U_\infty = 10\ m/s$. The grey region represents the area covered by the rotor. (RO ----, TN ——).

in a comparable format that allows readers to interpret the differences more clearly.

Convergence study

- The grid convergence section currently relies on only two simulations. While this gives a first impression, two points are generally not sufficient to establish a clear convergence trend. I would therefore suggest either adding at least one additional

grid resolution to demonstrate a systematic trend, or explicitly referring to convergence studies from previous publications with a closely similar setup.

- In addition, the statement *"the results obtained using the coarse and fine grids are extremely close … the curves of the angle of attack are almost indistinguishable"* comes across as rather qualitative. It would strengthen the section to include a more quantitative measure of convergence (e.g. RMS difference, maximum deviation along the span, or an error norm), so readers can assess the actual magnitude of the differences. Further, the figure of the angle-of-attack comparison in the grid comparison study would be much more clear if it would be scaled to include a maximum angle of 15 degrees. The angles of attack on the cylindrical root section are not of interest to the reader.

Tip-loss models

- I also noticed that different formulations of the tip-loss model are used across the compared solvers. Specifically:
  - OpenFAST applies the classical Prandtl tip and root loss correction (l. 222).
  - The CFD-CSD solver employs the Shen et al. (2005) correction (l. 144).
- Since the Shen formulation is not equivalent to the standard Prandtl approach, the choice of model may itself contribute to differences in the results.
- For the sake of a fair comparison, you could report clearly which correction is applied in detail in each case and provide the coefficients used in the Shen model. This would make the comparison more transparent to the reader.

— Detailed Critique Points —

2 Methodologies
- 2.1 CFD-CSD Solver
  - L. 121: The subsection caption currently reads "2.1 CFD-CSD solver". From what I know, subsection titles should follow a consistent style. Please either use title case throughout (e.g., "2.1 CFD-CSD Solver") or sentence case consistently. At present, there is a mix (for example, "4 Results and Discussion" in title case versus "3 Flow and structural setup" in sentence case). Perhaps also check the WES journal guidelines, whether to use title or sentence case.
  - L. 123–125: The text states: *"The simulations of the flow around the wind turbine are carried out through Large-Eddy Simulations (LESs) of the incompressible, filtered, 3D Navier-Stokes equations, employing our in-house UTD-WF solver (Santoni et al., 2020)."* To my understanding, the first introduction of the UTD-WF code was actually in *Santoni et al. (2015), "Development of a high fidelity CFD code for wind farm control"*. The 2020 paper (*Santoni et al., 2020*) applies the solver at wind farm scale, but the code originates from the earlier work. Since UTD-WF is not as widely established as solvers like OpenFOAM, it would strengthen the methodology section if you also reference publications where the solver has been validated against either an established CFD solver of similar fidelity or experimental data.

- ○ L. 138: The manuscript currently cites *Troldborg (2009)* for the Actuator Line Method (ALM). A more appropriate primary reference would be *Sørensen and Shen (2002)*, who first introduced the ALM.
- ○ L. 144: You mention comparing the Shen tip-loss correction with the Prandtl tip-loss model. Could you please specify the chosen free parameters in the Shen model? It would also help to clarify how you ensured that both tip-loss approaches are consistent, so that observed differences in results are not simply due to discrepancies between the models themselves
- ○ L. 159–160: The structural model is attributed to *Della Posta et al. (2022, 2023)*. To provide readers with a clearer overview of how the full solver framework was assembled, it would be useful to briefly note the sequence of developments: *Santoni et al. (2015)*: introduction of the ALM–LES flow solver (UTD-WF), *Santoni et al. (2017)*: tower–nacelle representation using the immersed boundary method, *Della Posta et al. (2022)*: coupling of the aeroelastic solver, *Della Posta et al. (2023)*: implementation of the unsteady aerodynamics model (Leishman–Beddoes dynamic stall). You might phrase it along the lines of: *"The structural model used in the present study builds upon the UTD-WF framework progressively developed in Santoni et al. (2015, 2017) and further extended by Della Posta et al. (2022, 2023), where the aeroelastic solver and the Leishman–Beddoes dynamic stall model were implemented."*

3 Flow and Structural Setup
- ● L. 235: The subsection caption currently reads *"3 Flow and structural setup"*. For consistency with the other section titles, the first letter of nouns should be capitalized (e.g., *"3 Flow and Structural Setup"*). Please check the WES journal guidelines and apply the chosen style consistently throughout
- ● L. 248–249: The text states: *"The turbine location is 4 diameter units from the inlet and centered in the spanwise direction."* Could you please elaborate on whether this positioning is sufficient to ensure that there is no unintended influence between the turbine and neighboring boundaries? It would be helpful to explain how these dimensions were chosen—either by referencing a small domain-geometry convergence study (perhaps in the appendix) or by citing relevant literature that uses a similar domain setup.
- ● L. 250: The manuscript notes: *"Furthermore, we impose (...) a radiative outlet boundary condition."* This boundary condition type is relatively uncommon in CFD literature. A short explanation of its formulation would therefore be valuable for readers. For instance, *Santoni et al. (2017)* also use this condition and explicitly provide the form of the equation: $\frac{\partial \tilde{U}_i}{\partial t} + C \frac{\partial \tilde{U}_i}{\partial n} = 0$. To my knowledge, this BC is commonly referred to as "convective boundary condition" If you wish to use the term "radiative boundary condition," it would be helpful to clarify this naming choice and cite *Santoni et al. (2017)* for context.

4 Results and Discussion
- ● All captions of the subsections, similar to the caption of chapter three, should have the first letter of the nouns in capital letters. So instead of "4.1 Flow analysis" -> "4.1 Flow Analysis" and the same for 4.2, 4.3, and 4.4

- 4.1 Flow Analysis
  - L. 280–283: The manuscript states: *"Contrary to what Santoni et al. (2017) observed in their work on the 5MW reference turbine invested by a uniform inflow, the rotor-averaged velocity for the TN configuration in the wake remains slightly lower than for the OR case, indicating that wake recovery is slightly hindered by the presence of the tower."* It is useful to compare with *Santoni et al. (2017)*, but in order for readers to interpret the comparison correctly, it would be important to also highlight the notable differences between the setups (e.g., *Santoni et al. (2017)* used no-slip boundary conditions at all walls — top, bottom, and lateral — and neglected aeroelastic effects). Furthermore, I recommend caution in how the observed results are phrased. Since this finding is based on a single CFD-based aeroelastic solver, it would be safer to state that wake recovery *appears* to be hindered by tower presence, but that further validation is required as the result does not fully align with expected aerodynamics. In the following sentence, the text currently shifts from observation to causality: *"The reason for this behavior can be found in the different aspect ratio of the tower for the present turbine."* Here, I would suggest a more cautious phrasing such as: *"One possible explanation for this unexpected trend could be differences in the tower-to-rotor aspect ratio."*
  - L. 291: The sentence *"Figure 5 represents the time-averaged TKE for both configurations on different planes."* refers to a figure that is placed two pages later. This is not a critical issue, but if the layout allows, placing the figure closer to its first mention would improve readability.
  - L. 296–297: The manuscript states: *"This suggests that the tower does not increase the kinetic energy entrainment but it rather has a slight shielding effect on wake recovery."* Similar to my earlier comment, I would recommend avoiding the phrasing *"this suggests"* here, as it conveys a level of certainty not yet supported by validation against other solvers or experimental data. A more neutral formulation (e.g., *"This result may indicate …"* or *"In this simulation, the tower effect appears to …"*) would be more appropriate.
- 4.2 Aerodynamic loads on the blade
  - Figure 7: It may improve clarity if the angle of attack (AoA) is shown in a zoomed-in range (e.g., [0°, 15°]), as otherwise it is difficult to interpret the details. The legend also appears to be missing for this plot, which makes comparison harder. In addition, the BeamDyn AoA and forces in the most outboard two blade sections seem to approach zero, which looks unexpected. It would be helpful to add a short explanation if there is a known reason for this, or otherwise acknowledge it as an open point.
  - L. 339–341: The sentence currently reads: *"Indeed, the blade-tower interaction leads to an oscillations of the aerodynamic forces and of the incidence angle around θ = 180◦, i.e., when the blade is pointing down."* There is a grammar error here: it should read *"leads to oscillations"*.
  - Figure 8b: There appears to be a notable disagreement in F2 between CFD-CSD/OV and ElastoDyn, even though the two models agreed reasonably well on AoA. Moreover, given that CFD-CSD/OV predicts a lower AoA compared to ElastoDyn, one would expect a lower F2 compared to ElastoDyn, yet the opposite is observed. Could you please clarify this?

○ L. 353–355: The manuscript states: *"On the other hand, when torsional feedback is included, CFD-CSD/T and BeamDyn solvers agree rather well for all the quantities considered, regardless of the linearity or non-linearity of the models."* I would suggest reconsidering this phrasing. In Figure 8a there are visible discrepancies across all azimuthal angles, and in Figure 8c there are large differences at both low and high azimuthal angles. Instead of stating "agree rather well for all quantities," it might be more precise to describe where the agreement is reasonable and where significant differences remain.

○ Figure 9: To improve readability, it would be helpful to use a consistent color scheme across the colormaps, so that interpretation of the results is more straightforward.

○ L. 374–377: The manuscript notes: *"Although some mild differences can be observed in their amplitudes and phases, the frequency of these oscillations appears consistent between the two solvers and comparable with the natural frequency of the first torsional mode."* Here I would recommend two adjustments:
1. The differences in amplitude, especially for α and F3, are not "mild" but rather notable. In some cases (e.g., BeamDyn-TN), the qualitative shape also differs, showing more oscillations than CFD-CSD/T.
2. Instead of subjective wording like "mild" or "strong," it would be clearer to describe the differences in quantitative or qualitative terms (e.g., "larger amplitude," "different oscillation patterns").

● 4.3 Power and Thrust coefficients
○ L. 388–390: The manuscript states: *"The results reflect the dependency of the power and thrust coefficients on the tangential aerodynamic force F2 and the normal aerodynamic force F3 at the 80% of the blade, respectively."* For clarity and consistency, please use a single naming convention throughout. At present, F2/F3 are sometimes referred to as tangential/normal, and in other places as edgewise/flapwise. Consistency will make it easier for readers to follow.

○ L. 391–392: The text notes: *"Notice that, also here, we can observe that the drop in the Cp curve appears to be rather consistently predicted by BEM and CFD."* I would recommend revising this statement. The qualitative shape of the drop differs between the two methods: the BEM prediction exhibits notable oscillations before and after the drop, whereas these are not present in the CFD results. Thus, describing the predictions as "consistently" aligned may be misleading.

○ L. 395–399: The manuscript explains: *"Indeed, the flow induced by a thinner tower (in diameter units), as in the case of the 15-MW wind turbine, might be better described by a potential flow solution compared to the one induced by a thicker tower, as in the case of the 5-MW wind turbine, and may thus lead to the observed improved agreement between BEM and CFD results."* Since you make several direct comparisons with the results of 5MW turbine simulations, these comparisons become a central part of the discussion. To improve readability, I suggest including the relevant 5MW results directly in the figures, rather than expecting readers to switch between different papers.

○ L. 407–408: The sentence *"Overall, it can be said that the performance drop due to the passage in front of the tower is somewhat more limited for the*

*15MW NREL turbine than for the 5MW counterpart (…)"* could be made more precise. The repeated references to the 5MW turbine again suggest that the corresponding results should be explicitly shown and discussed here. Also, the phrases *"it can be said"* and *"somewhat"* are too subjective. A more precise wording might be: *"Results indicate that the performance drop relative to the corresponding average performance is smaller for the 15MW turbine than for the 5MW counterpart."* Ideally, a quantified comparison (e.g., percentage reduction or absolute values) would make this statement more rigorous. Further, the 5MW simulations were performed with uniform inflow, while the 15 MW simulations are performed using a sheared inflow. The lower wind speed in the lower part of the rotor plane means that the bottom half of the rotor plane, where the tower is located, will produce less than half of the total power. This will inevitably cause the performance drop due to the tower to be smaller relative to the total produced power. Therefore the observed difference is not only due to the change in turbine size, but also due to the change in inflow conditions.

- ○ L. 410–411: The manuscript states: *"Moreover, results seem to suggest that for very large rotors the presence of the tower may constitute a less critical issue for the blade deformations than for smaller rotors (…)"*. This phrasing is somewhat misleading. It is not possible to draw conclusions about "issues" such as fatigue without dedicated simulations or experiments. What is actually shown are differences in blade deformation predicted by aeroelastic simulations, which have not yet been validated for this case. I would therefore recommend rephrasing along the lines of: *"Moreover, the present simulation results predict that for very large rotors the tower effect on blade deformations is less pronounced than for smaller rotors."* At the same time, it would be important to acknowledge that this conclusion is based only on one 15MW sheared inflow simulation, where the tower is located in the part of the rotor plane where deflections are generally smaller, and one 5MW uniform inflow simulation, so broader generalizations are not yet possible.

- 4.4 Structural response
  - ○ L. 458–459: The manuscript states: *"The amplitude of the deformation is however consistent with that obtained by Trigaux et al. (2024) using LES."* The wording *"consistent"* is somewhat ambiguous here. It would strengthen the statement if you could clarify in which sense the amplitudes agree — e.g., whether they are similar in absolute magnitude, in relative deviation from a reference case, or primarily in qualitative trend.
  - ○ L. 474–476: The text reads: *"However, the gap between the BEM and the CFD-CSD/T curves is quite large. This can be attributed to the different aerodynamic and structural model used in BEM and LES."*
    *1.* The expression *"quite large"* is subjective; it would be clearer to either quantify the gap (e.g., percentage difference, RMS error) or replace it with a more neutral term such as *"substantial"* or *"noticeable."*
    *2.* The phrase *"this can be attributed to"* presents speculation as a fact. Since both the aerodynamic solvers (BEM vs LES) and structural solvers (modal vs torsional models) differ, the discrepancies cannot be uniquely traced to one factor. A more balanced formulation would be: *"These differences likely arise*

*from the combined effects of both aerodynamic and structural modeling approaches."*

- ○ L. 491–493: *"the second and third flapwise natural frequencies are indeed recovered by all the numerical models".* These peaks are visible, because they are at the 13th and 26th multiple of the rotational frequency (13p and 26p). They are not visible, because they are at the second and third flapwise mode, because flapwise modes have very large aerodynamic damping, which is also why the first flapwise mode is not visible.

**5 Conclusions**

- L. 522–527: The manuscript states: *"The entrainment of kinetic energy driven by the tower leads to higher turbulence levels in the near wake, but then result into a slightly decreased mixing behind the turbine, differently to what has been found for the NREL 5MW wind turbine, whose wake recovery was found to be promoted by the presence of the tower. This finding can have important implications for the aerodynamic loads on downstream turbines in wind farms and overall farm efficiency."*
This is a counter-intuitive result, as prior studies (e.g., the NREL 5MW turbine case) suggested the opposite effect. Since the observation is based on a single solver without confirmation from experiments or other CFD-based aeroelastic simulations, I recommend formulating this more cautiously. For example, instead of *"this finding can have important implications"*, you could write that *"this result requires further examination, as it appears counter-intuitive and has not yet been confirmed by other studies."* This way, the uncertainty is acknowledged while still highlighting the potential significance.
- L. 551–553: The manuscript concludes that *"CFD-CSD can capture complex aerodynamic loading and turbulent effects better than BEM."* While this is likely true in general, the specific evidence presented here — larger amplitudes at lower frequencies — does not in itself constitute proof of that statement. It would strengthen the conclusion to either rephrase more cautiously (e.g., *"In this case, the CFD-CSD solver captured larger low-frequency fluctuations than BEM"*) or to provide additional evidence that directly supports the broader claim.
- L. 562–563: The differences between CFD-CSD/T and BeamDyn are described in a way that attributes them to solver fidelity. However, since the two approaches differ in both aerodynamic fidelity (LES vs. BEM) and structural fidelity (torsional vs. modal models), it is not possible to unambiguously assign the discrepancies to one component. I recommend rephrasing along the lines of: *"The observed differences likely stem from the combined effects of differences in aerodynamic and structural fidelity, and cannot be uniquely attributed to one component alone."*

**Appendix A Grid Convergence Study for the LES Simulation**

- L. 573: To demonstrate a clear trend in grid convergence, more than two grid resolutions are typically required. With only coarse and fine grids, it is difficult to establish whether the solution is truly converging. One possible approach would be to use the finest grid as a reference and compute a root-mean-square (RMS) error relative to it. Plotting the RMS across multiple grid resolutions would then allow you to illustrate the convergence trend more quantitatively.

- L. 579: The manuscript states: *"The comparison in figure A1 shows that the results obtained using the coarse and fine grids are extremely close each other along the entire blade span."* This phrasing is somewhat vague and subjective. To make the convergence assessment clearer, it would be better to quantify the agreement, for example by reporting RMS errors or percentage deviations. Also, there is a small grammar correction: it should read *"close to each other"* rather than *"close each other."*

Appendix B. Validation of the Structural model

- Caption: Please ensure consistent use of title case or sentence case across the entire manuscript. For example, the current caption *"Appendix B. Validation of the Structural model"* mixes styles. The WES journal guidelines specify which convention should be followed — it would be good to align with that.
- L. 599–600: The manuscript states: *"The computed values of the modal frequencies appear to be consistent with the other results, although some discrepancies in the higher-order modes are observed."* Since the 6th mode (corresponding to the first torsional mode) is likely the most relevant for the present study, it would strengthen the paper to explicitly address these discrepancies. Please clarify in which sense the results are "consistent" and provide more detail on how the 6th mode compares across solvers.
- Figure B2:
  - From my understanding, the 6th mode (first torsional mode) should be central to your analysis. I would expect clustering of results for solvers without torsion degrees of freedom (e.g., purple, green) as distinct from those including torsion (e.g., red, blue, grey, black). However, this clustering is not clearly visible. Is it possible that there has been a mix-up between the 6th and 7th modes in Table B1? For instance, in the 7th mode the aeroelastic solvers without torsion do not appear, whereas they do for the 6th mode. Even if this is not the case, a short discussion of why this expected separation is not observed would improve clarity.
  - There is also a large spread in the higher-order modes. It could be informative to explicitly separate the results into torsion and no-torsion degrees of freedom. For example, ElastoDyn does not include torsional degrees of freedom, so one would expect systematic differences relative to models that do. The same applies to H2-PTNT.

Sources

- The reference *"Hansen, M. (2015). Aerodynamics of wind turbines. Routledge."* should be adapted to match the citation style required by WES. At the moment, the publisher and format do not appear consistent with standard referencing. Please check the journal guidelines to ensure proper formatting. For example, depending on the required style, the same reference would look as follows:
  - APA Style: Hansen, M. O. L. (2015). *Aerodynamics of wind turbines* (3rd ed.). Earthscan.

- ○ Harvard Style: Hansen, Martin Otto Laver, 2015. *Aerodynamics of wind turbines*. 3rd edn, Earthscan, London, UK.
- ○ Chicago Style: Hansen, Martin O. L. 2015. *Aerodynamics of wind turbines*. 3rd ed. London, UK: Earthscan.
- ○ Vancouver Style: Hansen MOL. *Aerodynamics of wind turbines*. 3rd ed. London, UK: Earthscan; 2015.

— Conclusion and a Note —

**Conclusion:** The manuscript demonstrates a strong understanding of aeroelastic wind turbine simulations and clearly represents a substantial amount of work. At the same time, there are a number of methodological and presentation aspects that require improvement before the paper can be considered for publication. For this reason, I must recommend a major revision. To make the work more robust and easier for readers to follow, I suggest paying particular attention to:

- **Completeness:** Clearly indicate how the in-house solver has been validated in prior literature; include the 5MW results that are repeatedly discussed; note limitations and differences when comparing with *Bernardi et al. (2023, 2025)*; and provide justification for the chosen domain size and boundary conditions (either by citing literature or through a convergence study that also covers domain geometry). For the grid convergence study, additional grid levels are needed to demonstrate an actual trend.

- **Details:** Ensure correct and consistent citations, apply a uniform style for captions, use consistent terminology for forces (e.g., flapwise/edgewise vs. normal/tangential), and add legends where missing.

- **Scientific phrasing:** Avoid turning unvalidated observations (e.g., comparisons between 5MW and 15MW turbines with different inflow conditions) into firm statements. Whenever possible, support claims with quantitative or qualitative measures rather than subjective wording.

- **Critical reflection:** The reduced wake recovery attributed to tower presence is unexpected in light of prior literature. This result should either be further examined, discussed in more detail, or acknowledged as uncertain rather than used as the basis for speculative conclusions.

- **Conciseness:** At 31 pages, the paper is longer than necessary for the number of findings presented. For example, the introduction (2.5 pages) and theoretical background (4 pages) could be shortened, with common equations (e.g., Navier–Stokes or ALM) omitted. Tables and figures that add little to the analysis (e.g., Table 1, Figures 6, 9, 11, 13) could be streamlined or integrated into the text. A tighter discussion that focuses on observations and plausible explanations would also improve readability.

**Personal note:** I realize my comments are detailed and may sound strict, but they are not meant to be discouraging. On the contrary, they reflect the high potential I see in your work. Addressing these points will make the paper more rigorous, credible, and citable. You are very close to a strong and notable publication

---

## Referee Comment (RC2)

[revised manuscript text omitted]

$$M\ddot{q} + [D + D^{Co}(\Omega)]\dot{q} + [K + K^{c}(\Omega) + K^{Eu}(\dot{\Omega})]q = e + e^{c}(\Omega) + e^{Eu}(\dot{\Omega}),$$
(6)

where M, D and K denote the modal structural mass, damping, and stiffness matrices, respectively, and e are the external loads expressed in modal basis. The remaining terms are inherently related to the various contributions to the acceleration in a moving FOR. Terms with the superscript Co, c and Eu are related to the Coriolis, centrifugal, and Euler accelerations, respectively. The discrete evaluation of the additional inertial terms in Equation (6) is expressed as a function only of the information known from the structural finite-element method (FEM) model at model and matrices are a finite element model of the blade based on complete beam elements with 6 Dometric Euler-Bernoulli behavior for bending in directions  $X_2$  and  $X_3$ , and linear shape functions for advance the structural dynamic equation in time, which is unconditionally stable for linear problems, and second-order ate. We assume a lumped-mass representation, and we take into account the local offset of the considering the local twist. Details about the modal analysis are provided in Appendix A.

**2.1.3 Fluid-Structure Interaction model**

The two-way coupling aeroelastic model employs the ALM sectional approach, whereby the angle of attack (AoA) and relative velocity are locally modified following the instantaneous blade motion provided by the structural dynamics. In particular, the distribution of the AoA along each blade is evaluated as a function of the velocity of the fluid, the angular velocity of the rotor, and the instantaneous elastic state of the blade. The latter is generally constructed from the deformation velocity  $u_{def} = d$  and the local vector of the deformation angles  $\theta$  (torsion and length length) derived from the structura er, which is forced by the updated aerodynamic loads. The algorithm restricts inter-field communications solely at the beginning of each RK substep, thereby ensuring optimal comp = bnal efficiency. The impact of torsional dynamics was deemed to be limited in light of the result soptained in previous studies on the effect of torsion for smaller wind turbines (Chen, 2017). In order to investigate this issue for the large rotor 15MW wind turbine, in this study we compare two different CSD models. In particular, we consider as a baseline a two-way coupling that includes the effect of blade deformation velocity as a sole variable (CFD-CSD/OV, for Only Velocity), and a more complete model including the torsional deformation in the coupling (CFD-CSD/T, for Torsional). In general, the relative velocity for a rotating blade can be defined with the following expression:

$$u_{rel} = u_{abs} - \Omega \times r_{OP} - u_{def}, \tag{7}$$

where  $u_{abs}$  is the filte elocity from the fluid solver at the actuator line,  $r_{OP}$  is the general radial vector pointing to the converged section,  $\Omega$  is the rotor rotational speed, and  $u_{def}$  is the deformation velocity of the structure at the same position. As a result, the AoA used to determine the air load coefficients is defined as follows:

$$\alpha = \operatorname{atan}\left(\frac{\boldsymbol{u}_{rel} \cdot \boldsymbol{E}_2}{-\boldsymbol{u}_{rel} \cdot \boldsymbol{E}_3}\right) - \phi - \frac{\theta_{tors}}{-\theta_{tors}} = \operatorname{atan}\left[\frac{(\boldsymbol{u}_{abs} - \boldsymbol{u}_{def}) \cdot \boldsymbol{E}_2}{\Omega r - (\boldsymbol{u}_{abs} - \boldsymbol{u}_{def}) \cdot \boldsymbol{E}_3}\right] - \phi - \theta_{tors}, \tag{8}$$
where  $\phi$  is the local twist angle of the  $\boldsymbol{\psi}$ ,  $\theta_{tors}$  is the local torsional deformation,  $\boldsymbol{E}_i$  are the

where  $\phi$  is the local twist angle of the  $v_{total}$ ,  $\theta_{tors}$  is the local torsional deformation,  $E_i$  are the unit vectors of the relative FOR rotating with the structure, and hence,  $v_2 = u_{def} \cdot E_2$  is the flapwise deformation velocity component, and  $v_3 = u_{def} \cdot E_3$  is the edgewise deformation velocity component. The simplified coupling procedure benefits from the sectional one-dimensional formulation of the ALM, which avoids the complex treatment of the fluid-solid interface with the associated kinematic and traction conditions.

**2.2 OpenFAST modules**

For comparison putes, wind turbine simulations have been also conducted using the OpenFAST solver Release v3.2.0 (July 29, 2022). OpenFAST is a widely utilized open-source numerical code developed by the NREL that combines different specialized modules for simulating the coupled aero-hydro-servo-elastic response of wind turbines. The aerodynamic computations are performed by the AeroDyn (Jonkman et al., 2015) module which is based on the BEM theory. A Prandtl loss model is applied to account for the tip and root effects. The structural module dedicated to the computation of the blade deformation is contained in the BeamDyn module, which relies on the geometrically exact beam theory and may resolve geometric non-linearities and large deflections (Wang et al., 2016).

BeamDyn has replaced the simplified ElastoDyn module, based on a modal approach and suitable for blade deformation dominated by bending. In order to compare the CFD-CSD results with a modal structural analysis, we also performed simulations using the standalone ElastoDyn module. It is worth to notice that the latter does not take into account the torsional degree of freedom, so it is to be directly compared to the CFD-CSD/OV model, which also does not account for the coupling between the torsional deformation and the angle of attack. As reported in the original manual of AeroDyn (Moriarty and Hansen, 2005), OpenFAST couples the fluid and structural solvers in a similar way to our CFD-CSD solvers. In particular, the local angle of attack is determined taking into account the local deformation velocities.

**235 3 Flow and structural setup**

blades given by N=80 equally-spaced nodes were chosen.

In this work, we consider a stand-alone IEA 15-MW wind turbine (Gaertner et al., 2020) in its 236 monopile configuration. This wind turbine has a rotor diameter D=240 m with three blades of 237 length L=117 m. Table 1 provides the main features of the turbine. 238 The computational box has dimensions  $12.5 \times 5 \times 3$  diameter units, as shown in Figure 2. More-239 over, following the content of the Appendix A, the computational box has been discretized by a stagger of grid composed of  $2049 \times 513 \times 513$  points in the streamwise, wall-normal, 240 241 and spanwise directions, respectively. The orthogonal grid is equally spaced in the streamwise and 242 spanwise directions and is stretched vertically, with a gradually wider spacing starting from the region 243 above the rotor. The grid spacing described leads to an actuator line discretized by 86 points per 244 blade. The time resolution of the LES computation is tied to the spatial resolution, as defined by 245 the stability requirements of the numerical scheme adopted. Simulations are carried out at a constant 246 Courant–Friedrichs–Lewy (CFL) number (Courant et al., 1967) CFL = 0.65, which ensures an aver-247 age time step  $\overline{\Delta t} = 0.024s$ . The turbine location is 4 diameter units from the inlet and centered in 248 the spanwise direction. Furthermore, we impose a she laminar inflow velocity profile, defined by a 249 power law with the exponent  $\alpha = 0.05$ , and a radiative power law with the exponent  $\alpha = 0.05$ , and a radiative power law with the exponent  $\alpha = 0.05$ , and a radiative power law with the exponent  $\alpha = 0.05$ , and a radiative power law with the exponent  $\alpha = 0.05$ , and a radiative power law with the exponent  $\alpha = 0.05$ , and a radiative power law with the exponent  $\alpha = 0.05$ , and a radiative power law with the exponent  $\alpha = 0.05$ , and a radiative power law with the exponent  $\alpha = 0.05$ , and a radiative power law with the exponent  $\alpha = 0.05$ , and a radiative power law with the exponent  $\alpha = 0.05$ , and a radiative power law with the exponent  $\alpha = 0.05$ , and a radiative power law with the exponent  $\alpha = 0.05$ , and a radiative power law with the exponent  $\alpha = 0.05$ , and  $\alpha = 0.05$ 250 rection, periodic boundary conditions are imposed. Moreover, slip and no-slip conditions are enforced 251 at the top and bottom boundaries, respectively. The turbine is subjected to a flow with a Reynolds 252 number  $Re \approx 10^8$  and operates at its nominal tip speed ratio (TSR) of  $\lambda = 9$ . The streamwise 253 undisturbed velocity at the hub height is constant and equal to  $U_{\infty} = 10 \ m/s$ . The simulations were 254 conducted for a time interval of 300 s over the initial transient, which corresponds to 35 revolutions 255 of the rotor. To identify the optimal configuration for the structural model, we conducted a preliminary sensitivity analysis and then validated the structural eigenfre cies with the results found in the literature. A 258 more detailed insight into this analysis is presented in Appendix B, where the structural properties 259 of this turbine are shown. Finally, a number of modes  $M_s = 15$  and a structural discretization of the 260

| Parameter              | Units | Value  |  |
|------------------------|-------|--------|--|
| Power rating           | MW    | 15     |  |
| Rotor diameter $(D)$   | m     | 240    |  |
| Rotor orientation      | _     | Upwind |  |
| Number of blades       | _     | 3      |  |
| Blade length $(L)$     | m     | 117    |  |
| Hub height             | m     | 150    |  |
| Hub radius $(R_{hub})$ | m     | 3.97   |  |
| Rated wind speed       | m/s   | 10.59  |  |
| Design tip speed ratio | _     | 9      |  |
| Maximum rotor speed    | RPM   | 7.56   |  |

Table 1: IEA 15-MW (Gaertner et al., 2020) wind turbine main features

Figure 2: Sketch of the computational box where the incoming sheared flow and the position of the turbine are highlighted.

**262 4 Results and Discussion**

263

264

265

267

268

269

This section presents the results of two set of simulations: one modeling a rotor-only configuration (RO) and the other including the tower and nacelle (TN). Furthermore, both configurations are subjected to comparative analysis using the OpenFAST submodules. Firstly, the near-wake aerodynamic characteristics and the wake recovery of both configurations determined by the CFD-CSD solvers are discussed. Then, the aerodynamic loads on the blades are analyzed and the outcomes from both solvers are compared. Finally, the overall turbine performance and the effects on the blade deformation are assessed.

272

273

274

275

278

279

280

281

282

283

284

285

286

291

292

293

**4.1 Flow analysis**

As a first step, we analyze the flow field variables, as obtained using the CFD-CSD/T solver. Figure 3 illustrates the main coherent flow structures in the field by means of an instantaneous isosurface of the Q-criterion colored by the streamwise velocity for both cases. It is evident that the presence of the tower affects the vorticity intensity distribution along the vertical direction. In particular, the occurrence of a low-velocity recirculation zone at the tower height for the TN case can be identified, which is a result of the tower shadowing (see Figure 3b). Moreover, the TN case demonstrates a more rapid dissolution of the endogenous coherent hub vortex structures if compared to the RO case (see Figure 3a). On the other hand, the tip vortex structures appear to be minimally influenced by the presence of the tower. Figure 4 shows the rotor-averaged streamwise velocity along the flow direction, time-averaged over 30 revolutions of the rotor. Contrary to what Santoni et al. (2017) observed in their work on the 5MW reference turbine invested by a uniform inflow, the rotor-averaged velocity for the TN configuration in the wake remains slightly lower than for the OR case, indicating that wake recovery is slightly hindered by the presence of the tower. The reason for this behavior can be found in the different aspect ratio of the tower for the present turbine. In particular, for the NREL 5-MW turbine, the ratio between the tower diameter and the rotor diameter is about equal to 0.047, whereas, for the 15MW turbine, it is only about 0.027 (the tower diameters being 6m and 6.5m, respectively). Thus, the thinner shape (in terms of diameter units) of the tower, as well as the lower value of the incoming velocity at the tower height due to the presence of shear at the inflow, result into a decreased mixing behind the turbine which leads to a slower wake recovery.

Figure 3: Q-criterion contour of the instantaneous velocity field colored by the streamwise velocity for the rotor-only case (RO) (a) and tower and nacelle (TN) (b).

From an energy perspective, the wake recovery process can be depicted by examining the Turbulent Kinetic Energy (TKE) in the wake. Figure 5 represents the time-averaged TKE for both configurations on different planes. The TN case exhibits high TKE values in the near wake, in the region just downstream of the tower and nacelle. The top view of the TN case shows that the TKE in the wake presents an asymmetric distribution as De Cillis et al. (2022) observed, among the others, in their

296

297

298

300

301

302

303

305

306

307

308

309

310

Figure 4: Rotor-averaged velocity along the streamwise direction normalized by the undisturbed velocity at the rotor height, namely,  $U_{\infty} = 10 \ m/s$ . The grey region represents the area covered by the rotor. (RO ----, TN ——).

work. On the contrary, the RO configuration shows large TKE only in the far wake region, with large values also in the region above hub height. This suggests that the tower does not increase the kinetic energy entrainment but it rather has a slight shielding effect on wake recovery. Although not favoring kinetic energy entrainment, the tower still plays a strong role in the wake dynamics, as it can be visualized in figure 6, showing slices of instantaneous streamwise velocity at different tower heights corresponding to 80% of the blade (top) and to the tip of the blade (bottom), when the blade is in front of the tower, i.e.  $\theta = 180^{\circ}$  (left), and when it is far from it (right). In particular, it can be observed that the turbulent mixing right downstream of the tower is already very high in the near wake compared to that close to the tip of the blades. Probably due to asymmetry induced by the rotation of the blades, inside the rotor disk, it can be seen that the tower wake bends in the spanwise direction (Figure 6, top frames), whereas it is rather spanwise independent at a height corresponding to the blade's tip (bottom frames). Moreover, one can see that the passage of the blade in front of the tower (left frames) induces a strong perturbation in the flow field already upstream of the tower. In the following section, the effect of this perturbation on the phase oscillations of several relevant quantities (aerodynamic forces, power coefficient, etc.) will be discussed.

**4.2 Aerodynamic loads on the blade**

The analysis of the aerodynamic loads on the blade has been conducted using the present CFD-CSD 311 models and the engineering software OpenFAST. The same laminar sheared inflow is imposed for both 312 solvers using a power law with the same exponent and reference streamwise velocity at the hub height. 313 We have chosen not to impose a turbulent inflow to avoid differences in the definition of the turbulent 314 inflow itself which might have hindered the comparison between the results of the two codes. 315 Figure 7 depicts the following time-averaged aerodynamic quantities along the span of the blade: the 316 local angle of attack  $\alpha$  (Figure 7a); the aerodynamic pitching moment per unit length  $M_{aero}$  (Figure 317 7b); the flapwise and edgewise components o aerodynamic force per unit length  $F_2$  (Figure 7c) 318 and  $F_3$  (Figure 7d), respectively. In particular, Figure 7a shows that a goo eement of the local 319

321

324

325

Figure 5: Top (upper slices) and lateral (lower slices) views of the time-averaged Turbulent Kinetic Energy on slices passing through the hub. TN (left), RO (right).

Figure 6: Instantaneous streamwise velocity on horizontal slices at different tower heights corresponding to  $80^{\circ}$  he blade (top slices), and the tip of the blade (bottom slices). In the left configuration, the blade is in front of the tower ( $\theta = 180^{\circ}$ ), while on the right the blade is far from the tower.

incidence angle computed by both CFD-CSD models (solid lines) with that computed by ElastoDyn (circles) and BeamDyn (squares) is obtained from the 20% up to the 80% of the blade length. Indeed, the differences in the root area are ascribable to the presence of the hub which is modeled differently by the solvers. The discrepancy of the incidence angle observed towards the tip subsequently affects the aerodynamic loads. The  $F_2$  force in Figure 7c shows a very good fit of the CFD-CSD/T results with that of the nonlinear solver BeamDyn, despite the linearity of our in-house CSD model. The strong discrepancies with respect to the values obtained by ElastoDyn can be ascribed to the absence

of the torsional deformation in the latter solver. Indeed, the CFD-CDS/OV solver, which neglects the torsional feedback in the coupling, shows very similar results to the ElastoDyn solver. A similar effect can be observed by examining the reduction in  $F_3$  towards the tip of the blade (see Figure 7d). The distribution of the aerodynamic pitching moment presents instead a maximum gap of about 8% from the BEM-based solvers.

Figure 7: Average aerodynamic quantities along the blade compared between CFD-CSD/OV (black solid line), CFD-CSD/T (blue solid line), ElastoDyn (red rounded markers), BeamDyn (magenta squared markers). (a) ence angle, (b) Aerodynamic pitch moment, (c) flapwise aerodynamic force, (d) edgewise aerodynamic force.

As demonstrated by Hansen (2015), the outer third of the blade span is the most critical region in terms of deflections and deformations due to the combination of higher aerodynamic loads and reduced structural stiffness. Therefore, a phase average of the aerodynamic quantities at the 80% of the blade has been performed. Figure 8 reports the evolution of the incidence angle and of the aerodynamic force components at  $\frac{r-R_{hub}}{L}=0.8$  (being  $R_{hub}$  the hub radius and L the blade length) versus the blade rotation angle  $\theta$ . The dynamical behavior of the aerodynamic quantities in the presence (solid lines) or in the absence (dashed lines) of the tower underlines that the passage of the blade in front of the tower represents the main source of instability for the flow conditions considered. Indeed, the blade-tower interaction leads to an oscillations of the aerodynamic forces and of the incidence angle around  $\theta=180^{\circ}$ , i.e., when the blade is pointing down. However, unlike the case of the NREL 5-MW turbine (Bernardi et al., 2023), this effect appears to be stronger for the BEM computations than for the CFD-CSD solver. Concerning this point, we should recall that, as pointed out by Bernardi

Figure 8: Phase-averaged values of: (a) the local incidence angle, (b) flapwise aerody angential aerodynamic force at the 80% of the blade. CFD-CSD/OV: TN -CFDCSD/T: TN —, RO ----. ElastoDyn: TN —, RO ----. BeamDyn: TN —

et al. (2023), the complex flow dynamics resulting from the interaction between the blade and the tower, shown in Figure 6, may not be well described by OpenFAST, which uses a simple potential 345 flow model. It can be observed that, between the rotor and the tower, a region with low streamwise 346 velocity is observed. We can expect that the passage of the blade in front of the tower thus induces 347 an alteration of the aerodynamic forces on the blade due to the decrease/increase of the streamwise 348 velocity. This issue will be further discussed in the following, where a possible reason for the different 349 behavior observed for the IEA 15-MW with respect to the NREL 5-MW turbine will be discussed. 350 Apart from the effect of the tower, one can observe a rather good match between the CFD-CSD/OV 351 and ElastoDyn solvers for both the incidence angle and the edgewise component of the aerodynamic 352 353 force, while the flapwise component presents some discrepancies. On the other hand, when torsional feedback is included, CFD-CSD/T and BeamDyn solvers agree rather well for all the quantities con-354 sidered, regardless of the linearity or non-linearity of the models. 355 To better investigate the local response of the different models during the blade revolution, we con-356 ducted a comparative analysis of the aerodynamic loads, employing phase-averaged quantities over 357 the span. Figure 9 illustrates the percentage difference of the phase-averaged aerodynamic quantities on the rotor plane of the ElastoDyn (BeamDyn) solver with respect to the CFD-CSD/OV (CFD-359 CSD/T) model, respectively. In particular, in comparison to ElastoDyn, a higher value of the absolute 360 incidence angle in the range of  $\Delta \alpha / \alpha^{CFD-CSD/OV}$  | = [17%, 25%] is found in the zone after the tower (see Figure 9a). The difference we spect to the results obtained by BeamDyn tends to be higher moving from the root to the tip with a discontinuity in the tower area, spanning the range 361 362 363  $|\langle \Delta \alpha / \alpha^{CFD} \rangle^{\%}| = [35\%, 60\%]$  in the last 20% of the blade span. Furthermore, the angle of attack 364 distribution affects the components of the aerodynamic force. In fact, the distribution of the flapwise 365 component of the force follows the same pattern of the incidence angle (see Figure 9b). On the other 366 hand, for the edgewise component the major discrepancies are concentrated in the final radial sections 367 of the blade toward the tip (see Figure 9c). In general, we can conclude that the most significant 368 discrepancies are observed in the tip region where the three-dimensional effects are more relevant and 369 where the complexity of the fluid flow is strongly affected by the presence of the tower.

Figure 9: Phase-averaged contour plots of the percentual differences of the aerodynamic quantities between CFD-CSD/C rsus ElastoDyn (left), and CFD-CSD/T versus BeamDyn (right), respectively. (a) Incidence angle, (b) flapwise aerodynamic force, (c) edgewise aerodynamic force.

Notably, similar discrepancies are observed when comparing the CFD-CSD/T solver with the Beam-371 Dyn solvers. However, in this case some high-frequency oscillations are observed for the three aero-372 dynamic quantities. In fact, the same oscillations are observed in the phase averaged quantities at 80% of the blade shown in Figure 8, for both the CFD-CSD/T solver and BeamDyn. Although some 374 mild differences can be observed in their amplitudes and phases, the frequency of these oscillations 375 appears consistent between the two solvers and comparable with the natural frequency of the first 376 torsional mode. Again, this observation indicates that including the torsional degree of freedom in 377 the structural solver is crucial for describing accurately the amplitude and dynamical behaviour of the 378 aerodynamic quantities.

**4.3 Power and Thrust coefficients**

The aerodynamic loads previously presented are also useful to evaluate the power and thrust coefficients, defined as follows:

$$C_p = \frac{P_d}{\frac{1}{2}\rho A U_\infty^3}, \quad C_t = \frac{T_{aero}}{\frac{1}{2}\rho A U_\infty^2},\tag{9}$$

385

386

387

388 389

390

391

392

393

395

396

397

398

400

401

402

403

404

406

407

408

Figure 10: Phase-averaged power (a) and thrust (b) coefficients. CFD-CSD/OV: TN ——, RO ---- CFD-CSD/T: TN ——, RO ---- . ElastoDyn: TN ——, RO ---- . BeamDyn: TN ——, RO ----

where  $A = \pi D^2/4$  represents the rotor area,  $P_d$  is the aerodynamic power transferred to the rotor and  $T_{aero}$  is the overall aerodynamic thrust on the turbine.

Starting from the time history of  $C_p$  and  $C_t$ , we computed their phase-averaged evolution as reported in Figure 10. The periodic passage of the blades in front of the tower for the TN configuration produces a drop of the curves of about 10%. Eventually, the performance is restored to the value obtained in the RO case through a dynamical behavior consistent with the elastic nature of the structure. The results reflect the dependency of the power and thrust coefficients on the tangential aerodynamic force  $F_2$  and the normal aerodynamic force  $F_3$  at the 80% of the blade, respectively (see Figures 8c and 8b), which are strongly influenced by the presence of the tower. Notice that, also here, we can observe that the drop in the  $C_p$  curve appears to be rather consistently predicted by BEM and CFD. The opposite was observed for the NREL 5-MW turbine (Bernardi et al., 2023), where this performance drop is considerably underestimated by the BEM computations. A possible factor that may contribute to this different behaviour may reside in the different relative geometry of the two wind turbines. Indeed, the flow induced by a thinner tower (in diameter units), as in the case of the 15-MW wind turbine, might be better described by a potential flow solution compared to the one induced by a thicker tower, as in the case of the 5-MW wind turbine, and may thus lead to the observed improved agreement between BEM and CFD results. Moreover, the differences in the flow impinging on the blade might also have an effect. In fact, in Bernardi et al. (2023) a uniform inflow was imposed. Whereas, in the present case, due to the shear imposed at the inflow and the limited distance from the ground of the tip of the blade (only  $\approx 0.125D$  for the 15MW turbine), the blade is invested by a flow having a much smaller velocity compared to the given value of  $U_{\infty}$  at hub height, further confirming the increased suitability of a potential flow solution upstream of the tower. Nevertheless, we should recall that this remains a very strong approximation, as also demonstrated by the differences in the forces and angles that have been observed in the previous section (see Figure 9, for instance).

Overall, it can be said that the performance drop due to the passage in front of the tower is somewhat more limited for the 15MW NREL turbine than for the 5MW counterpart, and it is more consistently

Figure 11: Power Spectral Density (PSD) of the power (a) and thrust (b) coefficients. The vertical dashed lines highlight the rotational frequency of the rotor  $P = f_{rot}$  and the multiples of 3P, respectively. CFD-CSD/OV ——, CFD-CSD/T ——, ElastoDyn ——.

predicted by BEM theory and CFD.

Moreover, results seem to suggest that for very large rotors the presence of the tower may constitute a less critical issue for the bla formations than for smaller rotors, although it should yet be taken into account for accurately averabing the turbine's performance oscillations as it still represents a major source of unsteadiness.

The average value of the power coefficient is much larger when the torsional deformation is neglected. This feature is consistently observed by both CFD and BEM approaches. However, one can observe that ElastoDyn underestimates the value of  $C_p$  with respect to the corresponding non-torsional CFD model, while the opposite is observed when comparing BeamDyn with the torsional CFD solver. This is most probably due to the fact that BeamDyn predicts higher values of the aerodynamic tangential forces with respect to the CFD-CSD/T approach, which are linked to a smaller torsional deformation as will be shown in figure 12f in the next section.

as will be shown in figure 12f in the next section. Figure 11 shows the premultiplied Power Spectral Density (PSD) of the power (Figure 11a) and thrust (Figure 11b) coefficients evolution. The PSD is normalized by the variance of each coefficient  $\sigma^2$  and plotted versus the frequency normalized by the rotational frequency of the rotor.  $f/f_{rot}$ . In both cases, the CFD-CSD solvers seem to provide a richer representation of the aer mic coefficients, capturing the full range of flow-structure interactions. Indeed, an examination of the low-frequency behavior reveals that both quantities exhibit isolated low-frequency peaks when using the BEM-based solvers, a phenomenon not observed with the CFD-CSD, where the low-frequency range is rather

broadband and does not present particular peaks. It is important to notice that the frequency 1P can be directly linked to the frequency of the passage of the blade in front of the tower, but also to wind shear loads on the blades. Concerning the first point, a potential flow solution as that used in the BEM solver is keen to provide a simple, single-frequency response, whereas a complex, turbulent flow 431 is expected to result in a more broadband spectrum. Concerning the second point, we have to consider 432 that in LES, the power law profile is imposed at the of the domain but it is free to evolve for 433 4 diameters before the wind turbine, altering in a non-trivial way the flow field and the consequent 434 frequency response of the blades. This outcome indicates that the BEM-based solvers tend to overcut the power oscillations associated with low-frequencies that are not exactly equal to 1P or 2P. For all solvers, however, the strongest PSD peaks are to be found at much larger frequencies (3P-6P-9P-12P), 437 as also observed by Pagamonci et al. (2023) by means of URANS aeroelastic simulations of the NREL 438 5-MW, the DTU 10-MW, and the IEA 15-MW turbines. One can also notice that the amplitude 439 associated with the 3P frequency appears to be consistently described by the two solvers, although also in this range the BEM solver appears to overdamp the frequencies in between different peaks. Moreover, a good agreement is evident between the two set of results concerning the value of the 442 frequencies and the level of the PSD for frequencies that are multiples of 3P. 443

**4.4 Structural response**

444

This section presents the analysis of the structural dynamics. Figure 12 reports the phase-averaged dynamic response of the free extremity of the blade (left column) tion of the entire span (right column). Figure 12a shows how the the time-averaged deformalane deformation is mainly governed by the aerodynamic component of the force normal to the rotor plane and, hence, to the 448 aerodynamic effects, heavily affected by the tower. In fact, it is visible how the tower placed at 449  $\theta = 180^{\circ}$  produces a drop in the deformation, followed by an elastic dynamic response which restores 450 the value far from the pointing-down position. The time-averaged maximum deformation predicted 451 by the CFD-CSD/OV solver is 16% higher compared to the ElastoDyn module and 17% compared to BeamDyn (see Figure 12b). On the other hand, the same quantity predicted by the CFD-CSD/T 453 solver is 17% lower compared to the ElastoDyn module and 13% compared to BeamDyn (see Figure 454 12b). This is consistent with the fact that including the torsional degree of freedom reduces the loads 455 (see figure 8b) and the resulting deformation. Although the trend of deformation with respect to the blade span appears consistent with previous predictions based on URANS (see Pagamonci et al. (2023)), the out-of-plane deformation is rather larger, reaching 16 m at the blade's tip. The amplitude 458 of the deformation is however consis with that obtained by Trigaux et al. (2024) using LES. Figure 459 12c depicts instead the in-plane determation, which is mostly due to gravity. The results show that 460 the shadowing effect of the tower does not influend | squantity. Furthermore, the discrepancies 461 obtained between ElastoDyn and BeamDyn can be a tted to the lack of modes used by the former 462 463 model to describe the translation in the edgewise direction (see Figure 12d). discrepancy does not seem to be linked to the linearity of this model, as the result of the CFD T solver, which 464 is linear as well, is much closer to the BeamDyn results. Moreover, the results of the CFD-CSD/OV 465 and the CFD-CSD/T models are very close to each other. It can be noticed that the amplitude of the 466 oscillation of the in-plane deflection is consistent with that reported by ux et al. (2024), although 467 the sign is opposite due to the different frame of reference used.

Figure 12: Phase-averaged deflections at the tip of the blade (left column) and time-averaged deflections along the blade span (right column). CFD-CSD/OV: TN ——, RO ----. CFD-CSD/T: TN ——, RO ----. ElastoDyn: TN ——, RO ----. BeamDyn: TN ——, RO ----.

A further significant insight into the deformation phenomenon is provided by the torsional DoF. Figure 12e shows a comparison of the torsional angle at the tip with *BeamDyn*. Significant discrepancies can

be observed between the LES and the BEM approaches, which cannot be reconducted to the different

473

474 475

476

477

478

479

480

Figure 13: Power Spectral Density (PSD) of the out-of-plane (a), in-plane (b), and torsional (c) deformation of the blade. The vertical dashed lines represent the first 8 eigenfrequencies of the system. CYD-CSD/OV —, CFD-CSD/T — ElastoDyn —, BeamDyn —.

coupling procedures adopted by the models. On the one hand, BeamDyn and CFD-CSD/T both take into account the deformation angle in the coupling (Wang et al., 2016), while in the CFD-CSD/OV solver the angle of attack depends only on the deformation velocity (see Equation 18). However, the gap between the BEM and the CFD-CSD/T curves is quite large. This can be used to the different aerodynamic and structural model used in BEM and LES. The latter is confirmed by the time-averaged torsional deformation along the space ported in Figure 12f where the maximum percentual gap of BeamDyn reaches 29% for the CSD-OCD/OV, and 24% for the CFD-CSD/T. It is noteworthy that the lower torsional deformation resulting from BeamDyn leads to the higher aerodynamic loads observed in figure 8c.

Finally, figure 13 illustrates the Power Spectral Density (PSD) of the blade's tip deformation components for the TN configuration (which is characterized by more complex fluid-structure interactions). The premultiplied PSD values are normalized by the variance of the signal,  $\sigma^2$ , and plotted versus the frequency normalized by the rotor frequency,  $f/f_{rot}$ . Spectral results have been corroborated through use of the Welch and Lomb-Scargle PSD estimation algorithms. 485 Figure 13a shows the out-of-plane deformation, which we showed to be influenced mostly by the 486 aerodynamic loading. The results indicate that, for all the numerical approaches used, the observed 487 structural response does not ex a peak corresponding to the first flapwise natural frequency, 488 suggesting that the intrinsic dy so of the structure might play a less prominent role in the deformation process. This is consistent to the results of Trigaux et al. (2024) (see Fire 6 of the cited paper) for the same turbine and similar inflow conditions. However, the second third flapwise 490 491 natural frequencies are indeed recovered by all the numerical models, despite their contribution has a 492 limited energy content. Both the CFD-CSD solvers show larger amplitudes for a broadband range of 493 frequencies than OpenFAST, suggesting that they are more able to capture complex flow interactions, including turbulence-induced vibrations. This effect is particularly pronounced at the lower frequencies, probably due to the large-scale three-dimensional structure of the flow impinging on the turbine, 496 which is not captured by OpenFAST. These aspects seem to be under-represented in the ElastoDyn 497 . Although the ElastoDyn curve aligns with both the CFD-CSD solvers at and BeamDun solu 498 some key frequency veaks, it does not fully account for the fine-scale flow-structure interactions. On 499 the other hand, the BeamDyn curve provides better agreement with the CFD-CSD solvers, especially 500 at higher frequencies near the blade's natural modes, suggesting that BeamDyn captures more of the 501 structural dynamics, particularly the aeroels response. Figure 13b shows the in-plane deforma-502 tion, which is primarily influenced by gravity, centrifugal, and Coriolis forces acting on the blade. The 503 CFD-CSD solvers again demonstrate stronger low-frequency components. 504 Figure 13c presents the torsional deformation for the CFD-CSD/T and BeamDyn solvers, excluding 506 ElastoDyn, which neglects the torsional DoF in the model. Additionally, also this quantity demonstrates that the CFD-CSD curves predict higher amplitudes at low frequencies. However, a good 507 agreement between the two solvers is evident at higher frequencies, especially in the range around the 508 first torsional eigenfrequency. 509

**5 Conclusions**

510

This study investigated the aeroelastic response of the IEA 15-MW wind turbine by employing a 511 high-fidelity Computational Fluid Dynamics (CFD) solver that couples Large-Eddy Simulation (LES) 513 with a Computational Structural Dynamics (CSD) solver. Two different CSD solvers are considered: the CFD-CSD/OV solver, in which the only structural quantity contributing to the definition of the 514 angle of attack is the deformation velocity, and the CFD-CSD/T solver, in which the instantaneous 515 torsional deformation is also considered when defining the local effective incidence. The results of the 516 two CFD-CSD solvers are compared with those of traditional engineering solvers such as BeamDyn 517 and ElastoDyn, both relying on Blade Element Momentum (BEM) theory. Two case studies were 518 examined: a rotor-only configuration (RO) and one that included the tower and nacelle (TN). 519 In the first instance, a flow analysis uncovered important considerations regarding wake entrainment. 520 In particular, the study found that for the considered turbine, impinged by a laminar sheared inflow,

wake recovery is only slightly hindered by the presence of the tower. The entrainment of kinetic energy driven by the tower leads to higher turbulence levels in the near wake, but then result into a slightly decreased mixing behind the turbine, differently to what has been found for the NREL 5MW wind turbine, whose wake recovery was found to be promoted by the presence of the tower. This finding can have important implications for the aerodynamic loads on downstream turbines in wind farms and overall farm efficiency.

In addition, the Power Spectral Density (PSD) of the power and thrust coefficients revealed that the 528 CFD-CSD solver captures a broader range of flow-structure interactions, with a more broadband low-529 frequency response, compared to the BEM-based solvers. The isolated low-frequency peaks found in 530 BeamDyn and ElastoDyn suggest that these solvers tend to over-simplify the aerodynamic fluctuations 531 associated with phenomena such as wind shear and tower shadowing. For the IEA 15-MW turbine, 532 the performance drop caused by tower passage is less pronounced compared to the data available for 533 smaller turbines such as the NREL 5-MW. The BEM-based solvers show a less pronounced discrepancy with the CFD-CSD predictions, although in this case the resulting oscillations appear overestimated by BEM, while they were underestimated for the 5-MW wind turbine as observed by Bernardi et al. 536 (2023). This may be due to the thinner tower of the 15-MW turbine relative to its rotor diameter, 537 and to the different inflow conditions (sheared in the present case, uniform in previous studies on the 538 5-MW turbine). 539

Concerning the forces on the blade and the incidence angle, one can observe a rather good match between the CFD-CSD/OV solver and *ElastoDyn*, as well as between the CFD-CSD/T model and the *BeamDyn* solver. This is likely due to the presence – or not – of the torsional feedback, while non-linearities of the structural solver appear to have only a limited impact on the observed quantities. In agreement with previous studies, the results thus suggest that including the torsional degree of freedom in the structural solver is crucial for accurately describing the amplitude and dynamical behaviour of the aerodynamic quantities.

Moreover, it is observed that duly taking into account the torsional degree of freedom reduces the value of  $C_p$ . This feature is consistently observed by both CFD and BEM approaches. However, one can observe that BeamDyn predicts lower values of the torsional deformation and thus higher values of the aerodynamic tangential forces with respect to the CFD-CSD/T approach, leading to a larger  $C_p$  value than that predicted by LES. The CFD-CSD solvers tend to exhibit larger amplitudes at lower frequencies with respect to BEM ones, suggesting a stronger capability to capture complex aerodynamic loading and turbulence effects.

The structural response of the wind turbine blade has been assessed by comparing the out-of-plane, inplane, and torsional deformations obtained from the CFD-CSD solvers, *ElastoDyn*-based, and *Beam-Dyn*-based OpenFAST solver. In-plane deformation, influenced significantly by centrifugal forces, appears to be better captured by the CFD-CSD solvers, especially in the low-frequency range. Concerning the out-of plane deflection, large discrepancies are seen between the two CFD-CSD solvers, as well as between both BEM modules and the LES.

Our results underscore the importance of incorporating torsional deformation effects in the definition of the angle of attack and using high-fidelity aeroelastic models to ensure accurate predictions of wind turbine blade performance with a richer fluid dynamics. Whereas, the linearity of the structural model does not appear to have a strong effect on the aerodynamical quantities, deformations and loads. In

574

575

576

581

582

583

586

587

general, the comparison of the results of the CFD-CSD solver with those of the engineering solver shows differences especially in the region behind the tower. This suggests that the use of a highfidelity CFD approach may be crucial for determining the effect of the tower, the dynamic response of the turbine blades and the wake recovery process. Future work will explore the effect of turbulent fluctuations at the inlet to better investigate the impact

Future work will explore the effect of turbulent fluctuations at the inlet to better investigate the impact of the atmospheric boundary layer on the aerodynamic forces, loads and deformations of the present turbine.

**571 A Appendix A. Grid Convergence Study for the LES Simulation**

A grid convergence study was conducted to evaluate the sensitivity of the LES results to spatial and temporal resolution. Two simulations were carried out using grids of different densities: a coarse mesh and a finer mesh, with the latter having approximately 40% more grid points than the former in each spatial direction. This allowed for a more detailed resolution of flow structures and aerodynamic quantities. Moreover, the simulation ran with the finer grid uses the same CFL = 0.65 as the coarse grid one. The average time step obtained and the other key parameters of the two LES runs are summarized in Table A1.

| Parameter                  | Coarse Grid          | Fine Grid            |
|----------------------------|----------------------|----------------------|
| Total number of cells      | $5.37 \times 10^{8}$ | $1.36 \times 10^{9}$ |
| Largest cell diagonal (m)  | 5                    | 3.5                  |
| Smallest cell diagonal (m) | 2.5                  | 1.7                  |
| Actuator points per blade  | 86                   | 128                  |
| Average time step (s)      | 0.024                | 0.012                |
| Total number of threads    | 512                  | 768                  |

Table A1: Comparison of the main parameters for the coarse and fine meshes.

The comparison in figure A1 shows that the results obtained using the coarse and fine grids are extremely close each other along the entire blade span. In particular, the curves of the angle of attack are almost indistinguishable, even in the outer portion of the blade, where stronger differences were expected due to tip effects and local three-dimensionality. The maximum deviation of the incidence angle  $\alpha$  between the two simulations at 80% of the span reaches a value of  $\Delta \alpha_{max} \approx 0.2^{\circ}$ , corresponding to a relative difference of 1.6%. Similarly, the aerodynamic force component distributions exhibit negligible variation between the two resolutions, confirming the overall consistency of the LES solution examined in the Sec. 4 with respect to mesh refinement.

These results indicate that the coarse grid accurately captures the main aerodynamic features, making the use of a finer mesh unjustified given its higher computational cost and minimal accuracy gain.

Figure A1: Average aerodynamic quantities along the blade obtained from the coarse grid (black line), and the thick grid (green line). (a) Incidence angle, (b) Aerodynamic pitch moment, (c) flapwise aerodynamic ford edgewise aerodynamic force.

**B Appendix B. Validation of the structural model**

A preliminary study was conducted to validate the structural model prior to coupling it with the CFD solver. Figure B1 shows the distributions of the structural and constructive properties along the blade, which were utilized as input for the modal CSD analysis. A convergence study to determine the proper number of elements,  $N_e$ , (not reported here for brevity) was conducted, leading to the choice  $N_e = 80$ . Furthermore, the results of the present structural analysis were compared with those of five models including: the prismatic Timoshenko model without torsion (H2-PTNT); the Timoshenko model with a fully populated stiffness matrix (H2-FPM) from the study of Rinker et al. (2020); the 3D Finite Element Analysis (3D FEA) selected from Zhang et al. (2023); the ElastoDyn model; the BeamDyn model. Figure B2 shows the first 8 eigenfrequencies using the present method compared with the results of these models. The computed values of the modal frequencies appear to be consistent with the other results, although some discrepancies in the higher-order modes are observed. Moreover, an analysis of the most important modes was conducted: Table B1 provides the classification of the first 8 modes, whereas, Figures B3, B4, and B5 show the modal displacements for the first spanwise, edgewise, and torsional modes, respectively.

Figure B1: Structural properties of the blade along the span: (a) stiffness, (b) inertia, (c) density, (d) local twist angle.

Figure B2: A comparison of the eigenfrequencies computed by different structural models.

| # | $f_n[Hz]$ | Mode          |
|---|-----------|---------------|
| 1 | 0.5369    | 1st flapwise  |
| 2 | 0.7267    | 1st edgewise  |
| 3 | 1.577     | 2nd flapwise  |
| 4 | 2.267     | 2nd edgewise  |
| 5 | 3.113     | 3rd flapwise  |
| 6 | 3.642     | 1st torsional |
| 7 | 4.571     | 3rd edgewise  |
| 8 | 5.385     | 4th flapwise  |

Table B1: Classification of the first 8 structural modes.

Figure B3: Mode 1 shape for all the DoFs.

Figure B4: Mode 2 shape for all the DoFs.

Figure B5: Mode 6 shape for all the DoFs.

- 604 Author contribution. CB: Investigation, Writing Original draft, Formal analysis, Methodology, Soft-
- ware, Validation. SC: Conceptualization, Investigation, Writing Review & Editing, Supervision. FM:
- Methodology, Software, Validation. GDP: Formal analysis, Writing Review & Editing, Methodol-
- 607 ogy, software. SL: Conceptualization, Software, Supervision. PDP: Conceptualization, Investigation,
- 608 Writing Review & Editing, Supervision.
- 609 Competing interests. The authors declare that they have no competing interests.
- 610 Acknowledgments. This study has been partially funded under the National Recovery and Resilience
- 611 Plan (NRRP), Mission 4 Component 2 Investment 1.3 Call for tender No. 1561 of 11.10.2022
- 612 Project code PE0000021, Project title "Network 4 Energy Sustainable Transition NEST", and
- under the PRIN grant 20229YJP33, "Diffuser augmented Wind Turbines for URBban environments"
- 614 (DWTURB). Both are grants of Ministero dell'Università e della Ricerca (MUR), funded by the
- European Union NextGenerationEU. The cooperative work on the 15MW NREL wind turbine
- within the IEA WIND TASK 47 TURBINIA is also acknowledged.

   performance of wind turbines by elastic actuator line model. Applied Energy 330, 120361.

---

## Author Comment (AC1)

**Point-to-point reply to reviewers**

Paper wes-2025-120

Title:

**Large Eddy Simulation of the IEA 15-MW Wind Turbine Using a Two-Way Coupled Fluid-Structure Interaction Model**

September 26, 2025

The authors wish to thank the reviewers for their comments and suggestions on the paper to enhance the manuscript. The paper has been revised accordingly. The revised parts have been highlighted in red in the updated manuscript. If mentioned, the page number refers to the revised paper. In the following we report point-to-point replies concerning the questions and comments raised by the reviewers.

**Reviewer #1**

**Main Critique Points**

*Documenting the software and its validation status.*

- *As a reader, I found it a bit hard to trace how the different pieces of your in-house UTD-WF framework were developed across prior work, and what has been validated under which conditions. This is important to know that the in-house code you use - i.e., the foundation for all your work and insights - is credible and correct in the first place. Adding a short provenance/validation paragraph would strengthen the manuscript.*

  A paragraph about the provenance of the in-house code and the validation of its different version has been now added at the beginning of Section 2.1.1 (red paragraph on page 4).

- *Clarifying which subsets constitute formal validation would help readers map prior evidence to your present setup. In that light, two clarifications would make your contribution easier to interpret:*

  1. *Scope of prior validation vs. present cases: Prior validations include uniform and ABL inflows for a 5 MW turbine; your study addresses the IEA-15 MW case for a sheared laminar inflow configuration. Since larger rotors can amplify aeroelastic effects, it would be helpful to note explicitly that the present application extends the validated setting and to discuss any implications or arising uncertainties.*

     We have now included a small paragraph discussing this important point in Section 2.1.1 (last lines of the red paragraph on page 4). In particular, the added lines are as follows: "Notice that prior validations by (Della Posta et al. 2022) of the CFD-CSD solver were made on a laminar uniform and a turbulent sheared inflows for a 5 MW NREL turbine, whereas our study extends the validated setting to the IEA-15 MW case for a sheared

laminar inflow configuration. However, as discussed in the framework of the IEA Wind TCP Task 47 (Cacciola et al., 2025), turbulent fluctuations appear to have a much stronger impact than shear on load response of aero-elastic numerical codes. Moreover, high-fidelity codes appear rather consistent in predicting loads, while engineering models tend to overpredict fatigue loads, particularly for large rotors".

2. *Citations for solver provenance: Where you cite the solver's origin, consider also referencing Santoni et al. (2015) as the first UTD-WF publication, prior to Santoni et al. (2020). Further mentioning the following papers and validations that were added to Santoni et al. (2015) would help readers that are unfamiliar with the code's history. Just so they know that the inhouse code you use is validated within a certain range.*

We have now used Santoni et al. (2015) as first reference to the in-house code.

*Comparing BEM with CFD for aeroelastic simulations*

- *The conclusions drawn about the relative suitability of CFD-based versus BEM-based aeroelastic simulations are not entirely straightforward, because the setups being compared differ in more than one respect. In particular:*

  - *Case A (CFD-CSD) couples a finite-volume LES solver with ALM to a linear structural model (A.1: OV; A.2: TN).*
  - *Case B (OpenFAST with BeamDyn) uses BEM aerodynamics with a nonlinear structural solver.*
  - *Case C (OpenFAST with AeroDyn/ElastoDyn) combines BEM aerodynamics with a linear structural solver.*

*Because both the aerodynamic and structural solvers vary simultaneously, it is difficult to isolate whether observed differences stem from the flow solver or from the aeroelastic solver (or from their specific implementations). Any causal conclusions should therefore be presented with appropriate caution.*

At the beginning of section 4.2, we added as a preliminary comment the following sentence: "It is important to note that the four solvers employed differ in both their aerodynamic and structural modeling approaches. As a result, it is not always possible to unambiguously determine whether the observed discrepancies in the results originate from the fluid-dynamic models or from the structural formulations."

Moreover, We have rephrased all along the paper and in the Conclusions section, several sentences providing causal conclusions drawn by the comparison between these codes.

- *A related point concerns the comparison with Bernardi et al. (2023). That study focused on a different turbine (NREL 5 MW), under uniform inflow, and reports blade load distributions that do not match those of the present paper. Drawing direct inferences from those results to the IEA 15 MW case with sheared inflow is thus challenging, as multiple differences may drive the discrepancies. In addition, the current manuscript discusses the 5 MW results extensively without showing them, requiring the reader to switch between papers to follow the discussion. To strengthen this part, I would suggest either limiting the cross-paper comparison, or alternatively presenting a direct comparison in the present manuscript — for instance, by reproducing a nondimensionalized plot from Santoni et al. (2017) in a comparable format that allows readers to interpret the differences more clearly.*

[Figure]

Figure 1: Rotor-averaged velocity along the streamwise direction normalized by the undisturbed velocity at the rotor height, namely, $U_\infty = 10\ m/s$, for the present data (black curves) and the work of [1] (red curves). The grey region represents the area covered by the rotor. (RO - - - -, TN ——).

As suggested by the reviewer, we have included in figure 4 of the revised paper, reported here for completeness, a reproduction of the nondimensionalized plot from Santoni et al. (2017) in a comparable format that allows the reader to interpret the differences more clearly. Moreover, we have eliminated some comparison with the 5MW turbine results in the conclusions.

*Convergence study*

- *The grid convergence section currently relies on only two simulations. While this gives a first impression, two points are generally not sufficient to establish a clear convergence trend. I would therefore suggest either adding at least one additional grid resolution to demonstrate a systematic trend, or explicitly referring to convergence studies from previous publications with a closely similar setup.*

Following the suggestion of this referee, we have added a computation with a coarser grid, consisting of $1.3 \times 10^8$ grid points. The outcome of the simulation in terms of angle of attack, forces and momentum is provided in Figure A.2 (page 25). All quantities are found to compare rather well, as discussed in the manuscript.

- *In addition, the statement "the results obtained using the coarse and fine grids are extremely close ... the curves of the angle of attack are almost indistinguishable" comes across as rather qualitative. It would strengthen the section to include a more quantitative measure of convergence (e.g. RMS difference, maximum deviation along the span, or an error norm), so readers can assess the actual magnitude of the differences. Further, the figure of the angle-of-attack comparison in the grid comparison study would be much more clear if it would be scaled to include a maximum angle of 15 degrees. The angles of attack on the cylindrical root section are not of interest to the reader.*

We have included in the discussion a quantitative measure of convergence, in particular the absolute and relative errors on the angle of attack at 80% of the span. Further, following the suggestion of the referee, the figure of the angle-of-attack comparison in the grid comparison study has been scaled to include a maximum angle of 15 degrees.

- *Tip-loss models: I also noticed that different formulations of the tip-loss model are used across the compared solvers. Specifically: OpenFAST applies the classical Prandtl tip and root loss correction (line 222). The CFD-CSD solver employs the [2] correction (line 144). Since the Shen formulation is not equivalent to the standard Prandtl approach, the choice of model may*

*itself contribute to differences in the results. For the sake of a fair comparison, you could report clearly which correction is applied in detail in each case and provide the coefficients used in the Shen model. This would make the comparison more transparent to the reader.*

Indeed OpenFast and the CFD-CSD code use two different tip-loss models, but we have used a cross-code calibration to set the coeffcents of our model to match the forces close to the tip reported by OpenFAST, which uses the Prandtl model. Concerning the tip-loss model used for the CFD-CSD code, we have used the Shen model, whose coefficients $c_1$ and $c_2$ have been set in the following way. In particular, $c_1$ has been set to the value reported in the Shen 2005 paper ($c_1 = 0.125$), which is based on a comparison with experimental distributions of normal forces near the tip for the NREL rotor at a wind speed of $10\,m/s$ , and a tip speed ratio of 3.79. Whereas, the second coefficient has been increased with respect to the value reported by Shen (2005), after a calibration of the coefficient to match the forces close to the tip reported by OpenFAST for the same turbine and flow case, leading to the choice of $c_2 = 32$. A comment has been added to the manuscript reporting these details.

**Detailed Critique Points**

*2 Methodologies*

- *2.1 CFD-CSD Solver*

    - *L. 121: The subsection caption currently reads "2.1 CFD-CSD solver". From what I know, subsection titles should follow a consistent style. Please either use title case throughout (e.g., "2.1 CFD-CSD Solver") or sentence case consistently. At present, there is a mix (for example, "4 Results and Discussion" in title case versus "3 Flow and structural setup" in sentence case). Perhaps also check the WES journal guidelines, whether to use title or sentence case.*

      Looking at the journal guidelines, it is stated that "Titles and headings follow sentence-style capitalization i.e. first word and proper nouns only". Thus, we now used the sentence style, not capitalizing the first letter of nouns. This has been now done throughout the paper, in a consistent way.

    - *L. 123–125: The text states: "The simulations of the flow around the wind turbine are carried out through Large-Eddy Simulations (LESs) of the incompressible, filtered, 3D Navier-Stokes equations, employing our in-house UTD-WF solver (Santoni et al., 2020)." To my understanding, the first introduction of the UTD-WF code was actually in Santoni et al. (2015), "Development of a high fidelity CFD code for wind farm control". The 2020 paper (Santoni et al., 2020) applies the solver at wind farm scale, but the code originates from the earlier work. Since UTD-WF is not as widely established as solvers like OpenFOAM, it would strengthen the methodology section if you also reference publications where the solver has been validated against either an established CFD solver of similar fidelity or experimental data.*

      We have now included more extensive comments concerning the validation of the code, both at the beginning of Section 2.1.1 and in the two validation Appendices (A.1 and A.2).

    - *L. 138: The manuscript currently cites Troldborg (2009) for the Actuator Line Method (ALM). A more appropriate primary reference would be Sørensen and Shen (2002), who first introduced the ALM.*

Thank you for the suggestion, now we have included Sørensen and Shen (2002) as primary reference.

- *L. 144: You mention comparing the Shen tip-loss correction with the Prandtl tip-loss model. Could you please specify the chosen free parameters in the Shen model? It would also help to clarify how you ensured that both tip-loss approaches are consistent, so that observed differences in results are not simply due to discrepancies between the models themselves.*

  As reported above, the coefficient $c_1$ of the Shen model has been set to the value reported in the Shen 2005 paper ($c_1 = 0.125$), which is based on a comparison with experimental distributions of normal forces near the tip for the NREL rotor at a wind speed of $10 m/s$ , and a tip speed ratio of 3.79. Whereas, the second coefficient has been increased with respect to the value reported by Shen (2005), after a calibration of the coefficient to match the forces close to the tip reported by OpenFAST for the same turbine and flow case, leading to the choice of $c_2 = 32$.

- *L. 159–160: The structural model is attributed to Della Posta et al. (2022, 2023). To provide readers with a clearer overview of how the full solver framework was assembled, it would be useful to briefly note the sequence of developments: Santoni et al. (2015): introduction of the ALM–LES flow solver (UTD-WF), Santoni et al. (2017): tower–nacelle representation using the immersed boundary method, Della Posta et al. (2022): coupling of the aeroelastic solver, Della Posta et al. (2023): implementation of the unsteady aerodynamics model (Leishman–Beddoes dynamic stall). You might phrase it along the lines of: "The structural model used in the present study builds upon the UTD-WF framework progressively developed in Santoni et al. (2015, 2017) and further extended by Della Posta et al. (2022, 2023), where the aeroelastic solver and the Leishman–Beddoes dynamic stall model were implemented."*

  This suggestion has been implemented at the beginning of section 2.1.1 of the new version of the paper.

*3 Flow and Structural Setup*

- *L. 235: The subsection caption currently reads "3 Flow and structural setup". For consistency with the other section titles, the first letter of nouns should be capitalized (e.g., "3 Flow and Structural Setup"). Please check the WES journal guidelines and apply the chosen style consistently throughout.*

  This has been now done throughout the paper, in a consistent way.

- *L. 248–249: The text states: "The turbine location is 4 diameter units from the inlet and centered in the spanwise direction." Could you please elaborate on whether this positioning is sufficient to ensure that there is no unintended influence between the turbine and neighboring boundaries? It would be helpful to explain how these dimensions were chosen—either by referencing a small domain-geometry convergence study (perhaps in the appendix) or by citing relevant literature that uses a similar domain setup.*

  In the revised paper, we have discussed the choice of the layout with reference to the data avalilable in the recent literature.

- L. 250: The manuscript notes: "Furthermore, we impose (...) a radiative outlet boundary condition." This boundary condition type is relatively uncommon in CFD literature. A short explanation of its formulation would therefore be valuable for readers. For instance, Santoni et al. (2017) also use this condition and explicitly provide the form of the equation: $\frac{\partial \tilde{U}_i}{\partial t} + C \frac{\partial \tilde{U}_i}{\partial n} = 0$. To my knowledge, this BC is commonly referred to as "convective boundary condition" If you wish to use the term "radiative boundary condition," it would be helpful to clarify this naming choice and cite Santoni et al. (2017) for context.

We have now explicitly reported the form of this boundary condition, and changed the name with "convective boundary condition".

*4 Results and Discussion*

- All captions of the subsections, similar to the caption of chapter three, should have the first letter of the nouns in capital letters. So instead of "4.1 Flow analysis" − > "4.1 Flow Analysis" and the same for 4.2, 4.3, and 4.4

Done (see previous questions).

- *4.1 Flow Analysis*

    – L. 280–283: The manuscript states: "Contrary to what Santoni et al. (2017) observed in their work on the 5MW reference turbine invested by a uniform inflow, the rotor-averaged velocity for the TN configuration in the wake remains slightly lower than for the OR case, indicating that wake recovery is slightly hindered by the presence of the tower." It is useful to compare with Santoni et al. (2017), but in order for readers to interpret the comparison correctly, it would be important to also highlight the notable differences between the setups (e.g., Santoni et al. (2017) used no-slip boundary conditions at all walls — top, bottom, and lateral — and neglected aeroelastic effects). Furthermore, I recommend caution in how the observed results are phrased. Since this finding is based on a single CFD-based aeroelastic solver, it would be safer to state that wake recovery appears to be hindered by tower presence, but that further validation is required as the result does not fully align with expected aerodynamics. In the following sentence, the text currently shifts from observation to causality: "The reason for this behavior can be found in the different aspect ratio of the tower for the present turbine." Here, I would suggest a more cautious phrasing such as: "One possible explanation for this unexpected trend could be differences in the tower-to-rotor aspect ratio."

    We agree with the referee. We have now restated the comparison following the suggestions of the referee.

    – L. 291: The sentence "Figure 5 represents the time-averaged TKE for both configurations on different planes." refers to a figure that is placed two pages later. This is not a critical issue, but if the layout allows, placing the figure closer to its first mention would improve readability.

    Done.

    – L. 296–297: The manuscript states: "This suggests that the tower does not increase the kinetic energy entrainment but it rather has a slight shielding effect on wake recovery." Similar to my earlier comment, I would recommend avoiding the phrasing "this suggests"

*here, as it conveys a level of certainty not yet supported by validation against other solvers or experimental data. A more neutral formulation (e.g., "This result may indicate . . . " or "In this simulation, the tower effect appears to . . . ") would be more appropriate.*

Done.

- *4.2 Aerodynamic loads on the blade*

  - *Figure 7: It may improve clarity if the angle of attack (AoA) is shown in a zoomed-in range (e.g., [0°, 15°]), as otherwise it is difficult to interpret the details. The legend also appears to be missing for this plot, which makes comparison harder. In addition, the BeamDyn AoA and forces in the most outboard two blade sections seem to approach zero, which looks unexpected. It would be helpful to add a short explanation if there is a known reason for this, or otherwise acknowledge it as an open point.*

    Concerning the AoA, this is due to the fact that Prantl tip-loss correction used by OpenFAST is applied to the velocity, and not to the forces, leading to a zero angle of attack at the tip. Concerning the forces, both tip-loss correction models are constructed in order to lead to zero forces at the tip.

  - *L. 339–341: The sentence currently reads: "Indeed, the blade-tower interaction leads to an oscillations of the aerodynamic forces and of the incidence angle around $\theta = 180$, i.e., when the blade is pointing down." There is a grammar error here: it should read "leads to oscillations"*

    Done.

  - *Figure 8b: There appears to be a notable disagreement in F2 between CFD-CSD/OV and ElastoDyn, even though the two models agreed reasonably well on AoA. Moreover, given that CFD-CSD/OV predicts a lower AoA compared to ElastoDyn, one would expect a lower F2 compared to ElastoDyn, yet the opposite is observed. Could you please clarify this?*

    Here, $F_2$ is the flapwise force, the one perpendicular to the plane of rotation of the blades. Since $F_2$ depends on lift times the cosine of the flow angle, it should decrease when the AoA increases with a constant pitch and twist angle. Thus, we think that the trend observed in figure 8b is consistent with that of figure 8a. Concerning the error, since the relation between the AoA and the force is not linear, but it depends on the cosine of the flow angle, we think that the errors on these two variables are not supposed to scale linearly. In particular, on the AoA we observe an error of $\approx 4.5\%$, while on $F_2$ the error is $\approx 12\%$. Since the AoA is rather low in the considered plots, its cosine will be rather high, justifying the increased weight of the error on the flapwise force.

  - *L. 353–355: The manuscript states: "On the other hand, when torsional feedback is included, CFD-CSD/T and BeamDyn solvers agree rather well for all the quantities considered, regardless of the linearity or non-linearity of the models." I would suggest reconsidering this phrasing. In Figure 8a there are visible discrepancies across all azimuthal angles, and in Figure 8c there are large differences at both low and high azimuthal angles. Instead of stating "agree rather well for all quantities," it might be more precise to describe where the agreement is reasonable and where significant differences remain.*

We have modified the paragraph, discussing in detail these differences (see lines 393-395).

– *Figure 9: To improve readability, it would be helpful to use a consistent color scheme across the colormaps, so that interpretation of the results is more straightforward.*

The colormaps in figure 9 have been unified.

– *L. 374–377: The manuscript notes: "Although some mild differences can be observed in their amplitudes and phases, the frequency of these oscillations appears consistent between the two solvers and comparable with the natural frequency of the first torsional mode." Here I would recommend two adjustments:*

   1. *The differences in amplitude, especially for $\alpha$ and F3, are not "mild" but rather notable. In some cases (e.g., BeamDyn-TN), the qualitative shape also differs, showing more oscillations than CFD-CSD/T.*
   2. *Instead of subjective wording like "mild" or "strong," it would be clearer to describe the differences in quantitative or qualitative terms (e.g., "larger amplitude," "different oscillation patterns").*

We have now described the differences in a quantitative way, reporting the approximate errors for all the curves (lines 416-418).

• *4.3 Power and Thrust coefficients*

– *L. 388–390: The manuscript states: "The results reflect the dependency of the power and thrust coefficients on the tangential aerodynamic force F2 and the normal aerodynamic force F3 at the 80 % of the blade, respectively." For clarity and consistency, please use a single naming convention throughout. At present, F2/F3 are sometimes referred to as tangential/normal, and in other places as edgewise/flapwise. Consistency will make it easier for readers to follow.*

We have adopted the naming flapwise/edgewise throughout the revised paper.

– *L. 391–392: The text notes: "Notice that, also here, we can observe that the drop in the Cp curve appears to be rather consistently predicted by BEM and CFD." I would recommend revising this statement. The qualitative shape of the drop differs between the two methods: the BEM prediction exhibits notable oscillations before and after the drop, whereas these are not present in the CFD results. Thus, describing the predictions as "consistently" aligned may be misleading.*

We have now modified this sentence, following the comments of the referee (see lines 433-434).

– *L. 395–399: The manuscript explains: "Indeed, the flow induced by a thinner tower (in diameter units), as in the case of the 15-MW wind turbine, might be better described by a potential flow solution compared to the one induced by a thicker tower, as in the case of the 5-MW wind turbine, and may thus lead to the observed improved agreement between BEM and CFD results." Since you make several direct comparisons with the results of 5MW turbine simulations, these comparisons become a central part of the discussion. To improve readability, I suggest including the relevant 5MW results directly in the figures,*

*rather than expecting readers to switch between different papers.*

To follow this comment of the referee, we have now included a figure concerning the wake recovery data extracted by the paper of Santoni et al 2017 [1] (see reponses above).

- *L. 407–408: The sentence "Overall, it can be said that the performance drop due to the passage in front of the tower is somewhat more limited for the 15MW NREL turbine than for the 5MW counterpart (...)" could be made more precise. The repeated references to the 5MW turbine again suggest that the corresponding results should be explicitly shown and discussed here. Also, the phrases "it can be said" and "somewhat" are too subjective. A more precise wording might be: "Results indicate that the performance drop relative to the corresponding average performance is smaller for the 15MW turbine than for the 5MW counterpart." Ideally, a quantified comparison (e.g., percentage reduction or absolute values) would make this statement more rigorous. Further, the 5MW simulations were performed with uniform inflow, while the 15 MW simulations are performed using a sheared inflow. The lower wind speed in the lower part of the rotor plane means that the bottom half of the rotor plane, where the tower is located, will produce less than half of the total power. This will inevitably cause the performance drop due to the tower to be smaller relative to the total produced power. Therefore the observed difference is not only due to the change in turbine size, but also due to the change in inflow conditions.*

The sentence has been rephrased in the following way: "It can be concluded that the performance loss induced by the passage in front of the tower is less pronounced for the 15 MW NREL turbine ($\approx 5\%$) compared to the 5 MW turbine ($\approx 15\%$), with both BEM theory and CFD yielding similar predictions in the case of the 15 MW turbine." Of course, also the sheared inflow can have a non negligible effect, and this point is now clearly aknowledged at several points in the paper (see, for instance, lines 453-459).

- *L. 410–411: The manuscript states: "Moreover, results seem to suggest that for very large rotors the presence of the tower may constitute a less critical issue for the blade deformations than for smaller rotors (...)". This phrasing is somewhat misleading. It is not possible to draw conclusions about "issues" such as fatigue without dedicated simulations or experiments. What is actually shown are differences in blade deformation predicted by aeroelastic simulations, which have not yet been validated for this case. I would therefore recommend rephrasing along the lines of: "Moreover, the present simulation results predict that for very large rotors the tower effect on blade deformations is less pronounced than for smaller rotors." At the same time, it would be important to acknowledge that this conclusion is based only on one 15MW sheared inflow simulation, where the tower is located in the part of the rotor plane where deflections are generally smaller, and one 5MW uniform inflow simulation, so broader generalizations are not yet possible.*

The sentence has been changed according to the suggestion of the reviewer and refernce to the different inlet condition has been added at lines 453-459.

- 4.4 Structural response

  - *L. 458–459: The manuscript states: "The amplitude of the deformation is however consistent with that obtained by Trigaux et al. (2024) using LES." The wording "consistent" is somewhat ambiguous here. It would strengthen the statement if you could clarify in which sense the amplitudes agree — e.g., whether they are similar in absolute magnitude,*

*in relative deviation from a reference case, or primarily in qualitative trend.*

We have now reported quantitatively the values of the oscillations shown by Trigaux et al. (2024) in their figure 7b (see lines 511-512 of the present paper).

- *L. 474–476: The text reads: "However, the gap between the BEM and the CFD-CSD/T curves is quite large. This can be attributed to the different aerodynamic and structural model used in BEM and LES."*

    1. *The expression "quite large" is subjective; it would be clearer to either quantify the gap (e.g., percentage difference, RMS error) or replace it with a more neutral term such as "substantial" or "noticeable."*
    2. *The phrase "this can be attributed to" presents speculation as a fact. Since both the aerodynamic solvers (BEM vs LES) and structural solvers (modal vs torsional models) differ, the discrepancies cannot be uniquely traced to one factor. A more balanced formulation would be: "These differences likely arise from the combined effects of both aerodynamic and structural modeling approaches."*

    We have now rephrased these sentences as suggested by the referee, also providing a quantitative estimate of the error (see lines 517-518).

- *L. 491–493: "the second and third flapwise natural frequencies are indeed recovered by all the numerical models". These peaks are visible, because they are at the 13th and 26th multiple of the rotational frequency (13p and 26p). They are not visible because they are at the second and third flapwise mode, because flapwise modes have very large aerodynamic damping, which is also why the first flapwise mode is not visible.*

    We agree with the referee, although we haven't noticed that before. We have now changed the text according to this referee's comment (lines 543-546).

*5 Conclusions*

- *L. 522–527: The manuscript states: "The entrainment of kinetic energy driven by the tower leads to higher turbulence levels in the near wake, but then result into a slightly decreased mixing behind the turbine, differently to what has been found for the NREL 5MW wind turbine, whose wake recovery was found to be promoted by the presence of the tower. This finding can have important implications for the aerodynamic loads on downstream turbines in wind farms and overall farm efficiency." This is a counter-intuitive result, as prior studies (e.g., the NREL 5MW turbine case) suggested the opposite effect. Since the observation is based on a single solver without confirmation from experiments or other CFD-based aeroelastic simulations, I recommend formulating this more cautiously. For example, instead of "this finding can have important implications", you could write that "this result requires further examination, as it appears counter-intuitive and has not yet been confirmed by other studies." This way, the uncertainty is acknowledged while still highlighting the potential significance.*

    Thank you for the suggestion, we have restated this sentence in the manuscript (lines 579-580).

- *L. 551–553: The manuscript concludes that "CFD-CSD can capture complex aerodynamic loading and turbulent effects better than BEM." While this is likely true in general, the specific evidence presented here — larger amplitudes at lower frequencies — does not in itself*

*constitute proof of that statement. It would strengthen the conclusion to either rephrase more cautiously (e.g., "In this case, the CFD-CSD solver captured larger low-frequency fluctuations than BEM") or to provide additional evidence that directly supports the broader claim.*

We have rephrased the sentence in a more cautiously way (see line 604-605).

- *L. 562–563: The differences between CFD-CSD/T and BeamDyn are described in a way that attributes them to solver fidelity. However, since the two approaches differ in both aerodynamic fidelity (LES vs. BEM) and structural fidelity (torsional vs. modal models), it is not possible to unambiguously assign the discrepancies to one component. I recommend rephrasing along the lines of: "The observed differences likely stem from the combined effects of differences in aerodynamic and structural fidelity, and cannot be uniquely attributed to one component alone."*

We have rephrased the sentence as suggested by the referee (see lines 617-619).

*Appendix A Grid Convergence Study for the LES Simulation*

- *L. 573: To demonstrate a clear trend in grid convergence, more than two grid resolutions are typically required. With only coarse and fine grids, it is difficult to establish whether the solution is truly converging. One possible approach would be to use the finest grid as a reference and compute a root-mean-square (RMS) error relative to it. Plotting the RMS across multiple grid resolutions would then allow you to illustrate the convergence trend more quantitatively.*

Done (see answers above).

- *L. 579: The manuscript states: "The comparison in figure A1 shows that the results obtained using the coarse and fine grids are extremely close each other along the entire blade span." This phrasing is somewhat vague and subjective. To make the convergence assessment clearer, it would be better to quantify the agreement, for example by reporting RMS errors or percentage deviations. Also, there is a small grammar correction: it should read "close to each other" rather than "close each other."*

The updated figure has now been discussed in much more detail. Also, the typo has been corrected.

*Appendix B. Validation of the Structural model*

- *Caption: Please ensure consistent use of title case or sentence case across the entire manuscript. For example, the current caption "Appendix B. Validation of the Structural model" mixes styles. The WES journal guidelines specify which convention should be followed — it would be good to align with that.*

Looking at the journal guidelines (Titles and headings follow sentence-style capitalization i.e. first word and proper nouns only), we now used the sentence style, not capitalizing the first letter of nouns. This has been now done throughout the paper, in a consistent way.

- *L. 599–600: The manuscript states: "The computed values of the modal frequencies appear to be consistent with the other results, although some discrepancies in the higher-order modes*

*are observed." Since the 6th mode (corresponding to the first torsional mode) is likely the most relevant for the present study, it would strengthen the paper to explicitly address these discrepancies. Please clarify in which sense the results are "consistent" and provide more detail on how the 6th mode compares across solvers.*

As now clearly reported in the paper, the structural model for the IEA 15MW wind turbine has been cross-validated with many other aeroelastic numerical codes within the framework of the International Energy Agency (IEA) Wind TCP Task 47 TURBINIA [3]. In this IEA Task, a consortium of research institutions and industrial partners benchmarked their own aeroelastic codes on the IEA 15 MW wind turbine [4]. Since we cannot report in this paper data from all these partners, we have provided here a comparison of literature data. Concerning the latter, we have now clarified the discrepancies and added more details about this comparison.

- *Figure B2:*

  - *From my understanding, the 6th mode (first torsional mode) should be central to your analysis. I would expect clustering of results for solvers without torsion degrees of freedom (e.g., purple, green) as distinct from those including torsion (e.g., red, blue, grey, black). However, this clustering is not clearly visible. Is it possible that there has been a mix-up between the 6th and 7th modes in Table B1? For instance, in the 7th mode the aeroelastic solvers without torsion do not appear, whereas they do for the 6th mode. Even if this is not the case, a short discussion of why this expected separation is not observed would improve clarity.*

    We thank the reviewer for the comment as it highlighted a mistake in the order of the modes from the literature.

    Indeed, we confirm that our modal analysis provides a 6th mode with torsional nature. However, as noticed by the reviewer, some of the mentioned solvers do not include torsional degrees of freedom in their structural models. Hence, as the natural frequencies of the torsional modes are missing, there cannot be a perfect correspondence between the order of the modes of the various models. In particular, the frequencies erroneously indicated as 6th mode for ElastoDyn and H2-PTNT can only correspond to flexural modes.

    According to these observations, we have corrected the figure and we have moved the natural frequencies of the former 6th mode of ElastoDyn and H2-PTNT to our 7th mode, which is instead an edgewise mode that can be captured by these models and has a similar frequency as well.

  - *There is also a large spread in the higher-order modes. It could be informative to explicitly separate the results into torsion and no-torsion degrees of freedom. For example, ElastoDyn does not include torsional degrees of freedom, so one would expect systematic differences relative to models that do. The same applies to H2-PTNT.*

    We thank the reviewer for the suggestions, and we trust that the previous answer clarified the separation between torsional and non-torsional modes.

    Spread between the higher-order natural frequencies in the literature, instead, may reasonably stem from the differences in the models used. As higher frequencies are considered, we observe that coupling between the torsional, flapwise, and edgewise degrees of freedom arise, which may be potentially affected by the different modeling assumptions adopted in the definition of the mass and stiffness matrices, or even by differences in the material properties used. However, we believe that our results are within the tolerance observed in the literature and thus guarantee a proper validation of the structural model used in this work.

*Sources*

- *The reference "Hansen, M. (2015). Aerodynamics of wind turbines. Routledge." should be adapted to match the citation style required by WES. At the moment, the publisher and format do not appear consistent with standard referencing. Please check the journal guidelines to ensure proper formatting. For example, depending on the required style, the same reference would look as follows:*

We have used the bst file provided by copernicus for the bibliography section. We carefully checked the biblio file and the reference above is correctly inserted. We think that the publisher may fix some formatting problems on the bibliography, in case they would be an issue.

- *APA Style: Hansen, M. O. L. (2015). Aerodynamics of wind turbines (3rd ed.). Earthscan.*
- *Harvard Style: Hansen, Martin Otto Laver, 2015. Aerodynamics of wind turbines. 3rd edn, Earthscan, London, UK.*
- *Chicago Style: Hansen, Martin O. L. 2015. Aerodynamics of wind turbines. 3rd ed. London, UK: Earthscan.*
- *Vancouver Style: Hansen MOL. Aerodynamics of wind turbines. 3rd ed. London, UK: Earthscan; 2015.*

**The revised manuscript with the highlighted changes mentioned in this point-to-point document is attached in the following pages.**

[revised manuscript text omitted]
 introduced by Santoni et al. (2015). The UTD-WF framework has been progressively developed by Santoni et al. (2017, 2020) and further extended by Della Posta et al. (2022, 2023), where the aeroelastic solver and the Leishman–Beddoes dynamic stall model were implemented. The solver has been validated in its non-aeroelastic version by Santoni et al. (2017) against wind-tunnel data reproducing the NTNU "Blind Test" and comparing simulations to Krogstad et al. (2015) measurements, also considering the impact of tower and nacelle. Whereas, the recently developed version of the code including the two-way FSI coupling Della Posta et al. (2023) has been validated through comparison against reference datasets, including HAWC2-based results reported by Heinz (2013). The IEA 15MW wind turbine configuration cosidered here has been cross-validated with many other aeroelastic numerical codes in the International Energy Agency (IEA) Wind TCP Task 47 (Cacciola et al., 2025), also considering turbulent inflow conditions (Schepers et al., 2025). Notice that prior validations by Della Posta et al. (2022) of the CFD-CSD solver were made on a laminar uniform and a turbulent sheared inflows for a 5 MW NREL turbine, whereas our study extends the validated setting to the IEA-15 MW case for a sheared laminar inflow configuration. However, as discussed in the framework of the IEA Wind TCP Task 47 Schepers et al. (2025), turbulent fluctuations appear to have a much stronger impact than shear on load response of aero-elastic numerical codes. Moreover, high-fidelity codes appear rather consistent in predicting loads, while engineering models tend to overpredict fatigue loads, particularly for large rotors (Cacciola et al., 2025).

The code implements a second-order accurate centered finite difference scheme for the spatial discretization on a staggered Cartesian grid. A hybrid low-storage third-order-accurate Runge–Kutta (RK) scheme is used for time integration of the non-linear terms (Orlandi, 2012), while the linear terms are treated implicitly using a Crank-Nicolson scheme. The filtered governing equations are:

$$\frac{\partial u_i}{\partial t} + \frac{\partial u_i u_j}{\partial x_j} = -\frac{\partial p}{\partial x_i} + \frac{1}{Re}\frac{\partial^2 u_i}{\partial x_j \partial x_j} - \frac{\partial \tau_{ij}}{\partial x_j} + \tilde{f}_i, \quad (1)$$

$$\frac{\partial u_i}{\partial x_i} = 0, \quad (2)$$

where $i, j \in \{1, 2, 3\}$ represent, in a Cartesian reference frame, the components along the streamwise (x), wall-normal (y), and spanwise (z) directions, respectively. The Reynolds number $Re = U_\infty D/\nu$ is defined by the undisturbed inlet velocity $U_\infty$, the turbine diameter $D$, and the kinematic viscosity of the fluid $\nu$. These quantities are used as reference values to make the equations non-dimensional. To solve the filtered equations, a Subgrid-Scale (SGS) stress model is needed. The latter describes the interaction between the large resolved and the sub-grid unresolved scales, as described by Pino Martín et al. (2000) and Santoni et al. (2017). Here, we employ the Smagorinsky model with constant $C_s = 0.09$ as discussed by Martinez-Tossas et al. (2018).

The effect of the blades on the flow is modeled by the Actuator Line Model (ALM) (Sorensen and Shen, 2002), by adding a forcing term to the Navier-Stokes equations, representing the force per unit volume exerted by the rotor on the fluid. By approximating the rotor blades as rigid straight lines discretized into segments, it is possible to estimate the lift and drag forces per unit length on a 2D plane as follows:

$$F_l = \frac{1}{2}\rho u_{rel}^2 C_l(\alpha)cF, \qquad F_d = \frac{1}{2}\rho u_{rel}^2 C_d(\alpha)cF, \quad (3)$$

where $\rho$ is the air density, $c$ is the local chord, $u_{rel}$ is the relative incoming velocity, $\alpha$ is the angle of attack, and $F$ represents the tip loss correction factor, which employs the tip-loss model proposed by Shen et al. (2005). The coefficients $c_1$ and $c_2$ of this model have been set in the following way: $c_1$ has been set to the value reported in the Shen et al. (2005) paper ($c_1 = 0.125$), whereas, $c_2$ has been chosen after a calibration with respect to the forces close to the tip reported by OpenFAST for the same turbine and flow case, leading to the choice of $c_2 = 32$. The forces are then projected on the flow employing a 2D Gaussian kernel, which spreads the lift and drag force vector, $\boldsymbol{f}^{aero}$, in cylinders surrounding the actuator line,

$$\tilde{\boldsymbol{f}} = -\boldsymbol{f}^{aero}\frac{1}{\epsilon^2\pi}exp\left[-\left(\frac{r_\eta}{\epsilon}\right)^2\right], \quad (4)$$

where $r_\eta$ is the radial distance of a generic point of the cylinder from the actuator line and $\epsilon$ is the spreading parameter, where $\epsilon/\Delta \geq 2$, with $\Delta = \sqrt{\Delta x^2 + \Delta y^2 + \Delta z^2}$, following Troldborg (2009). The tower and nacelle are modeled using the Immersed Boundary Method (IBM) following the approach described by Orlandi and Leonardi (2006).

**2.1.2 Structural solver**

From an aerodynamic standpoint, the rotor blades represent the most flexible components within a wind turbine. Several studies demonstrated that their modal properties have a significant impact on

[Figure]

Figure 1: Sketch of the frames of reference used for the CFD and for the CSD simulations.

the dynamics of the entire structure (Damgaard et al., 2013; Dong et al., 2018). Moreover, an analysis of the isolated blades is also sufficient to accurately estimate the aeroelastic properties of the entire structure, including the flutter speed (Abdel Hafeez and El-Badawy, 2018). Additionally, the tower and shaft exhibit minimal deflection due to their stiffness. In light of the above considerations, the aeroelastic model is constructed to encompass solely the structure of the blades.

The structural model used in the present study was previously presented by Della Posta et al. (2022, 2023). In order to model the working conditions, the blades are assumed to be rotating beams rigidly clamped at the hub (cantilever beams), under the assumption of small deformations with respect to a relative frame of reference (FOR). The direction of the pitching axis is denoted by $X_1$. This coincides with the neutral axis of the blade, defined as passing through the quarter of the chord. The direction of the out-of-plane flapwise motion is indicated by $X_2$ and is oriented in the positive streamwise direction. The in-plane edgewise direction of $X_3$ is defined such that the FOR is oriented as a right-handed coordinate system (Figure 1).

Under the assumption of linearity, the elastic generalised displacement $\boldsymbol{d}$, which includes translational $d_i$ and rotational $\theta_i$ degrees of freedom (DoFs), is decomposed along the coordinate $X_1$ on the neutral axis as:

$$\boldsymbol{d}\left(X_1,t\right) = \sum_{m=1}^{M} q_m(t)\,\boldsymbol{\psi}^m\left(X_1\right), \tag{5}$$

where $\psi^m\left(X_1\right)$ is the m-th elastic mode shape from the modal analysis of the structure, $q_m$ is the corresponding modal coordinate, and $M$ is the number of modes used. The general inertial coupling is included in a modal basis by means of the methodology introduced by Reschke (2005) and further

developed for the case of wind energy by Della Posta et al. (2022). In particular, the two-way coupling algorithm between rigid-body and structural dynamics does not take into account a modification of the rotor inertia caused by the deformation of the blades. Hence, the structural dynamics of the structure can be described by the following equation:

$$\boldsymbol{M}\ddot{\boldsymbol{q}} + [\boldsymbol{D} + \boldsymbol{D}^{Co}(\boldsymbol{\Omega})]\dot{\boldsymbol{q}} + [\boldsymbol{K} + \boldsymbol{K}^{c}(\boldsymbol{\Omega}) + \boldsymbol{K}^{Eu}(\dot{\boldsymbol{\Omega}})]\boldsymbol{q} = \boldsymbol{e} + \boldsymbol{e}^{c}(\boldsymbol{\Omega}) + \boldsymbol{e}^{Eu}(\dot{\boldsymbol{\Omega}}), \qquad (6)$$

where $\boldsymbol{M}$, $\boldsymbol{D}$ and $\boldsymbol{K}$ denote the modal structural mass, damping, and stiffness matrices, respectively, and $\boldsymbol{e}$ are the external loads expressed in modal basis. The remaining terms are inherently related to the various contributions to the acceleration in a moving FOR. Terms with the superscript $Co$, $c$ and $Eu$ are related to the Coriolis, centrifugal, and Euler accelerations, respectively. The discrete evaluation of the additional inertial terms in Equation (6) is expressed as a function only of the information known from the structural finite-element method (FEM) model and from the corresponding mode shapes, according to Saltari et al. (2017). For the modal analysis, we use a finite element model of the blade based on complete beam elements with 6 DoFs, with Euler-Bernoulli behavior for bending in directions $X_2$ and $X_3$, and linear shape functions for axial and torsional deformations. The generalized-$\alpha$ method (Chung and Hulbert, 1993) is employed to advance the structural dynamic equation in time, which is unconditionally stable for linear problems, and second-order accurate. We assume a lumped-mass representation, and we take into account the local offset of the centers of mass with respect to $X_1$. Finally, the structural matrices are assembled considering the local twist. Details about the modal analysis are provided in Appendix A.

**2.1.3 Fluid-structure interaction model**

The two-way coupling aeroelastic model employs the ALM sectional approach, whereby the angle of attack (AoA) and relative velocity are locally modified following the instantaneous blade motion provided by the structural dynamics. In particular, the distribution of the AoA along each blade is evaluated as a function of the velocity of the fluid, the angular velocity of the rotor, and the instantaneous elastic state of the blade. The latter is generally constructed from the deformation velocity $\boldsymbol{u}_{def} = \dot{\boldsymbol{d}}$ and the local vector of the deformation angles $\boldsymbol{\theta}$ (torsion and bendings) derived from the structural solver, which is forced by the updated aerodynamic loads. The algorithm restricts inter-field communications solely at the beginning of each RK substep, thereby ensuring optimal computational efficiency. The impact of torsional dynamics was deemed to be limited in light of the results obtained in previous studies on the effect of torsion for smaller wind turbines (Chen, 2017). In order to investigate this issue for the large rotor 15MW wind turbine, in this study we compare two different CSD models. In particular, we consider as a baseline a two-way coupling that includes the effect of blade deformation velocity as a sole variable (CFD-CSD/OV, for Only Velocity), and a more complete model including the torsional deformation in the coupling (CFD-CSD/T, for Torsional). In general, the relative velocity for a rotating blade can be defined with the following expression:

$$\boldsymbol{u}_{rel} = \boldsymbol{u}_{abs} - \boldsymbol{\Omega} \times \boldsymbol{r}_{OP} - \boldsymbol{u}_{def}, \qquad (7)$$

where $\boldsymbol{u}_{abs}$ is the filtered velocity from the fluid solver at the actuator line, $\boldsymbol{r}_{OP}$ is the general radial vector pointing to the considered section, $\Omega$ is the rotor rotational speed, and $\boldsymbol{u}_{def}$ is the deformation

velocity of the structure at the same position. As a result, the AoA used to determine the air load coefficients is defined as follows:

$$\alpha = \text{atan}\left(\frac{\boldsymbol{u}_{rel} \cdot \boldsymbol{E}_2}{-\boldsymbol{u}_{rel} \cdot \boldsymbol{E}_3}\right) - \phi - \theta_{tors} = \text{atan}\left[\frac{(\boldsymbol{u}_{abs} - \boldsymbol{u}_{def}) \cdot \boldsymbol{E}_2}{\Omega r - (\boldsymbol{u}_{abs} - \boldsymbol{u}_{def}) \cdot \boldsymbol{E}_3}\right] - \phi - \theta_{tors}, \qquad (8)$$

where $\phi$ is the local twist angle of the blade, $\theta_{tors}$ is the local torsional deformation, $\boldsymbol{E}_i$ are the unit vectors of the relative FOR rotating with the structure, and hence, $v_2 = \boldsymbol{u}_{def} \cdot \boldsymbol{E}_2$ is the flapwise deformation velocity component, and $v_3 = \boldsymbol{u}_{def} \cdot \boldsymbol{E}_3$ is the edgewise deformation velocity component. The simplified coupling procedure benefits from the sectional one-dimensional formulation of the ALM, which avoids the complex treatment of the fluid-solid interface with the associated kinematic and traction conditions.

**2.2 OpenFAST modules**

For comparison purposes, wind turbine simulations have been also conducted using the OpenFAST solver *Release v3.2.0* (July 29, 2022). OpenFAST is a widely utilized open-source numerical code developed by the NREL that combines different specialized modules for simulating the coupled aero-hydro-servo-elastic response of wind turbines. The aerodynamic computations are performed by the *AeroDyn* (Jonkman et al., 2015) module which is based on the BEM theory. A Prandtl loss model is applied to account for the tip and root effects. The structural module dedicated to the computation of the blade deformation is contained in the *BeamDyn* module, which relies on the geometrically exact beam theory and may resolve geometric non-linearities and large deflections (Wang et al., 2016b). *BeamDyn* has replaced the simplified *ElastoDyn* module, based on a modal approach and suitable for blade deformation dominated by bending. In order to compare the CFD-CSD results with a modal structural analysis, we also performed simulations using the standalone *ElastoDyn* module. It is worth to notice that the latter does not take into account the torsional degree of freedom, so it is to be directly compared to the CFD-CSD/OV model, which also does not account for the coupling between the torsional deformation and the angle of attack. As reported in the original manual of *AeroDyn* (Moriarty and Hansen, 2005), OpenFAST couples the fluid and structural solvers in a similar way to our CFD-CSD solvers. In particular, the local angle of attack is determined taking into account the local deformation velocities.

**3 Flow and structural setup**

In this work, we consider a stand-alone IEA 15-MW wind turbine (Gaertner et al., 2020b) in its monopile configuration. This wind turbine has a rotor diameter $D = 240$ m with three blades of length $L = 117$ m. Table 1 provides the main features of the turbine.

The computational box has dimensions $12.5 \times 5 \times 3$ diameter units, as shown in Figure 2. The distance of the turbine from the inlet of the computational domain (equal to 4D) has been determined on the base of the reference data available in the literature, which vary in the range 2D-5D. Smaller distances from the inlet (2D) have been employed for experimental set-up (Bartl and Satran, 2017; Krogstad et al., 2015), whereas larger distances (in the range 2.7D-5D) are typical of numerical simulations

(Porte-Agel and Wu, 2011; Ciri et al., 2017; Allah and Sha ei Mayam, 2017; Stevens et al., 2018). Moreover, we have verified numerically that pressure fluctuations do not generate spurious reflections at the inlet section in our simulations. The spanwise length of the computational domain (equal to 3D) is the same employed in previous numerical simulations (Ciri et al., 2017; Allah and Sha ei Mayam, 2017). We have verified that, using periodic boundary conditions, the blockage effect on the single turbine is negligible. Moreover, following the convergence study reported in the Appendix A, the computational box has been discretized by a staggered grid composed of $2049 \times 513 \times 513$ points in the streamwise, wall-normal, and spanwise directions, respectively. The orthogonal grid is equally spaced in the streamwise and spanwise directions and is stretched vertically, with a gradually wider spacing starting from the region above the rotor. The grid spacing described leads to an actuator line discretized by 86 points per blade. The time resolution of the LES computation is tied to the spatial resolution, as defined by the stability requirements of the numerical scheme adopted. Simulations are carried out at a constant Courant–Friedrichs–Lewy (CFL) number (Courant et al., 1967) $CFL = 0.65$, which ensures an average time step $\overline{\Delta t} = 0.024s$. The turbine location is 4 diameter units from the inlet and centered in the spanwise direction. Furthermore, we impose a sheared laminar inflow velocity profile, defined by a power law with the exponent $\alpha = 0.05$, and a convective outlet boundary condition, i.e., $\frac{\partial u_i}{\partial t} + C \frac{\partial u_i}{\partial x} = 0$, with the constant $C$ set to the average value of the outflow velocity. In the spanwise direction, periodic boundary conditions are imposed. Moreover, slip and no-slip conditions are enforced at the top and bottom boundaries, respectively. The turbine is subjected to a flow with a Reynolds number $Re \approx 10^8$ and operates at its nominal tip speed ratio (TSR) of $\lambda = 9$. The streamwise undisturbed velocity at the hub height is constant and equal to $U_\infty = 10 \ m/s$. The simulations were conducted for a time interval of 300 $s$ over the initial transient, which corresponds to 35 revolutions of the rotor.

To identify the optimal configuration for the structural model, we conducted a preliminary sensitivity analysis and then validated the structural eigenfrequencies with the results found in the literature. A more detailed insight into this analysis is presented in Appendix B, where the structural properties of this turbine are shown. Finally, a number of modes $M_s = 15$ and a structural discretization of the blades given by $N = 80$ equally-spaced nodes were chosen.

**4   Results and discussion**

This section presents the results of two set of simulations: one modeling a rotor-only configuration (RO) and the other including the tower and nacelle (TN). Furthermore, both configurations are subjected to comparative analysis using the OpenFAST submodules. Firstly, the near-wake aerodynamic characteristics and the wake recovery of both configurations determined by the CFD-CSD solvers are discussed. Then, the aerodynamic loads on the blades are analyzed and the outcomes from both solvers are compared. Finally, the overall turbine performance and the effects on the blade deformation are assessed.

| Parameter | Units | Value |
|---|---|---|
| Power rating | $MW$ | 15 |
| Rotor diameter ($D$) | $m$ | 240 |
| Rotor orientation | $-$ | Upwind |
| Number of blades | $-$ | 3 |
| Blade length ($L$) | $m$ | 117 |
| Hub height | $m$ | 150 |
| Hub radius ($R_{hub}$) | $m$ | 3.97 |
| Rated wind speed | $m/s$ | 10.59 |
| Design tip speed ratio | $-$ | 9 |
| Maximum rotor speed | $RPM$ | 7.56 |

Table 1: IEA 15-MW (Gaertner et al., 2020b) wind turbine main features

[Figure]

Figure 2: Sketch of the computational box where the incoming sheared flow and the position of the turbine are highlighted.

**4.1 Flow analysis**

As a first step, we analyze the flow field variables, as obtained using the CFD-CSD/T solver. Figure 3 illustrates the main coherent flow structures in the field by means of an instantaneous isosurface of the Q-criterion colored by the streamwise velocity for both cases. It is evident that the presence of the tower affects the vorticity intensity distribution along the vertical direction. In particular, the occurrence of a low-velocity recirculation zone at the tower height for the TN case can be identified, which is a result of the tower shadowing (see Figure 3b). Moreover, the TN case demonstrates a more rapid dissolution of the endogenous coherent hub vortex structures if compared to the RO case (see Figure 3a). On the other hand, the tip vortex structures appear to be minimally influenced by the presence of the tower. Figure 4 shows the rotor-averaged streamwise velocity along the flow direction, time-averaged over 30 revolutions of the rotor. Contrary to what Santoni et al. (2017) observed in

[Figure]

Figure 3: Q-criterion contour of the instantaneous velocity field colored by the streamwise velocity for the rotor-only case (RO) (a) and tower and nacelle (TN) (b).

their work on the 5MW reference turbine invested by a uniform inflow (see the red lines in figure 4), the rotor-averaged velocity for the TN configuration in the wake remains slightly lower than for the OR case, indicating that wake recovery is slightly hindered by the presence of the tower. Although further validation is required as the result does not fully align with this previous literature study, wake recovery appears thus to be hindered by tower presence. One possible explanation for this behaviour could be differences in the tower-to-rotor aspect ratio. In particular, for the NREL 5-MW turbine, the ratio between the tower diameter and the rotor diameter is about equal to 0.047, whereas, for the 15MW turbine, it is only about 0.027 (the tower diameters being $6m$ and $6.5m$, respectively). Thus, the thinner shape (in terms of diameter units) of the tower, as well as the lower value of the incoming velocity at the tower height due to the presence of shear at the inflow, result into a decreased mixing behind the turbine which leads to a slower wake recovery.

From an energy perspective, the wake recovery process can be depicted by examining the Turbulent Kinetic Energy (TKE) in the wake. Figure 5 represents the time-averaged TKE for both configurations on different planes. The TN case exhibits high TKE values in the near wake, in the region just downstream of the tower and nacelle. The top view of the TN case shows that the TKE in the wake presents an asymmetric distribution as De Cillis et al. (2022) observed, among the others, in their work. On the contrary, the RO configuration shows large TKE only in the far wake region, with large values also in the region above hub height. This result may indicate that the tower does not increase the kinetic energy entrainment but it rather has a slight shielding effect on wake recovery. Although not favoring kinetic energy entrainment, the tower still plays a strong role in the wake dynamics, as it can be visualized in figure 6, showing slices of instantaneous streamwise velocity at different tower heights corresponding to 80% of the blade (top) and to the tip of the blade (bottom), when the blade is in front of the tower, i.e. $\theta = 180°$ (left), and when it is far from it (right). In particular, it can be observed that the turbulent mixing right downstream of the tower is already very high in the near wake compared to that close to the tip of the blades. Probably due to asymmetry induced by the

[Figure]

Figure 4: Rotor-averaged velocity along the streamwise direction normalized by the undisturbed velocity at the rotor height, namely, $U_\infty = 10 \; m/s$, for the present data (black curves) and the work of Santoni et al. (2017) (red curves). The grey region represents the area covered by the rotor. (RO ----, TN ——).

[Figure]

Figure 5: Top (upper slices) and lateral (lower slices) views of the time-averaged Turbulent Kinetic Energy on slices passing through the hub. TN (left), RO (right).

rotation of the blades, inside the rotor disk, it can be seen that the tower wake bends in the spanwise direction (Figure 6, top frames), whereas it is rather spanwise independent at a height corresponding to the blade's tip (bottom frames). Moreover, one can see that the passage of the blade in front of the tower (left frames) induces a strong perturbation in the flow field already upstream of the tower. In the following section, the effect of this perturbation on the phase oscillations of several relevant quantities (aerodynamic forces, power coefficient, etc.) will be discussed.

**4.2   Aerodynamic loads on the blade**

The analysis of the aerodynamic loads on the blade has been conducted using the present CFD-CSD models and the engineering software OpenFAST. The same laminar sheared inflow is imposed for

[Figure]

Figure 6: Instantaneous streamwise velocity on horizontal slices at different tower heights corresponding to 80% of the blade (top slices), and the tip of the blade (bottom slices). In the left configuration, the blade is in front of the tower ($\theta = 180°$), while on the right the blade is far from the tower.

both solvers using a power law with the same exponent and reference streamwise velocity at the hub height. We have chosen not to impose a turbulent inflow to avoid differences in the definition of the turbulent inflow itself which might have hindered the comparison between the results of the two codes. It is important to note that the four solvers employed differ in both their aerodynamic and structural modeling approaches. As a result, it is not always possible to unambiguously determine whether the observed discrepancies in the results originate from the fluid-dynamic models or from the structural formulations. Figure 7 depicts the following time-averaged aerodynamic quantities along the span of the blade: the local angle of attack $\alpha$ (Figure 7a); the aerodynamic pitching moment per unit length $M_{aero}$ (Figure 7b); the flapwise and edgewise components of the aerodynamic force per unit length $F_2$ (Figure 7c) and $F_3$ (Figure 7d), respectively. In particular, Figure 7a shows that a good agreement of the local incidence angle computed by both CFD-CSD models (solid lines) with that computed by *ElastoDyn* (circles) and *BeamDyn* (squares) is obtained from the 20% up to the 80% of the blade length. Indeed, the differences in the root area could ascribable to the presence of the hub which is modeled differently by the solvers. The discrepancy of the incidence angle observed towards the tip subsequently affects the aerodynamic loads. The $F_2$ force in Figure 7c shows a very good fit of the CFD-CSD/T results with that of the nonlinear solver *BeamDyn*, despite the linearity of our in-house CSD model. The strong discrepancies with respect to the values obtained by *ElastoDyn* can be ascribed to the absence of the torsional deformation in the latter solver. Indeed, the CFD-CDS/OV solver, which neglects the torsional feedback in the coupling, shows very similar results to the *ElastoDyn* solver. A similar effect can be observed by examining the reduction in $F_3$ towards the tip of the blade (see Figure 7d). The distribution of the aerodynamic pitching moment presents instead a maximum gap of about 8% from the BEM-based solvers.

[Figure]

Figure 7: Average aerodynamic quantities along the blade compared between CFD-CSD/OV (black solid line), CFD-CSD/T (blue solid line), *ElastoDyn* (red rounded markers), *BeamDyn* (magenta squared markers). (a) Incidence angle, (b) Aerodynamic pitch moment, (c) flapwise aerodynamic force, (d) edgewise aerodynamic force.

As demonstrated by Hansen (2015), the outer third of the blade span is the most critical region in terms of deflections and deformations due to the combination of higher aerodynamic loads and reduced structural stiffness. Therefore, a phase average of the aerodynamic quantities at the 80% of the blade has been performed. Figure 8 reports the evolution of the incidence angle and of the aerodynamic force components at $\frac{r-R_{hub}}{L} = 0.8$ (being $R_{hub}$ the hub radius and $L$ the blade length) versus the blade rotation angle $\theta$. The dynamical behavior of the aerodynamic quantities in the presence (solid lines) or in the absence (dashed lines) of the tower underlines that the passage of the blade in front of the tower represents the main source of instability for the flow conditions considered. Indeed, the blade-tower interaction leads to oscillations of the aerodynamic forces and of the incidence angle around $\theta = 180°$, i.e., when the blade is pointing down. However, unlike the case of the NREL 5-MW turbine (Bernardi et al., 2023), this effect appears to be stronger for the BEM computations than for the CFD-CSD solver. Concerning this point, we should recall that, as pointed out by Bernardi et al. (2023), the complex flow dynamics resulting from the interaction between the blade and the tower, shown in Figure 6, may not be well described by OpenFAST, which uses a simple potential flow model. It can be observed that, between the rotor and the tower, a region with low streamwise velocity is observed. We can expect that the passage of the blade in front of the tower thus induces an alteration of the aerodynamic forces on the blade due to the decrease/increase of the streamwise velocity. This issue

[Figure]

Figure 8: Phase-averaged values of: (a) the local incidence angle, (b) flapwise aerodynamic force, and (c) edgewise aerodynamic force at the 80% of the blade. CFD-CSD/OV: TN ——, RO - - - -. CFD-CSD/T: TN ——, RO - - - -. *ElastoDyn*: TN ——, RO - - - -. *BeamDyn*: TN ——, RO - - - -.

will be further discussed in the following, where a possible reason for the different behavior observed for the IEA 15-MW with respect to the NREL 5-MW turbine will be discussed.

Apart from the effect of the tower, one can observe a rather good match between the CFD-CSD/OV and *ElastoDyn* solvers for both the incidence angle and the edgewise component of the aerodynamic force, while the flapwise component presents some discrepancies. On the other hand, when torsional feedback is included, CFD-CSD/T and *BeamDyn* solvers, regardless of the linearity or non-linearity of the models, agree rather well on the aerodynamic forces, especially on the flapwise one, which shows an error $\approx 2\%$, while the edgewise force reaches a $\approx 5\%$ error at azimuthal angles close to $\theta = 0$. Whereas, the error between the two solvers on the angle of attack reaches 8%.

To better investigate the local response of the different models during the blade revolution, we conducted a comparative analysis of the aerodynamic loads, employing phase-averaged quantities over the span. Figure 9 illustrates the percentage difference of the phase-averaged aerodynamic quantities on the rotor plane of the *ElastoDyn* (*BeamDyn*) solver with respect to the CFD-CSD/OV (CFD-CSD/T) model, respectively. In particular, in comparison to *ElastoDyn*, a higher value of the absolute incidence angle in the range of $|\langle \Delta\alpha/\alpha^{CFD-CSD/OV}\rangle^{\%}| = [17\%, 25\%]$ is found in the zone after the tower (see Figure 9a). The difference with respect to the results obtained by *BeamDyn* tends to be higher moving from the root to the tip with a discontinuity in the tower area, spanning the range $|\langle \Delta\alpha/\alpha^{CFD}\rangle^{\%}| = [35\%, 60\%]$ in the last 20% of the blade span. Furthermore, the angle of attack distribution affects the components of the aerodynamic force. In fact, the distribution of the flapwise component of the force follows the same pattern of the incidence angle (see Figure 9b). On the other hand, for the edgewise component the major discrepancies are concentrated in the final radial sections of the blade toward the tip (see Figure 9c). In general, we can conclude that the most significant discrepancies are observed in the tip region where the three-dimensional effects are more relevant and where the complexity of the fluid flow is strongly affected by the presence of the tower.

Notably, similar discrepancies are observed when comparing the CFD-CSD/T solver with the *Beam-Dyn* solvers. However, in this case some high-frequency oscillations are observed for the three aerodynamic quantities. In fact, the same oscillations are observed in the phase averaged quantities at 80%

[Figure]

Figure 9: Phase-averaged contour plots of the percentual differences of the aerodynamic quantities between CFD-CSD/OV versus *ElastoDyn* (left), and CFD-CSD/T versus *BeamDyn* (right), respectively. (a) Incidence angle, (b) flapwise aerodynamic force, (c) edgewise aerodynamic force.

of the blade shown in Figure 9, for both the CFD-CSD/T solver and *BeamDyn*. The frequency of these oscillations computed by the two solvers appear very close and comparable with the natural frequency of the first torsional mode, although some differences can be observed in the amplitudes of the signals, especially concerning the angle of attack ($\approx 8\%$ of error) and the edgewise aerodynamic force at azimuthal angles close to zero ($\approx 6\%$ of error). Again, this observation indicates that including the torsional degree of freedom in the structural solver is crucial for describing accurately the amplitude and dynamical behaviour of the aerodynamic quantities.

**4.3 Power and thrust coefficients**

The aerodynamic loads previously presented are also useful to evaluate the power and thrust coefficients, defined as follows:

$$C_p = \frac{P_d}{\frac{1}{2}\rho A U_\infty^3}, \quad C_t = \frac{T_{aero}}{\frac{1}{2}\rho A U_\infty^2}, \tag{9}$$

where $A = \pi D^2/4$ represents the rotor area, $P_d$ is the aerodynamic power transferred to the rotor and $T_{aero}$ is the overall aerodynamic thrust on the turbine.

[Figure]

Figure 10: Phase-averaged power (a) and thrust (b) coefficients. CFD-CSD/OV: TN ——, RO ‑ ‑ ‑ ‑.
CFD-CSD/T: TN ——, RO ‑ ‑ ‑ ‑. *ElastoDyn*: TN ——, RO ‑ ‑ ‑ ‑. *BeamDyn*: TN ——, RO ‑ ‑ ‑ ‑.

Starting from the time history of $C_p$ and $C_t$, we computed their phase-averaged evolution as reported in Figure 10. The periodic passage of the blades in front of the tower for the TN configuration produces a drop of the curves of about 10%. Eventually, the performance is restored to the value obtained in the RO case following the elastic dynamical behavior of the structure. The results reflect the dependency of the power and thrust coefficients on the edgewise aerodynamic force $F_3$ and the flapwise aerodynamic force $F_2$ at the 80% of the blade, respectively (see Figures 8c and 8b), which are strongly influenced by the presence of the tower. Notice that, also here, we can observe that the drop in the $C_p$ curve appears to be rather similarly predicted by BEM and CFD, although the BEM prediction exhibits notable oscillations before and after the drop, whereas these are not present in the CFD results. A different behaviour was observed for the NREL 5-MW turbine (Bernardi et al., 2023), where this performance drop is considerably underestimated by the BEM computations. A possible factor that may contribute to this different behaviour may reside in the different relative geometry of the two wind turbines. Indeed, the flow induced by a thinner tower (in diameter units), as in the case of the 15-MW wind turbine, might be better described by a potential flow solution compared to the one induced by a thicker tower, as in the case of the 5-MW wind turbine, and may thus lead to the observed improved agreement between BEM and CFD results. Moreover, the differences in the flow impinging on the blade might also have an effect. In fact, in Bernardi et al. (2023) a uniform inflow was imposed. Whereas, in the present case, due to the shear imposed at the inflow and the limited distance from the ground of the tip of the blade (only $\approx 0.125D$ for the 15MW turbine), the blade is invested by a flow having a much smaller velocity compared to the given value of $U_\infty$ at hub height, further confirming the increased suitability of a potential flow solution upstream of the tower. Nevertheless, we should recall that this remains a very strong approximation, as also demonstrated by the differences in the forces and angles that have been observed in the previous section (see Figure 9, for instance).

It can be concluded that the performance loss induced by the passage in front of the tower is less pronounced for the 15 MW NREL turbine in the present configuration ($\approx 5\%$) compared to the 5

[Figure]

Figure 11: Power Spectral Density (PSD) of the power (a) and thrust (b) coefficients. The vertical dashed lines highlight the rotational frequency of the rotor $P = f_{rot}$ and the multiples of $3P$, respectively. CFD-CSD/OV ——, CFD-CSD/T ——, *ElastoDyn* ——, *BeamDyn* ——.

MW turbine in the configuration considered in Bernardi et al. (2023)($\approx 15\%$), with both BEM theory and CFD yielding similar predictions in the case of the 15 MW turbine. However, it is worth recalling again that Bernardi et al. (2023) considered a uniform inflow, whereas here the inflow is sheared. This can be a possible reason for this different behaviour, since the lower wind speed in the lower part of the rotor plane leads to a lower production in the bottom half of the rotor plane, where the tower is located. This may cause a smaller performance drop due to the tower relative to the total produced power. Therefore, the observed difference can be not only due to the change in turbine size, but also due to the change in inflow conditions.

Moreover, the present results predict that, for very large rotors and a sheared inflow, the tower effect on blade deformations is less pronounced than for smaller rotors, although it should yet be taken into account for accurately describing the turbine's performance oscillations as it still represents a major source of unsteadiness.

The average value of the power coefficient is much larger when the torsional deformation is neglected. This feature is observed by both CFD and BEM approaches. However, one can observe that *ElastoDyn* underestimates the value of $C_p$ with respect to the corresponding non-torsional CFD model, while the opposite is observed when comparing *BeamDyn* with the torsional CFD solver. This is probably due to the fact that *BeamDyn* predicts higher values of the aerodynamic edgewise forces with respect to the CFD-CSD/T approach, which are linked to a smaller torsional deformation as will be shown in figure 12f in the next section.

Figure 11 shows the premultiplied Power Spectral Density (PSD) of the power (Figure 11a) and thrust (Figure 11b) coefficients evolution. The PSD is normalized by the variance of each coefficient $\sigma^2$ and plotted versus the frequency normalized by the rotational frequency of the rotor, $f/f_{rot}$. In both cases, the CFD-CSD solvers seem to provide a richer representation of the aerodynamic coefficients, capturing the full range of flow-structure interactions. Indeed, an examination of the low-frequency behavior reveals that both quantities exhibit isolated low-frequency peaks when using the BEM-based solvers, a phenomenon not observed with the CFD-CSD, where the low-frequency range is rather broadband and does not present particular peaks. It is important to notice that the frequency 1P can be directly linked to the frequency of the passage of the blade in front of the tower, but also to wind shear loads on the blades. Concerning the first point, a potential flow solution as that used in the BEM solver is keen to provide a simple, single-frequency response, whereas a complex, turbulent flow is expected to result in a more broadband spectrum. Concerning the second point, we have to consider that in LES, the power law profile is imposed at the inlet of the domain but it is free to evolve for 4 diameters before the wind turbine, altering in a non-trivial way the flow field and the consequent frequency response of the blades. This outcome indicates that the BEM-based solvers tend to overcut the power oscillations associated with low-frequencies that are not exactly equal to 1P or 2P. For all solvers, however, the strongest PSD peaks are to be found at much larger frequencies (3P-6P-9P-12P), as also observed by Pagamonci et al. (2023) by means of URANS aeroelastic simulations of the NREL 5-MW, the DTU 10-MW, and the IEA 15-MW turbines. One can also notice that the amplitude associated with the 3P frequency appears to be consistently described by the two solvers, although also in this range the BEM solver appears to overdamp the frequencies in between different peaks. Moreover, a good agreement is evident between the two set of results concerning the value of the frequencies and the level of the PSD for frequencies that are multiples of $3P$.

**4.4   Structural response**

This section presents the analysis of the structural dynamics. Figure 12 reports the phase-averaged dynamic response of the free extremity of the blade (left column) and the time-averaged deformation of the entire span (right column). Figure 12a shows how the out-of-plane deformation is mainly governed by the aerodynamic component of the force normal to the rotor plane and, hence, to the aerodynamic effects, heavily affected by the tower. In fact, it is visible how the tower placed at $\theta = 180°$ produces a drop in the deformation, followed by an elastic dynamic response which restores the value far from the pointing-down position. The time-averaged maximum deformation predicted by the CFD-CSD/OV solver is 16% higher compared to the *ElastoDyn* module and 17% compared to *BeamDyn* (see Figure 12b). On the other hand, the same quantity predicted by the CFD-CSD/T solver is 17% lower compared to the *ElastoDyn* module and 13% compared to *BeamDyn* (see Figure 12b). This is consistent with the fact that including the torsional degree of freedom reduces the loads (see figure 8b) and the resulting deformation. Although the trend of deformation with respect to the blade span appears similar to previous predictions based on URANS (see Pagamonci et al. (2023)), the out-of-plane deformation is rather larger, reaching $16\,m$ at the blade's tip. The amplitude of the deformation is however close to that obtained by Trigaux et al. (2024) using LES. Figure 12c depicts instead the in-plane deformation, which is mostly due to gravity. The results show that the shadowing effect of the tower does not influence this quantity. Furthermore, the discrepancies obtained

[Figure]

Figure 12: Phase-averaged deflections at the tip of the blade (left column) and time-averaged deflections along the blade span (right column). CFD-CSD/OV: TN ——, RO ----. CFD-CSD/T: TN ——, RO ----. *ElastoDyn*: TN ——, RO ----. *BeamDyn*: TN ——, RO ----.

between *ElastoDyn* and *BeamDyn* can be attributed to the lack of modes used by the former model to describe the translation in the edgewise direction (see Figure 12d). The discrepancy does not seem to be linked to the linearity of this model, as the result of the CFD-CSD/T solver, which is linear

as well, is much closer to the *BeamDyn* results. Moreover, the results of the CFD-CSD/OV and the CFD-CSD/T models are very close each other. It can be noticed that the amplitude of the oscillation of the in-plane deflection is consistent with that reported by Trigaux et al. (2024) (see Figure 7b of their paper, reporting an oscillation between $\approx -2.3$ and $\approx 0.2$ ), although the sign is opposite due to the different frame of reference used.

A further significant insight into the deformation phenomenon is provided by the torsional DoF. Figure 12e shows a comparison of the torsional angle at the tip with *BeamDyn*. Significant discrepancies can be observed between the LES and the BEM approaches, which cannot be reconducted to the different coupling procedures adopted by the models. On the one hand, *BeamDyn* and CFD-CSD/T both take into account the deformation angle in the coupling (Wang et al., 2016b), while in the CFD-CSD/OV solver the angle of attack depends only on the deformation velocity (see Equation 8). However, the gap between the BEM and the CFD-CSD/T curves is quite large, reaching approximatively 20% of the torsional deformation value. These differences likely arise from the combined effects of both aerodynamic and structural modeling approaches used in BEM and LES. This is confirmed by the time-averaged torsional deformation along the span reported in Figure 12f where the maximum percentual gap of *BeamDyn* reaches 29% for the CSD-CFD/OV, and 24% for the CFD-CSD/T. It is noteworthy that the lower torsional deformation resulting from *BeamDyn* leads to the higher aerodynamic loads observed in figure 8c.

Finally, figure 13 illustrates the Power Spectral Density (PSD) of the blade's tip deformation components for the TN configuration (which is characterized by more complex fluid-structure interactions). The premultiplied PSD values are normalized by the variance of the signal, $\sigma^2$, and plotted versus the frequency normalized by the rotor frequency, $f/f_{rot}$. Spectral results have been corroborated through use of the Welch and Lomb-Scargle PSD estimation algorithms.

Figure 13a shows the out-of-plane deformation, which we showed to be influenced mostly by the aerodynamic loading. The results indicate that, for all the numerical approaches used, the observed structural response does not exhibit a peak corresponding to the first flapwise natural frequency, suggesting that the intrinsic dynamics of the structure might play a less prominent role in the deformation process. A similar behavior is found in the results of Trigaux et al. (2024) (see figure 6 of the cited paper) for the same turbine and similar inflow conditions. Noticeably, all the numerical models recovered peaks at frequencies close to the (highly damped) second and third flapwise natural frequencies, but they appear to rather correspond to the $13^{th}$ and $26^{th}$ multiple of the rotational frequency (i.e., 13p and 26p). Both CFD-CSD solvers predict larger amplitude responses across a broad frequency range compared to OpenFAST, indicating a higher capability to capture complex flow interactions, including turbulence-induced vibrations. This effect is particularly pronounced at the lower frequencies, probably due to the large-scale three-dimensional structure of the flow impinging on the turbine, which is not captured by OpenFAST. These aspects seem to be under-represented in the *ElastoDyn* and *BeamDyn* solutions. Although the *ElastoDyn* curve aligns with both the CFD-CSD solvers at some key frequency peaks, it does not fully account for the fine-scale flow-structure interactions. On the other hand, the *BeamDyn* curve provides better agreement with the CFD-CSD solvers, especially at higher frequencies near the blade's natural modes, suggesting that *BeamDyn* captures more of the structural dynamics, particularly the aeroelastic response. Figure 13b shows the in-plane deformation, which is primarily influenced by gravity, centrifugal, and Coriolis forces acting on the blade. The

[Figure]

Figure 13: Power Spectral Density (PSD) of the out-of-plane (a), in-plane (b), and torsional (c) deformations of the blade. The vertical dashed lines represent the first 8 eigenfrequencies of the system. CFD-CSD/OV ——, CFD-CSD/T —— *ElastoDyn* ——, *BeamDyn* ——.

CFD-CSD solvers again demonstrate stronger low-frequency components.

Figure 13c presents the torsional deformation for the CFD-CSD/T and *BeamDyn* solvers, excluding *ElastoDyn*, which neglects the torsional DoF in the model. Additionally, also this quantity demonstrates that the CFD-CSD curves predict higher amplitudes at low frequencies. However, a good agreement between the two solvers is evident at higher frequencies, especially in the range around the first torsional eigenfrequency.

**5    Conclusions**

This study investigated the aeroelastic response of the IEA 15-MW wind turbine by employing a high-fidelity Computational Fluid Dynamics (CFD) solver that couples Large-Eddy Simulation (LES) with a Computational Structural Dynamics (CSD) solver. Two different CSD solvers are considered: the CFD-CSD/OV solver, in which the only structural quantity contributing to the definition of the angle of attack is the deformation velocity, and the CFD-CSD/T solver, in which the instantaneous torsional deformation is also considered when defining the local effective incidence. The results of the two CFD-CSD solvers are compared with those of traditional engineering solvers such as *BeamDyn* and *ElastoDyn*, both relying on Blade Element Momentum (BEM) theory. Two case studies were examined: a rotor-only configuration (RO) and one that included the tower and nacelle (TN).

In the first instance, a flow analysis uncovered important considerations regarding wake entrainment. In particular, the study found that, for the considered turbine, impinged by a laminar sheared inflow, wake recovery is only slightly hindered by the presence of the tower. The entrainment of kinetic energy driven by the tower leads to higher turbulence levels in the near wake, but then result into a slightly decreased mixing behind the turbine, differently to what has been found for the NREL 5MW wind turbine, whose wake recovery was found to be promoted by the presence of the tower. This result requires further examination, as it appears counter-intuitive and has not yet been confirmed by other studies.

In addition, the Power Spectral Density (PSD) of the power and thrust coefficients revealed that the CFD-CSD solver captures a broader range of flow-structure interactions, with a more broadband low-frequency response, compared to the BEM-based solvers. The isolated low-frequency peaks found in *BeamDyn* and *ElastoDyn* suggest that these solvers tend to over-simplify the aerodynamic fluctuations associated with phenomena such as wind shear and tower shadowing. For the large IEA 15-MW turbine, the performance drop caused by tower passage is not very pronounced and the resulting oscillations predicted by the BEM approach appear to be larger than the CFD-CSD solver.

Concerning the forces on the blade and the incidence angle, one can observe a rather good match between the CFD-CSD/OV solver and *ElastoDyn*, as well as between the CFD-CSD/T model and the *BeamDyn* solver. This is likely due to the presence – or not – of the torsional feedback, while non-linearities of the structural solver appear to have only a limited impact on the observed quantities. In agreement with previous studies, the results thus suggest that including the torsional degree of freedom in the structural solver is crucial for accurately describing the amplitude and dynamical behaviour of the aerodynamic quantities.

Moreover, it is observed that duly taking into account the torsional degree of freedom reduces the value of $C_p$. This feature is consistently observed by both CFD and BEM approaches. However, one can observe that *BeamDyn* predicts lower values of the torsional deformation and thus higher values of the aerodynamic edgewise forces with respect to the CFD-CSD/T approach, leading to a larger $C_p$ value than that predicted by LES. All in all, it can be concluded that for the considered setup, the CFD-CSD solvers tend to exhibit larger amplitudes at lower frequencies with respect to BEM ones.

The structural response of the wind turbine blade has been assessed by comparing the out-of-plane, in-plane, and torsional deformations obtained from the CFD-CSD solvers, *ElastoDyn*-based, and *Beam-Dyn*-based OpenFAST solver. In-plane deformation, influenced significantly by centrifugal forces,

appears to be better captured by the CFD-CSD solvers, especially in the low-frequency range. Concerning the out-of plane deflection, large discrepancies are seen between the two CFD-CSD solvers, as well as between both BEM modules and the LES.

Our results underscore the importance of incorporating torsional deformation effects in the definition of the angle of attack and using high-fidelity aeroelastic models to ensure accurate predictions of wind turbine blade performance with a richer fluid dynamics. Whereas, the linearity of the structural model does not appear to have a strong effect on the aerodynamical quantities, deformations and loads. In general, the comparison of the results of the CFD-CSD solver with those of the engineering solver shows differences especially in the region behind the tower. The observed differences likely stem from the combined effects of differences in aerodynamic and structural fidelity, and cannot be uniquely attributed to one component alone.

Future work will explore the effect of turbulent fluctuations at the inlet to better investigate the impact of the atmospheric boundary layer on the aerodynamic forces, loads and deformations of the present turbine.

**A Appendix A. Grid convergence study for LES**

A grid convergence study was conducted to evaluate the sensitivity of the LES results to spatial and temporal resolution. Two further simulations were carried out using grids of different densities: a coarser mesh and a finer mesh, having approximately 40% less and more grid points than the former in each spatial direction, respectively. This allowed for a more detailed resolution of flow structures and aerodynamic quantities. Moreover, both simulations use the same $CFL = 0.65$ as the present grid. The average time step obtained and the other key parameters of different LES runs are summarized in Table A1.

| Parameter | Coarse Grid | Present Grid | Fine Grid |
|---|---|---|---|
| Total number of cells | $1.31 \times 10^8$ | $5.37 \times 10^8$ | $1.36 \times 10^9$ |
| Largest cell diagonal (m) | 8.1 | 5.0 | 3.5 |
| Smallest cell diagonal (m) | 3.9 | 2.5 | 1.7 |
| Actuator points per blade | 54 | 86 | 128 |
| Average time step (s) | 0.043 | 0.024 | 0.012 |
| Total number of threads | 320 | 512 | 768 |

Table A1: Comparison of the main parameters for different meshes.

The comparison in figure A2 shows that the results obtained using the coarse and fine grids are extremely close to each other along the entire blade span. In particular, the curves of the angle of attack are almost indistinguishable for the coarser and the reference grid, even in the outer portion of the blade, where stronger differences were expected due to tip effects and local three-dimensionality. Slightly larger differences are recovered between the reference and the finer grid, but only at low radius. In particular, for these two grids the maximum deviation of the incidence angle $\alpha$ between the two simulations at 80% of the span reaches a value of $\Delta\alpha_{max} \approx 0.2°$, corresponding to a relative

difference of 1.6%. Whereas, the maximum deviation between the reference and the coarser grids at 80% of the blade span is $\Delta\alpha_{max} \approx 0.1°$, corresponding to a relative difference of 1.5%. Similarly, the aerodynamic forces component distributions exhibit negligible variation between the reference and finer resolutions, and less than 1% relative variations between the reference and the coarser grids, confirming the overall consistency of the LES solution examined in the Sec. 4 with respect to mesh refinement.

These results indicate that the coarse grid already accurately captures the main aerodynamic features, making the use of a finer mesh unjustified given its higher computational cost and minimal accuracy gain.

[Figure]

Figure A1

Figure A2: Average aerodynamic quantities along the blade obtained from the coarse grid (red line), the present grid (black line) and the finer grid (green line). (top left) Incidence angle, (top right) Aerodynamic pitch moment, (bottom left) flapwise aerodynamic force, (bottom right) edgewise aerodynamic force.

**B   Appendix B. Validation of the structural model**

The structural model for the IEA 15MW wind turbine has been cross-validated with many other aeroelastic numerical codes within the framework of the International Energy Agency (IEA) Wind TCP Task 47 TURBINIA (Schepers et al., 2025). In this IEA Task, a consortium of research institutions and industrial partners benchmarked their own aeroelastic codes on the IEA 15 MW wind turbine (Cacciola et al., 2025). Since we cannot report in this paper data from all these partners, we provide here a preliminary study was conducted to validate the structural model prior to coupling it with the CFD solver. Figure B1 shows the distributions of the structural and constructive properties along the

blade, which were utilized as input for the modal CSD analysis. A convergence study to determine the proper number of elements, $N_e$, (not reported here for brevity) was conducted, leading to the choice $N_e = 80$. Furthermore, the results of the present structural analysis were compared with those of five models including: the prismatic Timoshenko model without torsion (H2-PTNT); the Timoshenko model with a fully populated stiffness matrix (H2-FPM) from the study of Rinker et al. (2020); the 3D Finite Element Analysis (3D FEA) selected from Zhang et al. (2023); the ElastoDyn model; the BeamDyn model. Figure B2 shows the first 8 eigenfrequencies using the present method compared with the results of these models. The computed values of the modal frequencies appear to be consistent with the other results, although some discrepancies in the higher-order modes are observed. Moreover, an analysis of the most important modes was conducted: Table B1 provides the classification of the first 8 modes, whereas, Figures B3, B4, and B5 show the modal displacements for the first spanwise, edgewise, and torsional modes, respectively.

[Figure]

Figure B1: Structural properties of the blade along the span: (a) stiffness, (b) inertia, (c) density, (d) local twist angle.

[Figure]

Figure B2: A comparison of the eigenfrequencies computed by different structural models.

| # | $f_n[Hz]$ | Mode |
|---|---|---|
| 1 | 0.5369 | 1st flapwise |
| 2 | 0.7267 | 1st edgewise |
| 3 | 1.577 | 2nd flapwise |
| 4 | 2.267 | 2nd edgewise |
| 5 | 3.113 | 3rd flapwise |
| 6 | 3.642 | 1st torsional |
| 7 | 4.571 | 3rd edgewise |
| 8 | 5.385 | 4th flapwise |

Table B1: Classification of the first 8 structural modes.

[Figure]

Figure B3: Mode 1 shape for all the DoFs.

[Figure]

Figure B4: Mode 2 shape for all the DoFs.

[Figure]

Figure B5: Mode 6 shape for all the DoFs.

*Author contribution.* CB: Investigation, Writing - Original draft, Formal analysis, Methodology, Software, Validation. SC: Conceptualization, Investigation, Writing - Review & Editing, Supervision. FM: Methodology, Software, Validation. GDP: Formal analysis, Writing - Review & Editing, Methodology, software. SL: Conceptualization, Software, Supervision. PDP: Conceptualization, Investigation, Writing - Review & Editing, Supervision.

*Competing interests.* The authors declare that they have no competing interests.

*Acknowledgments.* This study has been partially funded under the National Recovery and Resilience Plan (NRRP), Mission 4 Component 2 Investment 1.3 - Call for tender No. 1561 of 11.10.2022 Project code PE0000021, Project title "Network 4 Energy Sustainable Transition – NEST", and under the PRIN grant 20229YJP33, "Diffuser augmented Wind Turbines for URBban environments" (DWTURB). Both are grants of Ministero dell'Università e della Ricerca (MUR), funded by the European Union – NextGenerationEU. The cooperative work on the 15MW NREL wind turbine within the IEA WIND TASK 47 - TURBINIA is also acknowledged.

---

## Author Comment (AC2)

**Point-to-point reply to reviewers**

Paper wes-2025-120

Title:

**Large Eddy Simulation of the IEA 15-MW Wind Turbine Using a Two-Way Coupled Fluid-Structure Interaction Model**

December 12, 2025

The authors wish to thank the reviewers for their comments and suggestions on the paper to enhance the manuscript. The paper has been revised accordingly. The revised parts have been highlighted in blue (for this referee) and red (for referee 1) in the updated manuscript. If mentioned, the page number refers to the revised paper. In the following we report point-to-point replies concerning the questions and comments raised by the reviewers.

**Reviewer #2**

**Main Critique Points**

1. *The term "aeroelasticity" is well known, I suggest removing this definition.*

   We have eliminated this sentence.

2. *Strictly speaking, fatigue itself is not an aeroelastic phenomenon, but aeroelastic effects can cause or accelerate fatigue in wind turbines.*

   We have specified that in the text. The referee can find the revised version of the manuscript attached and follow the text in blue.

3. *I suggest to slightly rephrase this sentence, softening it.I agree with the authors that BEM aerodynamics has significant limitations. Anyhow, it is currently the state-of-the-art solvers for practical industrial applications and certification. Moreover, beside highlighting its limitations, several works have demonstrated that, if properly tuned, it can be a valuable engineering-type solver, complementary to higher fidelity ones which have also higher computational costs. See, for instance, the results of IEA WIND Task 47..*

   We have restated the text in this paragraph, and cited the references reported by the referee.

4. *I don't agree with this point of view. It is true that panel methods (and in general potential flow solvers and/or free-vortex wake methods) and ALM solvers are computationally more expensive than BEM. Nevertheless, they are less expensive that blade-resolved CFD. Furthermore, many papers show that panel codes and free-wake vortex methods are able to capture unsteady blade/rotor aerodynamics with good accuracy in different operating conditions (including off-design) whenever massive flow separation phenomena do not occur. See, for instance:..*

   We have restated the text in this paragraph, and cited the references reported by the referee.

5. *From this sentence it appears that CFD is described as the best compromise between accuracy and computational costs. Differently, within the research topic addressed in this work, CFD should be seen as the reference high-fidelity approach that can be used for a (limited) number of computations in order to refine and/or assess the lower fidelity models.*

   We agree with the referee and we have restated the text in this part.

6. *This sentence is not clear. The pargraph describes tools such as FAST which is not based on a lifting-line aerodynamic formulation.*

   Sorry it was meant to be "BEM" instead of lifting line. We have corrected that in the text.

7. *At this point is not clear if, within the aeroelastic computation, the ALM elements follow blade deformation or not.*

   We have now specified that the ALM elements do not follow the blade deformation.

8. *Please provide more details about the procedure used to compute the flow velocity contribution to urel. Which approach was used (Line Averaging Technique, for instance, or others?).*

   The definition of $u_{rel}$ is given in eq. (7). The interpolation from the structural FEM nodes for $u_{def}$ to the Cartesian CFD mesh on which $u_{abs}$ is defined, is made with a simple linear interpolation. Then, the resulting $u_{rel}$ is interpolated on the nodes discretizing the blade.

9. *Although the authors refer to previous works for the structural model, it is preferable to briefly outline it here. Which type of beam model is used (Euler-Bernoulli, Tiomoshenko, others? Is it a linear model or a nonlinear one? Are the Coriolis forces and the spin-softening effect taken into account?*

   We have now specified that it is a linear model taking into account Coriolis, centrifugal and Euler effects. Concerning the type of beam model (Euler-Bernouilli), it is specified in the text under eq. 6.

10. *Please indicate if the modal analysis is performed on the rotating or nonrotating blade. In the latter case, is it performed past the deformed or undeformed configuration?*

    We have now specified this at several points in the paper.

11. *The authors state that they are using a modal approach to solve the aeroelastic equations. How is the Lumped Mass representation here mentioned linked to the modal approach? As a general comment, in my opinion the description of the structural model is not clear and lacks suitable clarifications*

    Following the approach of Reschke (2005) and Saltari et al. (2017), the inertial coupling terms arising from the noninertial frame of reference are described using a reduced set of coefficients, which are estimated via finite element method discretization. Details can be found for the specific case of wind turbine's blades in Della Posta et al. (2022). The lumped mass representation has thus been used to define the mass matrix of the blade, to evaluate the inertial coupling terms according to the mentioned method, and also as an input of the eigenproblem priviting the natural frequencies and modes used in the work.

    Also, we have considerably changed and extended the description of the structural model.

12. *Usually in modal approaches, the quantity d includes, for each blade section, flap (w), lag (v) and torsion (theta) DOFs. Consistently, also w' and v' will appear in the aeroelastic equations, so they will be computed as part of the aeroelastic solution. It is not clear to me*

*why the torsion angle is here accounted for in addition to d.*

Flap and lag are indeed considered in the structural dynamics solver, and their time derivatives are considered in the definition of the relative velocity in equation 7 as $\boldsymbol{u_{def}}$, which is defined on the basis of the translational contributions of $\dot{\boldsymbol{d}}$ only, as clarified now in the text. As explained after equation 8, $v_2 = u_{def} \cdot E_2$ is the flapwise deformation velocity component, and $v_3 = u_{def} \cdot E_3$ is the edgewise deformation velocity component, indicating the translational structural dynamics. The torsion is also considered as part of the elastic state relevant for the aeroelastic coupling, in order to take into account angular deformations that change the actual orientation of the local airfoil on the deformed blade with respect to the impinging flow.

13. *What do you mean by "bendings"?*

Bendings here indicate rotations around $X_2$ and $X_3$, which describe the edgewise and flapwise (in-/out-of-plane) angular deformations respectively.

14. *It is important to provide an indication of the computational costs of a typical solution.*

The increase in computational cost induced by the structural solver is negligible, as it accounts for less that 2%.

15. *Please describe how this is computed, as it is one of the critical aspects of ALM methods.*

Please, see answer to comment number 8 of the present referee.

16. *I am confused by the appearance of $_{tors}$ here. If I understood correctly, the modal dofs already include both bending (flap and lag) and torsion. Please clarify.*

As now mentioned in the revised methodology, once the modal dynamics is advanced in time, the displacement is projected back onto the physical base to recover its expression in the rotating frame of reference. We have also explicitly explained in the revised manuscript that $\boldsymbol{u_{def}}$ only considers the time derivative of the translational degrees of freedom, so that the presence of $_{tors}$ should be clearer in eq.8.

17. *OpenFAST is a widely used tool, so, in my opinion, there is no need to devote a specific section to its description. I suggest that simply specifying the main characteristics of the aerodynamic and structural formulation (sepcifically indicating if blade torsion has been included or not - i.e. if BeamDyn or Elastodyn has been used) is sufficient.*

We have now eliminated this subsection and included only a small paragraph in the next section.

18. *I presume the shear inflow is imposed at the inlet, is it correct? If so, using a no-slip condition at the bottom will modify the actual inflow at the rotor disk. Please clarify this aspect.*

We have now explained in the text that indeed the flow changes from the inflow to the turbine, but the provided profile comply with the no-slip condition.

19. *Rotating or nonrotating? Past the deformed or undeformed configuration (if rotating)?*

We have now specified that it is the rotating undeformed configuration.

20. *Is the pitching moment computed past the c/4 point of each blade section?*

We confirm that.

21. *I suggest changing the scale of the contour plot because it is difficult to appreciate the different TKE levels. If this is difficult considering the predicted values of TKE, it may help showing directly the TKE difference between the TN and OR case. Moreover, the figures are really small.*

    We have enlarged and reorganized the figures. The slices are now presented in a single column to enhance their readability.

22. *I don't understand the values y/D indicated in the plot. If I see correctly from Fig.5 the tower base is at y/D=0, so why would these horizontal planes be at y/D¡0? Moreover, top panels of Fig 6 show a deviation of the hub wake from the turbine rotation axis. Something similar happens when the rotor is in yaw, but here it is not the case. It would be interesting to provide a justification for this phenomenon.*

    In this figure y=0 was erroneously set at the hub of the turbine, we have changed that for consistency, introducing the value $h_{hub}$. The deviation of the hub wake from the turbine rotation axis is not due to the yaw but to the mutual effect of the asymmetry induced by the rotation of the blades and the wake meandering. Concerning the second effect, the wake is subject to low frequency oscillations, and in this snapshots it happens to be oscillating towards higher values of z, which combines positively to the slight effect towards higher values of z of the direction of rotation of the blades.

23. *Considering the number of lines in the figures it is quite difficult to recall everytime which result correspond to which line/marker. I suggest adding a graphical legend to the plots.*

    Done.

24. *As a general comment to Figs 8 and 10, I suggest showing only the fluctuation of the quantities of interest, while reporting the mean values in suitable tables. As the paper is dealing with aeroelasticity, unsteady fluctuations are more interesting than the mean values. Differently, using the current plots, they are not easily appreciable.*

    We have tried to plot only the fluctuations in figures 8, 10 and 12, removing the mean values, but the plot becomes very confusing since many of the linea are superposed. For completeness, we report the plot in figure 1 of the present document. This considered, we have decided not to remove the mean from the figures 8, 10, 12.

[Figure]

Figure 1

25. *Here flapwise and tangential force are mentioned. Before they were F1, F2. Please define a unique nomenclature for these quantities.*

[Figure]

Figure 2: Rotor-averaged velocity along the streamwise direction normalized by the undisturbed velocity at the rotor height, namely, $U_\infty = 10$ *m=s*, for the present data (black curves) and the work of Santoni (2017) (red curves). The grey region represents the area covered by the rotor. (RO ‑ ‑ ‑ ‑, TN ——).

We have now clearly defined the flapwise and edgewise forces at the end of page 12 : "flapwise and edgewise components (normal and tangential to the rotor disk),respectively, of the aerodynamic force per unit length $F_2$ and $F_3$". In the reminder of the paper, we now use $F_2$ and $F_3$ within the figures, and "flapwise and edgewise " in the text.

26. *This metrics is intuitively clear, but needs a formal definition.*
    Done.

27. *Why are the left and right contour maps different? In addition, it would help the figure clarity if a sketch of the rotor/tower were included (maybe using transparency).*

    The colormaps have been unified. Also, a sketch of the tower is now included in each subfigure.

28. *The parallel between the 5 MW and the 15 MW behaviour is very interesting and involve several aspects. Anyhow, for the sake of readibility, the results related to the 5 MW which are needed to support the comments here addressed must be here reproduced. Otherwise it is quite difficult to follow the comments on the basis of the textual description of plots not shown here.*

    Concerning the comparison with the 5MW turbine, as suggested by the reviewer, we have included in figure 4 of the revised paper, reported here for completeness as figure 2 of the present response, a reproduction of the nondimensionalized plot from Santoni et al. (2017) in a comparable format that allows the reader to interpret the differences more clearly. Also, we have included the results of Bernardi et al. (2023) in figure 10 of the paper. Moreover, to comply with another referee's comment, we have eliminated some further comparison with the 5MW turbine results.

29. *Which comparative 5 MW-15 MW result support this conclusion? An explicit comparison of the mentioned results is needed here.*

    We have now included the curve of the figure 3 of Bernardi et al. 2023 in which it can be clearly seen that the presence of the tower induces larger oscillations in the power coefficient. However, since the inflow was different in this literature case, we have restated the sentence as: "Moreover, the present results predict that, for very large rotors and a sheared inflow, the tower effect on blade deformations is less pronounced than for smaller rotors".

30. *Indicating the value of frot would help the readibility.*

    We have now reported this value.

31. *Although well known, a formal definition of 1P, 2P and so on is needed.*

    Done.

32. *Considering the combination of frequencies experienced by the rotor, the expected result in terms of thrust and power (both axial loads) is that only the multiple of blade passage frequency are retained in the PSD (so 3P, 6P and so on). This is actually confirmed by all proposed solvers here, except Beamdyn, which shows a significant peak at 1P. The authors should comments and justify this unespected finding.*

    We appreciate the reviewer's observation regarding the presence of a peak at 1P in the spectra of the power and thrust coefficients for the BeamDyn-only configuration.

    The presence of this peak may be likely explained by the fact that the baseline configuration of the wind turbine used in our simulations considers an additional degree of freedom that takes into account the mechanical connection between the blades.

    We acknowledge that this aspect warrants further investigation, and we plan to examine it in more detail in future work. Nevertheless, we would like to emphasize that the amplitude of the 1P peak is several orders of magnitude smaller than the dominant spectral components at higher frequencies. As a result, its contribution to the overall system dynamics—and thus to the conclusions drawn in this study—is negligible. Therefore, while the mechanism behind this peak is of scientific interest, it does not materially affect the results presented here.

33. *This aspect is quite critical for the comparison between BEM-based and LES-based solvers. How can the authors be sure that the imposed inflow at the rotor disk between the two approaches is the same (at least to a certain extent)? An analysis of the inflow velocities on the reference blade along the azimuth angle is needed to quantify the differences in the wind incoming to the rotor. Moreover, in Task 47 many analyses have been performed (aslo by some of the authors) on how to align the inflow to BEM and to CFD (altough in that case it referred to the turbulent inflow). Anyhow, I suggest to investigate if any of those methodologies can be used here to align the inflow to BEM and to CFD.*

    We have indeed compared the aerodynamics of the present code with other ones in the IEA Task 47 "TURBINIA". In figure 4.25 of the report of Task 47 (Shepers et al. 2025), the differences in rotor aerodynamic modeling between different codes are highlighted by the time traces of hub height wind speed and axial wind velocity at a co-rotating probe on the turbine blade. Before comparing the rotor simulation results it was ensured that identical inflows were prescribed between the different codes. Figure 4.25 displays that the lifting line codes do not agree well with CFD ones on the hub height wind speed and the axial wind speed of the most outboard co-rotating probe. The differences originate from the fact that for the CFD codes the rotor induction cannot be separated from the undisturbed wind field. Hence, also when identical inflows are imposed, CFD codes have a lower level of the axial wind speed compared to lifting line ones, due to the rotor blockage. This is an intrinsic different between these two types of modeling, which do not allow to make a one-to-one comparison of the aerodynamics. Although this discussion is very relevant to this paper, we cannot include it in the present paper since those data are not public. However, we have included a sentence about this point in the paper.

34. *Also for this analysis I suggest removing the mean values from the plot (mentioning them in a table) and focusing only on the fluctuations (this intended for the phase-average results).*

    We have tried to do that but the plot became confusing, since the lines will be mostly

superposed. Thus, we have chosen not to follow this comment.

35. *Again, if a literature results needs to be used as a comparison, it is better to show it here in order to avoid the reader going back and forth from this paper to those referenced.*

    We have now included the results of Trigaux et al (2024) in figure 12.

36. *This result is espected as the lag deformation is mainly driven by gravity.*

    We have now specified that.

37. *In my opionion this conclusion is quite weak if not supported by a sensitivity analysis on the number of needed modes in Elastodyn. Moreover, to me it is still unclear if the OV model really does not take into account blade torsion (see my comment on this aspect above), so, considering the abscence of torsion in Elastodyn, the discrepancy could be related also to this (although this is in contrast with the lower value of lag deformation at the tip). This aspect requires more in depth investigation.*

    Concerning our CFD code, as explained at page 8, the CFD-CSD/OV approach considers the torsional degree of freedom, but does not include it in the two-way coupling. Whereas, Elastodyn does not consider the torsional degree of freedom at all. Thus, we agree with the referee that this discrepancy can be related also to this point. However, we cannot extend the ElastoDyn module to consider more degrees of freedom, since it is embedded in the OpenFAST code only in this form. We have thus discussed this point in the text and stated at page 12-13 that "the four solvers employed differ in both their aerodynamic and structural modeling approaches. Moreover, the flow that impacts the turbine is not exactly the same for the CFD and OpenFAST solvers, since in the former case it is imposed at several diameters upstream the rotor plane. As a result, it is not always possible to unambiguously determine whether the observed discrepancies in the results originate from the fluid-dynamic models or from the structural formulations".

38. *Please reproduce the mentioned results in your plots!*

    Done

39. *General comments to Fig 13: - the labels indicating blade eigenfrequency are hardly readable; - I suggest adding the grid lines to better appreciate the different values of the peaks. Moreover, it seems clear that blade deformation dynamics is governed by the lower components of the spectrum (the analysis of the energy content of the different harmonics should confirm this). So, although frequencies higher than the 1st flap appear, they have a very small magnitude. I suggest to reduce the x range to show only the most significant harmonics. Also the y range should be changed because as it is now it is very difficult to appreciate the peaks values. I understand that the author's aim was to show the consistency of flap/lag/torsion prediction by evidencing the role of some of the blade natural frequencies. I suggest to do this analysis only for one DOF and change the other plots using a reduced frequency range.*

    We have increased the labels and added the grid lines. Also, we have increased the y axis for making the peaks more visible. The figure has an improved readbility, we thank the referee for this comment.

40. *This aspect is not clear (see my previous comments on this).*

    Please, see the answer to the comment number 37.

41. *This sentence is quite general and not supported by specific evidences. Aeroelasticiy is a multidisciplinary topic so it must be tackled by a step-by-step approach. I agree with the choice*

*presented in this work where the structural model is assessed against available literature data. What is somehow missing is its aerodynamic counterpart. Indeed, I suggest adding a purely aerodynamic comparison between the proposed CFD and other data (FAST can be an option, but ideally solvers with a similar fidelity as the one here proposed would be more appropriate). In this way the origin of the observed discrepancies can be related to the structural model, to the aeordynamic one or to their coupling.*

Please, see the answer to question 33 of this referee.

42. *As the response of the system should be governed mainly by the frequencies arising from the combination of contribution stemming from aerodynamics, I suggest to comment the origin of the peaks shown in Fig 13. For instance, the rotor rotation frequency appears clearly (and it is dominant). What about the others?*

This has been done at page 23.

43. *This is not a limitation of OpenFAST but is related to the different inflow provided to the two solvers (see my comment on this above).*

We have now stated that in the text, although we specify that the CFD inflow is not turbulent, but has indeed variations in the transverse direction.

44. *Please reproduce here the cited result.*

In this particular case we have not included the cited results in the plot since the curve has many peaks and extracting it point-by-point would have induced large errors. However, we have included all the referenced results about the power coefficient and deformations, whose curves are rather simpler and more suitable to the extracted point-by-point.

45. *This is not a limitation of OpenFAST but is related to the different inflow provided to the two solvers (see my comment on this above).*

We have recalled here about the different inflow, although we do not think it is only due to the inflow conditions, which differ only slightly, but to the simplified treatment of aerodynamics.

46. *This very difficult to be seen from the plots in their current layout. Moreover, the ability of Blade Element Momentum Theory (which is inherently steady) to capture high-frequency aerodynamics is questionable. Considering the very small values of the harmonics shown around blade modes, I am not convinced that the BEM-based simulations can be so reliable.*

Here we do not refer to the high-frequency aerodynamics effects (i.e., turbulence), but to the aeroelastic (structural) high frequency vibrations. We have now better specified that.

**Large Eddy Simulation of the IEA 15-MW Wind Turbine Using a Two-Way Coupled Fluid-Structure Interaction Model**

Claudio Bernardi[1], Stefania Cherubini[1], Felice Manganelli[1], Giacomo Della Posta[2], Stefano Leonardi[3], and Pietro De Palma[1]

[1]Department of Mechanics, Mathematics and Management, Polytechnic University of Bari, 70126, Bari, Italy (claudio.bernardi@poliba.it, stefania.cherubini@poliba.it, f.manganelli@phd.poliba.it, pietro.depalma@poliba.it)
[2]Department of Mechanical and Aerospace Engineering, Sapienza University of Rome, Rome, RM, 00184, Italy (giacomo.dellaposta@uniroma1.it)
[3]Department of Mechanical Engineering, University of Texas at Dallas, Richardson, TX, 75080, USA (stefano.leonardi@utdallas.edu)

**Correspondence:** Claudio Bernardi (claudio.bernardi@poliba.it)

**Abstract**

The aim of the work is studying the aeroelastic response of the IEA 15 MW Reference Wind Turbine (RWT) large-scale wind turbine using a high-fidelity fluid-structure interaction solver that combines large-eddy simulation with a modal computational structural dynamics solver through a two-way coupling. The fluid solver employs the actuator line model to simulate the interaction between the turbine blades and the fluid and the immersed boundary method to model the presence of the tower and nacelle. The results are compared with those obtained by the OpenFAST software, which is a well-known numerical tool for engineering predictions. A series of simulations have been performed with and without the presence of the tower and nacelle to better understand the effects of these components on flow structures and structural deformations. The largest discrepancies among the solvers have been observed in correspondence with the blade passage in front of the tower, which induces an abrupt alteration in the local incidence angle of the flow. Moreover, by comparing the outcomes of different structural approximations, it has been established that taking into account the torsional degree of freedom considerably affects the deformations, aerodynamic loads and power coefficient. Whereas, the nonlinearity of the solver appears to have a weak effect on the same quantities.

**Keywords**

Aeroelasticity, Large Eddy Simulation, Actuator Line Model, Fluid-Structure Interaction, Computational Fluid Dynamics, Computational Structural Dynamics, Blade Element Momentum, IEA-15MW Wind Turbine.

**1  Introduction**

Wind energy has become a crucial component of the global transition toward renewable energy sources. The increasing demand for clean energy has led to the development of large-scale wind turbines, such as the IEA 15-MW offshore wind turbine developed within IEA Wind Task 37 (Gaertner et al., 2020a). This turbine, with a rotor diameter of 240 meters and blades measuring 117 meters in length, represents a new frontier in wind energy technology (Gaertner et al., 2020b), and research is currently pointing towards even larger rotors, reaching 22-MW of power production (Zahle et al., 2024). The increasing scale and flexibility of such newly designed turbines present significant engineering challenges, particularly in predicting their aeroelastic response (Burton et al., 2011; Zheng et al., 2023). As turbines grow in size, their structural components, especially the blades, are subject to complex aerodynamic forces that cause deformations, which in turn affect the aerodynamic loads. Understanding these interactions is essential to improve the performance, reliability, and longevity of large-scale wind turbines (Manwell et al., 2010). In the worst cases, aeroelastic instabilities such as edgewise instability and flutter might even lead to blade damage, as reported for the Lunderskov Mobelfabrik 19 $m$ wind turbine blades (Moeller, 1997), with devastating effects on the turbine performance.

Aeroelasticity is critical in the design and analysis of modern wind turbines. Aeroelastic phenomena such as dynamic stall, flutter, and their effects on fatigue loadings can have significant effects on turbine performance, particularly as the blade length increases (Hansen, 2007). These blades experience varying aerodynamic forces along their span, which can lead to substantial deformations. When blades deform, they alter the local flow field, which in turn modifies the aerodynamic loads acting on them. This feedback loop between aerodynamic forces and structural deformation makes it very difficult to predict modern large-scale turbine performance under real-world operating conditions (Vermeer et al., 2003; Wang et al., 2016a). Accurate evaluation of these interactions is key for ensuring turbine efficiency and structural integrity, especially in offshore environments where wind conditions are more severe (Bayati et al., 2017).

The numerical modeling of the blades in most of the numerical aeroelastic codes used nowadays (Schepers et al., 2021) is accomplished by the Blade Element Momentum (BEM) model, due to its robustness and low computational cost. It has been shown in the framework of the IEA WIND Task 47 Boorsma et al. (2023, 2024) that, if properly tuned, BEM can be a valuable engineering-type solver, complementary to higher fidelity ones which have also higher computational costs. However, BEM has still some limitations, since it relies on simplifying assumptions made on the impinging flow, such as models of dynamic stall, dynamic inflow, yaw and tilt flows, and corrections of the aerofoil data for taking into account three-dimensional effects and tip losses. More computationally expensive models exist, such as panel methods and in general potential flow solvers and/or free-vortex wake methods, as well as the actuator disc methods. Panel and free-wake vortex methods are able to capture unsteady

blade/rotor aerodynamics with good accuracy in different operating conditions (including off-design) whenever massive flow separation phenomena do not occur Boorsma et al. (2018); Ribeiro et al. (2023). However, those models need reference high-fidelity data in order to refine and/or assess the reliability of these lower fidelity models. Therefore, the application of computational fluid dynamics (CFD) to full-scale turbines is needed as a reference for describing the complex aerodynamics of the flow field accurately (Sørensen, 2011), although for a limited number of flow case due to its high computational cost.

However, coupling three-dimensional CFD simulations with computational structural dynamics (CSD) solvers taking into account the deformation of the blade is not trivial. Three-dimensional structural finite-element models are in fact able to fully describe the complex shape of a wind turbine blade but, although accurate, these models are computationally expensive and hard to implement, leading to only a few examples of coupling with CFD codes (Bazilevs et al., 2011; Yu and Kwon, 2014). Since wind turbine blades are slender structures, their structural modeling can be more easily achieved using beam models, where the blade is discretized as a series of one-dimensional beam elements, each characterised by a given cross-sectional stiffness and mass per unit length. One-dimensional beam models can be either modal, since natural frequencies and mode shapes of a turbine are directly related to the natural frequencies of its blades, or they can rely on the geometrically exact beam theory including non-linear effects (Sabale and Gopal, 2019).

Due to their ability to provide a rapid evaluation of the turbine performance, numerical tools based on the BEM approach equipped with aeroelastic modules based on one-dimensional beam models, are currently widespread (Schepers et al., 2021). A notable example is OpenFAST, a numerical code developed at NREL (Jonkman, 2013) and widely used for aeroelastic simulations, which employs BEM theory for aerodynamic modeling and various structural solvers, such as ElastoDyn (Damiani et al., 2015) and BeamDyn (Wang et al., 2016b), for structural deformation analysis. However, it is still not clear whether the predictions of such lifting-line aeroelastic codes are sufficiently accurate for large-scale turbines, in which the effect of shear and inflow turbulence can lead to complex inflows and turbine aerodynamic responses. Comparing the predictions of OpenFAST with those of a Large-Eddy Simulation (LES) equipped with a structural one-dimensional beam model has shown that, for an NREL 5MW wind turbine, the passage in front of the tower leads to large deformations which are largely underestimated by OpenFAST (Bernardi et al., 2023).

Concerning rotors of even larger size, such as the IEA 15-MW reference turbine, it is not yet known whether these discrepancies in the predictions of lifting-line codes with respect to CFD are even more consistent. Using the unsteady Reynolds-Averaged Navier-Stokes (URANS) equations coupled with an aeroelastic module, as reported by Pagamonci et al. (2023), has shown that neglecting the flexibility of the blades in numerical simulations leads to an underestimation of the rotor thrust of approximately 2.5% for the IEA 15-MW turbine, which is not observed for the smaller NREL 5MW rotor. Moreover, this work also concluded that the deformation of long, slender blades may act as a filter for the high-frequency fluctuations arising from the flow field, proving that taking into account the blades' aeroelasticity in the design process of these machines is key for the future upscaling of turbine rotors. Furthermore, Trigaux et al. (2024) observed how the use of high-fidelity aerodynamic models is crucial to predict the aeroelastic effects of large rotors. These results suggest the need to investigate this issue resorting to LES, which is capable of describing the dynamics of the flow more accurately.

In this context, the present work aims at studying the aeroelastic response of a large-scale 15-MW wind turbine by means of LES, assessing the effect of the flexibility of the blades on the wake dynamics. The results are compared with those obtained by more simple and less computationally expensive models, such as the OpenFAST code. Computations are performed by an in-house LES code using the immersed boundary method to model the tower and nacelle and the Actuator Line Model (ALM) for blade modeling, coupled with a structural modal solver, originally developed by Della Posta et al. (2022).

The discussion of the results highlights the role of the tower and nacelle in the dynamics of the aerodynamical forces, thrust and power coefficients, as well as in the distribution of turbulent kinetic energy within the wake, which could have an impact on the aerodynamic loads of downstream turbines in wind farms. Moreover, the effect of the torsional degree of freedom has been investigated by comparing the outcomes of different structural approximations.

The work is structured as follows. In section 2, the aerodynamic and structural solvers of both CFD-CSD and OpenFAST codes are described in detail. In section 3, the numerical setup is presented. In section 4, relevant results are discussed, and conclusions are drawn in section 5.

**2 Methodologies**

**2.1 CFD-CSD solver**

**2.1.1 Flow solver**

The simulations of the flow around the wind turbine are carried out through Large-Eddy Simulations (LESs) of the incompressible, filtered, 3D Navier-Stokes equations, employing the in-house UTD-WF solver introduced by Santoni et al. (2015). The UTD-WF framework has been progressively developed by Santoni et al. (2017, 2020) and further extended by Della Posta et al. (2022, 2023), where the aeroelastic solver and the Leishman–Beddoes dynamic stall model were implemented. The solver has been validated in its non-aeroelastic version by Santoni et al. (2017) against wind-tunnel data reproducing the NTNU "Blind Test" and comparing simulations to Krogstad et al. (2015) measurements, also considering the impact of tower and nacelle. Whereas, the recently developed version of the code including the two-way FSI coupling Della Posta et al. (2023) has been validated through comparison against reference datasets, including HAWC2-based results reported by Heinz (2013). The IEA 15MW wind turbine configuration cosidered here has been cross-validated with many other aeroelastic numerical codes in the International Energy Agency (IEA) Wind TCP Task 47 (Cacciola et al., 2025), also considering turbulent inflow conditions (Schepers et al., 2025). Notice that prior validations by Della Posta et al. (2022) of the CFD-CSD solver were made on a laminar uniform and a turbulent sheared inflows for a 5 MW NREL turbine, whereas our study extends the validated setting to the IEA-15 MW case for a sheared laminar inflow configuration. However, as discussed in the framework of the IEA Wind TCP Task 47 Schepers et al. (2025), turbulent fluctuations appear to have a much stronger impact than shear on load response of aero-elastic numerical codes. Moreover, high-fidelity codes appear rather consistent in predicting loads, while engineering models tend to overpredict fatigue loads, particularly for large rotors (Cacciola et al., 2025).

The code implements a second-order accurate centered finite difference scheme for the spatial discretization on a staggered Cartesian grid. A hybrid low-storage third-order-accurate Runge–Kutta (RK) scheme is used for time integration of the non-linear terms (Orlandi, 2012), while the linear terms are treated implicitly using a Crank-Nicolson scheme. The filtered governing equations are:

$$\frac{\partial u_i}{\partial t} + \frac{\partial u_i u_j}{\partial x_j} = -\frac{\partial p}{\partial x_i} + \frac{1}{Re}\frac{\partial^2 u_i}{\partial x_j \partial x_j} - \frac{\partial \tau_{ij}}{\partial x_j} + \tilde{f}_i, \tag{1}$$

$$\frac{\partial u_i}{\partial x_i} = 0, \tag{2}$$

where $i, j \in \{1, 2, 3\}$ represent, in a Cartesian reference frame, the components along the streamwise (x), wall-normal (y), and spanwise (z) directions, respectively. The Reynolds number $Re = U_\infty D/\nu$ is defined by the undisturbed inlet velocity $U_\infty$, the turbine diameter $D$, and the kinematic viscosity of the fluid $\nu$. These quantities are used as reference values to make the equations non-dimensional. To solve the filtered equations, a Subgrid-Scale (SGS) stress model is needed. The latter describes the interaction between the large resolved and the sub-grid unresolved scales, as described by Pino Martín et al. (2000) and Santoni et al. (2017). Here, we employ the Smagorinsky model with constant $C_s = 0.09$ as discussed by Martinez-Tossas et al. (2018).

The effect of the blades on the flow is modeled by the Actuator Line Model (ALM) (Sorensen and Shen, 2002), by adding a forcing term to the Navier-Stokes equations, representing the force per unit volume exerted by the rotor on the fluid. By approximating the rotor blades as straight lines discretized into segments, it is possible to estimate the lift and drag forces per unit length on a 2D plane as follows:

$$F_l = \frac{1}{2}\rho u_{rel}^2 C_l(\alpha)cF, \qquad F_d = \frac{1}{2}\rho u_{rel}^2 C_d(\alpha)cF, \tag{3}$$

where $\rho$ is the air density, $c$ is the local chord, $u_{rel}$ is the relative incoming velocity, $\alpha$ is the angle of attack, and $F$ represents the tip loss correction factor, which employs the tip-loss model proposed by Shen et al. (2005). The coefficients $c_1$ and $c_2$ of this model have been set in the following way: $c_1$ has been set to the value reported in the Shen et al. (2005) paper ($c_1 = 0.125$), whereas, $c_2$ has been chosen after a calibration with respect to the forces close to the tip reported by OpenFAST for the same turbine and flow case, leading to the choice of $c_2 = 32$. The forces are then projected on the flow employing a 2D Gaussian kernel, which spreads the lift and drag force vector, $\boldsymbol{f}^{aero}$, in cylinders surrounding the actuator line,

$$\tilde{\boldsymbol{f}} = -\boldsymbol{f}^{aero}\frac{1}{\epsilon^2\pi}exp\left[-\left(\frac{r_\eta}{\epsilon}\right)^2\right], \tag{4}$$

where $r_\eta$ is the radial distance of a generic point of the cylinder from the actuator line and $\epsilon$ is the spreading parameter, where $\epsilon/\Delta \geq 2$, with $\Delta = \sqrt{\Delta x^2 + \Delta y^2 + \Delta z^2}$, following Troldborg (2009). The tower and nacelle are modeled using the Immersed Boundary Method (IBM) following the approach described by Orlandi and Leonardi (2006).

[Figure]

Figure 1: Sketch of the frames of reference used for the CFD and for the CSD simulations.

**2.1.2 Structural solver**

From an aerodynamic standpoint, the rotor blades represent the most flexible components within a wind turbine. Several studies demonstrated that their modal properties have a significant impact on the dynamics of the entire structure (Damgaard et al., 2013; Dong et al., 2018). Moreover, an analysis of the isolated blades is also sufficient to accurately estimate the aeroelastic properties of the entire structure, including the flutter speed (Abdel Hafeez and El-Badawy, 2018). Additionally, the tower and shaft exhibit minimal deflection due to their stiffness. In light of the above considerations, the aeroelastic model is constructed to encompass solely the structure of the blades.

The structural model used in the present study was extensively described by Della Posta et al. (2022, 2023) and will only be briefly outlined here. Under the assumption of small deformations with respect to a relative frame of reference (FOR), the blades are assumed to be rotating beams rigidly clamped at the hub (cantilever beams). Moreover, it is assumed that the blade deformation does not modify the rotor inertia. With these hypothesis, a linear structural dynamic equation is obtained, taking into account the Coriolis, centrifugal and Euler effects, that will be given in the following. Let us denote by $X_1$ the direction of the pitching axis. This coincides with the neutral axis of the blade, defined as passing through the quarter of the chord. The direction of the out-of-plane flapwise motion is indicated by $X_2$ and is oriented in the positive streamwise direction. The in-plane edgewise direction of $X_3$ is defined such that the FOR is oriented as a right-handed coordinate system (Figure 1). Under the assumption of linearity, the elastic generalised displacement $\boldsymbol{d} = (d_i, \theta_i)$, which includes translational $d_i$ and rotational $\theta_i$ (with $i = \{1, 2, 3\}$) degrees of freedom (DoFs), is decomposed along the coordinate $X_1$ on the neutral axis as:

$$\boldsymbol{d}\left(X_1,t\right) = \sum_{m=1}^{M} q_m(t)\,\boldsymbol{\psi}^m\left(X_1\right), \tag{5}$$

where $\psi^m\left(X_1\right)$ is the m-th elastic mode shape from the modal analysis of the structure, $q_m$ is the corresponding modal coordinate, and $M$ is the number of modes used. The effect of the generic motion of the FOR on the relative structural dynamics (one-way inertial coupling, since we assumed that the blade deformation does not modify the rotor inertia) is included in a modal basis by means of the methodology introduced in Reschke (2005) and further developed for the case of wind energy in Della Posta et al. (2022). Through this method, which exploits the decomposition of the acceleration in a moving FOR in the virtual work principle, we obtained a system of elastic equations with additional stiffening, damping, and loading terms depending on the angular velocity and acceleration of the rotating FOR, as:

$$\boldsymbol{M}\ddot{\boldsymbol{q}} + [\boldsymbol{D} + \boldsymbol{D}^{Co}(\boldsymbol{\Omega})]\dot{\boldsymbol{q}} + [\boldsymbol{K} + \boldsymbol{K}^c(\boldsymbol{\Omega}) + \boldsymbol{K}^{Eu}(\dot{\boldsymbol{\Omega}})]\boldsymbol{q} = \boldsymbol{e} + \boldsymbol{e}^c(\boldsymbol{\Omega}) + \boldsymbol{e}^{Eu}(\dot{\boldsymbol{\Omega}}), \tag{6}$$

where $\boldsymbol{M}$, $\boldsymbol{D}$ and $\boldsymbol{K}$ denote the modal structural mass, damping, and stiffness matrices, respectively, and $\boldsymbol{e}$ are the external loads expressed in modal basis, including the gravity force acting on the local centre of mass and the ALM aerodynamic forces acting on the local quarter of chord. The remaining terms are inherently related to the various contributions to the acceleration in a moving FOR. Terms with the superscript $Co$, $c$ and $Eu$ are related to the Coriolis, centrifugal, and Euler accelerations, respectively. Given the assumption of linearity, we apply all the forces to the reference undeformed configuration. The discrete evaluation of the additional inertial terms in Equation (6) is expressed as a function only of the information known from the structural finite-element method (FEM) model and from the corresponding mode shapes, according to Saltari et al. (2017). For the modal analysis, performed on the undeformed nonrotating blade, we use a finite element model of the blade based on complete beam elements with 6 DoFs, with Euler-Bernoulli behavior for bending in directions $X_2$ and $X_3$, and linear shape functions for axial and torsional deformations. We assume a lumped-mass representation, and we take into account the local offset of the centers of mass with respect to $X_1$. Finally, the structural matrices are assembled considering the local twist. The generalized-$\alpha$ method (Chung and Hulbert, 1993) is employed to advance the structural dynamic equation in time, which is unconditionally stable for linear problems, and second-order accurate. Details about the modal analysis are provided in Appendix A.

**2.1.3 Fluid-structure interaction model**

The two-way coupling aeroelastic model employs the ALM sectional approach, whereby the angle of attack (AoA) and relative velocity are locally modified following the instantaneous blade motion provided by the structural dynamics. In particular, the distribution of the AoA along each blade is evaluated as a function of the velocity of the fluid, the angular velocity of the rotor, and the instantaneous elastic state of the blade (which is projected back to the physical space from the modal one once the displacement is determined). The latter is generally constructed from the deformation velocity $\boldsymbol{u}_{def} = \dot{\boldsymbol{d}}_{\boldsymbol{tr}}$, considering the time derivative of the translational degrees of freedom only, and the local vector of the deformation angles $\boldsymbol{\theta}$ (torsion, and in-/out-of-plane angular deformations)

derived from the structural solver, which is forced by the updated aerodynamic loads. The algorithm restricts inter-field communications solely at the beginning of each RK substep, thereby ensuring optimal computational efficiency. The impact of the torsional dynamics was deemed to be limited in light of the results obtained in previous studies on the effect of torsion for smaller wind turbines (Chen, 2017). In order to investigate this issue for the large rotor 15MW wind turbine, in this study we compare two different CSD models. In particular, we consider as a baseline a two-way coupling that includes the effect of blade deformation velocity as a sole variable (CFD-CSD/OV, for Only Velocity), and a more complete model including the torsional deformation in the coupling (CFD-CSD/T, for Torsional). In general, the relative velocity for a rotating blade can be defined with the following expression:

$$\boldsymbol{u}_{rel} = \boldsymbol{u}_{abs} - \boldsymbol{\Omega} \times \boldsymbol{r}_{OP} - \boldsymbol{u}_{def}, \tag{7}$$

where $\boldsymbol{u}_{abs}$ is the filtered velocity from the fluid solver at the actuator line, $\boldsymbol{r}_{OP}$ is the general radial vector pointing to the considered section, $\Omega$ is the rotor rotational speed, and $\boldsymbol{u}_{def}$ is the deformation velocity of the structure at the same position. As a result, the AoA used to determine the air load coefficients is defined as follows:

$$\alpha = \operatorname{atan}\left(\frac{\boldsymbol{u}_{rel} \cdot \boldsymbol{E}_2}{-\boldsymbol{u}_{rel} \cdot \boldsymbol{E}_3}\right) - \phi - \theta_{tors} = \operatorname{atan}\left[\frac{(\boldsymbol{u}_{abs} - \boldsymbol{u}_{def}) \cdot \boldsymbol{E}_2}{\Omega r - (\boldsymbol{u}_{abs} - \boldsymbol{u}_{def}) \cdot \boldsymbol{E}_3}\right] - \phi - \theta_{tors}, \tag{8}$$

where $\phi$ is the local twist angle of the blade, $\theta_{tors}$ is the local torsional deformation, $\boldsymbol{E}_i$ are the unit vectors of the relative FOR rotating with the structure, and hence, $v_2 = \boldsymbol{u}_{def} \cdot \boldsymbol{E}_2$ is the flapwise deformation velocity component, and $v_3 = \boldsymbol{u}_{def} \cdot \boldsymbol{E}_3$ is the edgewise deformation velocity component. The simplified coupling procedure benefits from the sectional one-dimensional formulation of the ALM, which avoids the complex treatment of the fluid-solid interface with the associated kinematic and traction conditions.

**3 Flow and structural setup**

In this work, we consider a stand-alone IEA 15-MW wind turbine (Gaertner et al., 2020b) in its monopile configuration. This wind turbine has a rotor diameter $D = 240$ m with three blades of length $L = 117$ m. Table 1 provides the main features of the turbine.

The computational domain has dimensions $12.5 \times 5 \times 3$ diameter units, as shown in Figure 2. The distance of the turbine from the inlet of the computational domain (equal to 4D) has been determined on the base of the reference data available in the literature, which vary in the range 2D-5D. Smaller distances from the inlet (2D) have been employed for experimental set-up (Bartl and Satran, 2017; Krogstad et al., 2015), whereas larger distances (in the range 2.7D-5D) are typical of numerical simulations (Porte-Agel and Wu, 2011; Ciri et al., 2017; Allah and Sha ei Mayam, 2017; Stevens et al., 2018). Moreover, we have verified numerically that pressure fluctuations do not generate spurious reflections at the inlet section in our simulations. The spanwise length of the computational domain (equal to 3D) is the same employed in previous numerical simulations (Ciri et al., 2017; Allah and Sha ei Mayam, 2017). We have verified that, using periodic boundary conditions, the blockage effect

on the single turbine is negligible. Moreover, following the convergence study reported in the Appendix A, the computational box has been discretized by a staggered grid composed of $2049 \times 513 \times 513$ points in the streamwise, wall-normal, and spanwise directions, respectively. The orthogonal grid is equally spaced in the streamwise and spanwise directions and is stretched vertically, with a gradually wider spacing starting from the region above the rotor. The grid spacing described leads to an actuator line discretized by 86 points per blade. The time resolution of the LES computation is tied to the spatial resolution, as defined by the stability requirements of the numerical scheme adopted. Simulations are carried out at a constant Courant–Friedrichs–Lewy (CFL) number (Courant et al., 1967) $CFL = 0.65$, which ensures an average time step $\overline{\Delta t} = 0.024s$. The turbine location is 4 diameter units from the inlet and centered in the spanwise direction. Furthermore, we impose a sheared laminar inflow velocity profile, defined by a power law with the exponent $\alpha = 0.05$, and a convective outlet boundary condition, i.e., $\frac{\partial u_i}{\partial t} + C \frac{\partial u_i}{\partial x} = 0$, with the constant $C$ set to the average value of the outflow velocity. Notice that, as the shear is imposed at the inlet, the flow profile is allowed to change when reaching the turbine. However, since the power law profile complies with the no-slip conditions at the wall and with the slip conditions at the free-stream, the modifications are mostly due to the slight three-dimensionalization of the flow due to the presence of the turbine. In the spanwise direction, periodic boundary conditions are imposed. Moreover, slip and no-slip conditions are enforced at the top and bottom boundaries, respectively. The turbine is subjected to a flow with a Reynolds number $Re \approx 10^8$ and operates at its nominal tip speed ratio (TSR) of $\lambda = 9$. The streamwise undisturbed velocity at the hub height is constant and equal to $U_\infty = 10 \ m/s$. The simulations were conducted for a time interval of 300 $s$ over the initial transient, which corresponds to 35 revolutions of the rotor.

To identify the optimal configuration for the structural model, we conducted a preliminary sensitivity analysis and then validated the structural eigenfrequencies of the undeformed nonrotating blade with the results found in the literature. A more detailed insight into this analysis is presented in Appendix B, where the structural properties of this turbine are shown. Finally, a number of modes $M_s = 15$ and a structural discretization of the blades given by $N = 80$ equally-spaced nodes were chosen.

For comparison purposes, wind turbine simulations have been also conducted using the OpenFAST solver *Release v3.2.0* (July 29, 2022). The aerodynamic computations are performed by the *AeroDyn* (Jonkman et al., 2015) module which is based on the BEM theory. A Prandtl loss model is applied to account for the tip and root effects. The structural module dedicated to the computation of the blade deformation is contained in the *BeamDyn* module, which relies on the geometrically exact beam theory and may resolve geometric non-linearities and large deflections (Wang et al., 2016b). In order to compare the CFD-CSD results with a modal structural analysis, we also performed simulations using the standalone *ElastoDyn* module, based on a modal approach and suitable for blade deformation dominated by bending. It is worth to notice that the latter does not take into account the torsional degree of freedom, so it is to be directly compared to the CFD-CSD/OV model, which also does not account for the coupling between the torsional deformation and the angle of attack. As reported in the original manual of *AeroDyn* (Moriarty and Hansen, 2005), OpenFAST couples the fluid and structural solvers in a similar way to our CFD-CSD solvers. In particular, the local angle of attack is determined taking into account the local deformation velocities.

| Parameter | Units | Value |
|---|---|---|
| Power rating | $MW$ | 15 |
| Rotor diameter ($D$) | $m$ | 240 |
| Rotor orientation | $-$ | Upwind |
| Number of blades | $-$ | 3 |
| Blade length ($L$) | $m$ | 117 |
| Hub height | $m$ | 150 |
| Hub radius ($R_{hub}$) | $m$ | 3.97 |
| Rated wind speed | $m/s$ | 10.59 |
| Design tip speed ratio | $-$ | 9 |
| Maximum rotor speed | $RPM$ | 7.56 |

Table 1: IEA 15-MW (Gaertner et al., 2020b) wind turbine main features

[Figure]

Figure 2: Sketch of the computational domain where the incoming sheared flow and the position of the turbine are highlighted.

**4 Results and discussion**

This section presents the results of two set of simulations: one modeling a rotor-only configuration (RO) and the other including the tower and nacelle (TN). Furthermore, both configurations are subjected to comparative analysis using the OpenFAST submodules. Firstly, the near-wake aerodynamic characteristics and the wake recovery of both configurations determined by the CFD-CSD solvers are discussed. Then, the aerodynamic loads on the blades are analyzed and the outcomes from both solvers are compared. Finally, the overall turbine performance and the effects on the blade deformation are assessed.

[Figure]

Figure 3: Q-criterion contour of the instantaneous velocity field colored by the streamwise velocity for the rotor-only case (RO) (a) and tower and nacelle (TN) (b).

**4.1 Flow analysis**

As a first step, we analyze the flow field variables, as obtained using the CFD-CSD/T solver. Figure 3 illustrates the main coherent flow structures in the field by means of an instantaneous isosurface of the Q-criterion colored by the streamwise velocity for both cases. It is evident that the presence of the tower affects the vorticity intensity distribution along the vertical direction. In particular, the occurrence of a low-velocity recirculation zone at the tower height for the TN case can be identified, which is a result of the tower shadowing (see Figure 3b). Moreover, the TN case demonstrates a more rapid dissolution of the endogenous coherent hub vortex structures if compared to the RO case (see Figure 3a). On the other hand, the tip vortex structures appear to be minimally influenced by the presence of the tower. Figure 4 shows the rotor-averaged streamwise velocity along the flow direction, time-averaged over 30 revolutions of the rotor. Contrary to what Santoni et al. (2017) observed in their work on the 5MW reference turbine invested by a uniform inflow (see the red lines in figure 4), the rotor-averaged velocity for the TN configuration in the wake remains slightly lower than for the OR case, indicating that wake recovery is slightly hindered by the presence of the tower. Although further validation is required as the result does not fully align with this previous literature study, wake recovery appears thus to be hindered by tower presence. One possible explanation for this behaviour could be differences in the tower-to-rotor aspect ratio. In particular, for the NREL 5-MW turbine, the ratio between the tower diameter and the rotor diameter is about equal to 0.047, whereas, for the 15MW turbine, it is only about 0.027 (the tower diameters being $6m$ and $6.5m$, respectively). Thus, the thinner shape (in terms of diameter units) of the tower, as well as the lower value of the incoming velocity at the tower height due to the presence of shear at the inflow, result into a decreased mixing behind the turbine which leads to a slower wake recovery.

From an energy perspective, the wake recovery process can be depicted by examining the Turbulent Kinetic Energy (TKE) in the wake. Figure 5 represents the time-averaged TKE for both configurations on different planes. The TN case exhibits high TKE values in the near wake, in the region just

[Figure]

Figure 4: Rotor-averaged velocity along the streamwise direction normalized by the undisturbed velocity at the rotor height, namely, $U_\infty = 10 \ m/s$, for the present data (black curves) and the work of Santoni et al. (2017) (red curves). The grey region represents the area covered by the rotor. (RO -----, TN ——).

downstream of the tower and nacelle. The top view of the TN case shows that the TKE in the wake presents an asymmetric distribution as De Cillis et al. (2022) observed, among the others, in their work. On the contrary, the RO configuration shows large TKE only in the far wake region, with large values also in the region above hub height. This result may indicate that the tower does not increase the kinetic energy entrainment but it rather has a slight shielding effect on wake recovery. Although not favoring kinetic energy entrainment, the tower still plays a strong role in the wake dynamics, as it can be visualized in Figure 6, showing slices of instantaneous streamwise velocity at different tower heights corresponding to 80% of the blade (top) and to the tip of the blade (bottom), when the blade is in front of the tower, i.e. $\theta = 180°$ (left), and when it is far from it (right). In particular, it can be observed that the turbulent mixing right downstream of the tower is already very high in the near wake compared to that close to the tip of the blades. Due to the mutual effect of the asymmetry induced by the rotation of the blades and of the wake meandering, it can be seen that, inside the rotor disk, the tower wake bends in the spanwise direction (Figure 6, top frames), whereas it is rather spanwise independent at a height corresponding to the blade's tip (bottom frames). Moreover, one can see that the passage of the blade in front of the tower (left frames) induces a strong perturbation in the flow field already upstream of the tower. In the following section, the effect of this perturbation on the phase oscillations of several relevant quantities (aerodynamic forces, power coefficient, etc.) will be discussed.

**4.2 Aerodynamic loads on the blade**

The analysis of the aerodynamic loads on the blade has been conducted using the present CFD-CSD models and the engineering software OpenFAST. The same laminar sheared inflow is imposed for both solvers using a power law with the same exponent and reference streamwise velocity at the hub height. We have chosen not to impose a turbulent inflow to avoid differences in the definition of the turbulent inflow itself which might have hindered the comparison between the results of the two codes. It is important to note that the four solvers employed differ in both their aerodynamic and structural

[Figure]

Figure 5: Top (first and third slice) and lateral (second and fourth slice) views of the time-averaged Turbulent Kinetic Energy on slices passing through the hub. TN (first and second slice), RO (third and fourth slice).

modeling approaches. Moreover, the flow that impacts the turbine is not exactly the same for the CFD and OpenFAST solvers, since in the former case it is imposed at several diameters upstream the rotor plane. As a result, it is not always possible to unambiguously determine whether the observed discrepancies in the results originate from the fluid-dynamic models or from the structural formulations.

Figure 7 depicts the following time-averaged aerodynamic quantities along the span of the blade: the local angle of attack $\alpha$ (Figure 7a); the aerodynamic pitching moment per unit length $M_{aero}$ (Figure 7b); the flapwise and edgewise components (normal and tangential to the rotor disk, respectively) of the aerodynamic force per unit length $F_2$ (Figure 7c) and $F_3$ (Figure 7d), respectively. In particular,

[Figure]

Figure 6: Instantaneous streamwise velocity on horizontal slices at different tower heights corresponding to 80% of the blade (top slices), and the tip of the blade (bottom slices). In the left configuration, the blade is in front of the tower ($\theta = 180°$), while on the right the blade is far from the tower. $h_{hub} = 0.625D$ is the hub height.

Figure 7a shows that a good agreement of the local incidence angle computed by both CFD-CSD models (solid lines) with that computed by *ElastoDyn* (circles) and *BeamDyn* (squares) is obtained from the 20% up to the 80% of the blade length. Indeed, the differences in the root area could ascribable to the presence of the hub which is modeled differently by the solvers. The discrepancy of the incidence angle observed towards the tip subsequently affects the aerodynamic loads. The $F_2$ force in Figure 7c shows a very good fit of the CFD-CSD/T results with that of the nonlinear solver *BeamDyn*, despite the linearity of our in-house CSD model. The strong discrepancies with respect to the values obtained by *ElastoDyn* can be ascribed to the absence of the torsional deformation in the latter solver. Indeed, the CFD-CDS/OV solver, which neglects the torsional feedback in the coupling, shows very similar results to the *ElastoDyn* solver. A similar effect can be observed by examining the reduction in $F_3$ towards the tip of the blade (see Figure 7d). The distribution of the aerodynamic pitching moment presents instead a maximum gap of about 8% from the BEM-based solvers.
As demonstrated by Hansen (2015), the outer third of the blade span is the most critical region in terms of deflections and deformations due to the combination of higher aerodynamic loads and reduced structural stiffness. Therefore, a phase average of the aerodynamic quantities at the 80% of the blade

[Figure]

Figure 7: Average aerodynamic quantities along the blade compared between CFD-CSD/OV, CFD-CSD/T, *ElastoDyn*, and *BeamDyn*. (a) Incidence angle, (b) Aerodynamic pitch moment, (c) flapwise aerodynamic force, (d) edgewise aerodynamic force.

has been performed. Figure 8 reports the evolution of the incidence angle and of the aerodynamic force components at $\frac{r-R_{hub}}{L} = 0.8$ (being $R_{hub}$ the hub radius and $L$ the blade length) versus the blade rotation angle $\theta$. The dynamical behavior of the aerodynamic quantities in the presence (solid lines) or in the absence (dashed lines) of the tower underlines that the passage of the blade in front of the tower represents the main source of instability for the flow conditions considered. Indeed, the blade-tower interaction leads to oscillations of the aerodynamic forces and of the incidence angle around $\theta = 180°$, i.e., when the blade is pointing down. However, unlike the case of the NREL 5-MW turbine (Bernardi et al., 2023), this effect appears to be stronger for the BEM computations than for the CFD-CSD solver. Concerning this point, we should recall that, as pointed out by Bernardi et al. (2023), the complex flow dynamics resulting from the interaction between the blade and the tower, shown in Figure 6, may not be well described by OpenFAST, which uses a simple potential flow model. It can be observed that, between the rotor and the tower, a region with low streamwise velocity is observed. We can expect that the passage of the blade in front of the tower thus induces an alteration of the aerodynamic forces on the blade due to the decrease/increase of the streamwise velocity. This issue will be further discussed in the following, where a possible reason for the different behavior observed for the IEA 15-MW with respect to the NREL 5-MW turbine will be discussed.

Apart from the effect of the tower, one can observe a rather good match between the CFD-CSD/OV and *ElastoDyn* solvers for both the incidence angle and the edgewise component of the aerodynamic

[Figure]

Figure 8: Phase-averaged values of: (a) the local incidence angle, (b) flapwise aerodynamic force, and (c) edgewise aerodynamic force at the 80% of the blade. CFD-CSD/OV: TN ——, RO - - - -. CFD-CSD/T: TN ——, RO - - - -. *ElastoDyn*: TN ——, RO - - - -. *BeamDyn*: TN ——, RO - - - -.

force, while the flapwise component presents some discrepancies. On the other hand, when torsional feedback is included, CFD-CSD/T and *BeamDyn* solvers, regardless of the linearity or non-linearity of the models, agree rather well on the aerodynamic forces, especially on the flapwise one, which shows an error $\approx 2\%$, while the edgewise force reaches a $\approx 5\%$ error at azimuthal angles close to $\theta = 0$. Whereas, the error between the two solvers on the angle of attack reaches 8%.

To better investigate the local response of the different models during the blade revolution, we conducted a comparative analysis of the aerodynamic loads, employing phase-averaged quantities over the span. Figure 9 illustrates the percentage difference of the phase-averaged aerodynamic quantities on the rotor plane of the *ElastoDyn* (*BeamDyn*) solver with respect to the CFD-CSD/OV , defined as $|\langle \Delta\alpha/\alpha^{CFD-CSD/OV}\rangle^{\%}|$, and of the CFD-CSD/T model, defined as $|\langle \Delta\alpha/\alpha^{CFD-CSD/T}\rangle^{\%}|$, respectively. In particular, in comparison to *ElastoDyn*, a higher value of the absolute incidence angle in the range of $|\langle \Delta\alpha/\alpha^{CFD-CSD/OV}\rangle^{\%}| = [17\%, 25\%]$ is found in the zone after the tower (see Figure 9a). The difference with respect to the results obtained by *BeamDyn* tends to be higher moving from the root to the tip with a discontinuity in the tower area, spanning the range $|\langle \Delta\alpha/\alpha^{CFD}\rangle^{\%}| = [35\%, 60\%]$ in the last 20% of the blade span. Furthermore, the angle of attack distribution affects the components of the aerodynamic force. In fact, the distribution of the flapwise component of the force follows the same pattern of the incidence angle (see Figure 9b). On the other hand, for the edgewise component the major discrepancies are concentrated in the final radial sections of the blade toward the tip (see Figure 9c). In general, we can conclude that the most significant discrepancies are observed in the tip region where the three-dimensional effects are more relevant and where the complexity of the fluid flow is strongly affected by the presence of the tower.

Notably, similar discrepancies are observed when comparing the CFD-CSD/T solver with the *BeamDyn* solvers. However, in this case some high-frequency oscillations are observed for the three aerodynamic quantities. In fact, the same oscillations are observed in the phase averaged quantities at 80% of the blade shown in Figure 9, for both the CFD-CSD/T solver and *BeamDyn*. The frequency of these oscillations computed by the two solvers appear very close and comparable with the natural frequency of the first torsional mode, although some differences can be observed in the amplitudes of the

[Figure]

Figure 9: Phase-averaged contour plots of the percentual differences of the aerodynamic quantities between CFD-CSD/OV versus *ElastoDyn* (left), and CFD-CSD/T versus *BeamDyn* (right), respectively. (a) Incidence angle, (b) flapwise aerodynamic force, (c) edgewise aerodynamic force.

signals, especially concerning the angle of attack ($\approx 8\%$ of error) and the edgewise aerodynamic force at azimuthal angles close to zero ($\approx 6\%$ of error). Again, this observation indicates that including the torsional degree of freedom in the structural solver is crucial for describing accurately the amplitude and dynamical behaviour of the aerodynamic quantities.

**4.3 Power and thrust coefficients**

The aerodynamic loads previously presented are also useful to evaluate the power and thrust coefficients, defined as follows:

$$C_p = \frac{P_d}{\frac{1}{2}\rho A U_\infty^3}, \quad C_t = \frac{T_{aero}}{\frac{1}{2}\rho A U_\infty^2}, \tag{9}$$

where $A = \pi D^2/4$ represents the rotor area, $P_d$ is the aerodynamic power transferred to the rotor and $T_{aero}$ is the overall aerodynamic thrust on the turbine.

Starting from the time history of $C_p$ and $C_t$, we computed their phase-averaged evolution as reported

[Figure]

Figure 10: Phase-averaged power (a) and thrust (b) coefficients. CFD-CSD/OV: TN ——, RO - - - -.
CFD-CSD/T: TN ——, RO - - - -. *ElastoDyn*: TN ——, RO - - - -. *BeamDyn*: TN ——, RO - - - -.
From figure 3 of Bernardi et al. (2023): ∘ *LES + CSD flexible*, × *OpenFAST-AeroDyn*.

in Figure 10. The periodic passage of the blades in front of the tower for the TN configuration produces
a drop of the curves of about 10%. Eventually, the performance is restored to the value obtained in the
RO case following the elastic dynamical behavior of the structure. The results reflect the dependency of
the power and thrust coefficients on the edgewise aerodynamic force $F_3$ and the flapwise aerodynamic
force $F_2$ at the 80% of the blade, respectively (see Figures 8c and 8b), which are strongly influenced
by the presence of the tower. Notice that, also here, we can observe that the drop in the $C_p$ curve
appears to be rather similarly predicted by BEM and CFD, although the BEM prediction exhibits
notable oscillations before and after the drop, whereas these are not present in the CFD results. A
different behaviour was observed for the NREL 5-MW turbine (as in figure 3 of Bernardi et al. (2023),
included in Figure 10 of the present paper with symbols), where this performance drop is considerably
underestimated by the BEM computations. A possible factor that may contribute to this different
behaviour may reside in the different relative geometry of the two wind turbines. Indeed, the flow
induced by a thinner tower (in diameter units), as in the case of the 15-MW wind turbine, might be
better described by a potential flow solution compared to the one induced by a thicker tower, as in
the case of the 5-MW wind turbine, and may thus lead to the observed improved agreement between
BEM and CFD results. Moreover, the differences in the flow impinging on the blade might also have
an effect. In fact, in Bernardi et al. (2023) a uniform inflow was imposed. Whereas, in the present
case, due to the shear imposed at the inflow and the limited distance from the ground of the tip of the
blade (only $\approx 0.125D$ for the 15MW turbine), the blade is invested by a flow having a much smaller
velocity compared to the given value of $U_\infty$ at hub height, further confirming the increased suitability
of a potential flow solution upstream of the tower. Nevertheless, we should recall that this remains a
very strong approximation, as also demonstrated by the differences in the forces and angles that have
been observed in the previous section (see Figure 9, for instance).
It can be concluded that the performance loss induced by the passage in front of the tower is less

pronounced for the 15 MW NREL turbine in the present configuration ($\approx 5\%$) compared to the 5 MW turbine in the configuration considered in Bernardi et al. (2023)($\approx 15\%$, see figure 3 of this reference), with both BEM theory and CFD yielding similar predictions in the case of the 15 MW turbine. However, it is worth recalling again that Bernardi et al. (2023) considered a uniform inflow, whereas here the inflow is sheared. This can be a possible reason for this different behaviour, since the lower wind speed in the lower part of the rotor plane leads to a lower production in the bottom half of the rotor plane, where the tower is located. This may cause a smaller performance drop due to the tower relative to the total produced power. Therefore, the observed difference can be not only due to the change in turbine size, but also due to the change in inflow conditions.

Moreover, the present results predict that, for very large rotors and a sheared inflow, the tower effect on blade deformations is less pronounced than for smaller rotors, although it should yet be taken into account for accurately describing the turbine's performance oscillations as it still represents a major source of unsteadiness.

The average value of the power coefficient is much larger when the torsional deformation is neglected. This feature is observed by both CFD and BEM approaches. However, one can observe that *ElastoDyn* underestimates the value of $C_p$ with respect to the corresponding non-torsional CFD model, while the opposite is observed when comparing *BeamDyn* with the torsional CFD solver. This is probably due to the fact that *BeamDyn* predicts higher values of the aerodynamic edgewise forces with respect to the CFD-CSD/T approach, which are linked to a smaller torsional deformation as will be shown in figure 12f in the next section.

Figure 11 shows the premultiplied Power Spectral Density (PSD) of the power (Figure 11a) and thrust (Figure 11b) coefficients evolution. The PSD is normalized by the variance of each coefficient $\sigma^2$ and plotted versus the frequency normalized by the rotational frequency of the rotor, $f/f_{rot}$ where the latter is denoted as $1P = f_{rot} = 7.5RPM$ and its multiples will be denoted as $2P, 3P$ etc. In both cases, the CFD-CSD solvers seem to provide a richer representation of the aerodynamic coefficients, capturing the full range of flow-structure interactions. Indeed, an examination of the low-frequency behavior reveals that both quantities exhibit isolated low-frequency peaks when using the BEM-based solvers, a phenomenon not observed with the CFD-CSD, where the low-frequency range is rather broadband and does not present particular peaks. It is important to notice that the frequency 1P can be directly linked to the frequency of the passage of the blade in front of the tower, but also to wind shear loads on the blades. Concerning the first point, a potential flow solution as that used in the BEM solver is keen to provide a simple, single-frequency response, whereas a complex, turbulent flow is expected to result in a more broadband spectrum. Concerning the second point, we have to consider that in LES, the power law profile is imposed at the inlet of the domain but it is free to evolve for 4 diameters before the wind turbine, altering in a non-trivial way the flow field and the consequent frequency response of the blades. This outcome indicates that the BEM-based solvers tend to overcut the power oscillations associated with low-frequencies that are not exactly equal to 1P or 2P. For all solvers, however, the strongest PSD peaks are to be found at much larger frequencies (3P-6P-9P-12P), as also observed by Pagamonci et al. (2023) by means of URANS aeroelastic simulations of the NREL 5-MW, the DTU 10-MW, and the IEA 15-MW turbines. One can also notice that the amplitude associated with the 3P frequency appears to be consistently described by the two solvers, although also in this range the BEM solver appears to overdamp the frequencies in between different peaks.

[Figure]

(a)

(b)

Figure 11: Power Spectral Density (PSD) of the power (a) and thrust (b) coefficients. The vertical dashed lines highlight the rotational frequency of the rotor $1P = f_{rot} = 7.5 RPM$ and the multiples of $3P$, respectively. CFD-CSD/OV ——, CFD-CSD/T ——, *ElastoDyn* ——, *BeamDyn* ——.

Moreover, a good agreement is evident between the two set of results concerning the value of the frequencies and the level of the PSD for frequencies that are multiples of $3P$.

**4.4   Structural response**

This section presents the analysis of the structural dynamics. Figure 12 reports the phase-averaged dynamic response of the free extremity of the blade (left column) and the time-averaged deformation of the entire span (right column). Figure 12a shows how the out-of-plane deformation is mainly governed by the aerodynamic component of the force normal to the rotor plane and, hence, to the aerodynamic effects, heavily affected by the tower. In fact, it is visible how the tower placed at $\theta = 180°$ produces a drop in the deformation, followed by an elastic dynamic response which restores the value far from the pointing-down position. The time-averaged maximum deformation predicted by the CFD-CSD/OV solver is 16% higher compared to the *ElastoDyn* module and 17% compared to *BeamDyn* (see Figure 12b). On the other hand, the same quantity predicted by the CFD-CSD/T solver is 17% lower compared to the *ElastoDyn* module and 13% compared to *BeamDyn* (see Figure 12b). This is consistent with the fact that including the torsional degree of freedom reduces the loads

(a) Out-of-Plane deflection - tip

(b) Out-of-Plane deflection - span

(c) In-plane deflection - tip

(d) In-plane deflection - span

(e) Torsional deflection - tip

(f) Torsional deflection - span

Figure 12: Phase-averaged deflections at the tip of the blade (left column) and time-averaged deflections along the blade span (right column). CFD-CSD/OV: TN ——, RO ----. CFD-CSD/T: TN ——, RO ----. *ElastoDyn*: TN ——, RO ----. *BeamDyn*: TN ——, RO ----. From figure 13 of Pagamonci et al. (2023) ∘, and figure 7a and 7b of Trigaux et al. (2024) ×.

526 (see figure 8b) and the resulting deformation. Although the trend of deformation with respect to the
527 blade span appears similar to previous predictions based on URANS (see Pagamonci et al. (2023)),

the out-of-plane deformation is rather larger, reaching $16\,m$ at the blade's tip. The amplitude of the deformation is however close to that obtained by Trigaux et al. (2024) using LES. Figure 12c depicts instead the in-plane deformation, which is mostly due to gravity. The results show that the shadowing effect of the tower does not influence this quantity, which is expected as the lag deformation is mainly driven by gravity. Furthermore, the discrepancies obtained between *ElastoDyn* and *BeamDyn* can be attributed to the lack of modes used by the former model to describe the translation in the edgewise direction (see Figure 12d). The discrepancy does not seem to be linked to the linearity of this model, as the result of the CFD-CSD/T solver, which is linear as well, is much closer to the *BeamDyn* results. Moreover, the results of the CFD-CSD/OV and the CFD-CSD/T models are very close each other. It can be noticed that the amplitude of the oscillation of the in-plane deflection is consistent with that reported by Trigaux et al. (2024) (see Figure 7b of their paper, reporting an oscillation between $\approx -2.3$ and $\approx 0.2$ ), although the sign is opposite due to the different frame of reference used.

A further significant insight into the deformation phenomenon is provided by the torsional DoF. Figure 12e shows a comparison of the torsional angle at the tip with *BeamDyn*. Significant discrepancies can be observed between the LES and the BEM approaches, which cannot be reconducted to the different coupling procedures adopted by the models. On the one hand, *BeamDyn* and CFD-CSD/T both take into account the deformation angle in the coupling (Wang et al., 2016b), while in the CFD-CSD/OV solver the angle of attack depends only on the deformation velocity (see Equation 8). However, the gap between the BEM and the CFD-CSD/T curves is quite large, reaching approximatively 20% of the torsional deformation value. These differences likely arise from the combined effects of both aerodynamic and structural modeling approaches used in BEM and LES. Although in the present paper we have mostly focused on a comparison of the structural models, a thorough comparison of the aerodynamics modeling can be found in the report of IEA Task 47 Schepers et al. (2025), where results produced with the present code are included (see, for instance, figure 4.25 and following for non flexible blades). The discrepancy between the BEM and the CFD-CSD results is confirmed by the time-averaged torsional deformation along the span reported in Figure 12f where the maximum percentual gap of *BeamDyn* reaches 29% for the CSD-CFD/OV, and 24% for the CFD-CSD/T. It is noteworthy that the lower torsional deformation resulting from *BeamDyn* leads to the higher aerodynamic loads observed in figure 8c.

Finally, figure 13 illustrates the Power Spectral Density (PSD) of the blade's tip deformation components for the TN configuration (which is characterized by more complex fluid-structure interactions). The premultiplied PSD values are normalized by the variance of the signal, $\sigma^2$, and plotted versus the frequency normalized by the rotor frequency, $f/f_{rot}$. Spectral results have been corroborated through use of the Welch and Lomb-Scargle PSD estimation algorithms.

Figure 13a shows the out-of-plane deformation, which we showed to be influenced mostly by the aerodynamic loading. The results indicate that, for all the numerical approaches used, the observed structural response does not exhibit a peak corresponding to the first flapwise natural frequency, suggesting that the intrinsic dynamics of the structure might play a less prominent role in the deformation process. A similar behavior is found in the results of Trigaux et al. (2024) (see figure 6 of the cited paper) for the same turbine and similar inflow conditions. Noticeably, all the numerical models recovered peaks at frequencies close to the (highly damped) second and third flapwise natural frequencies, but they appear to rather correspond to the $13^{th}$ and $26^{th}$ multiple of the rotational fre-

[Figure]

Figure 13: Power Spectral Density (PSD) of the out-of-plane (a), in-plane (b), and torsional (c) deformations of the blade. The vertical dashed lines represent the first 8 eigenfrequencies of the system. CFD-CSD/OV ——, CFD-CSD/T —— *ElastoDyn* ——, *BeamDyn* ——.

quency (i.e., 13p and 26p). Both CFD-CSD solvers predict larger amplitude responses across a broad frequency range compared to OpenFAST, indicating a higher capability to capture complex flow interactions, including turbulence-induced vibrations. This effect is particularly pronounced at the lower frequencies, probably due to the large-scale three-dimensional structure of the flow impinging on the

turbine, which is not captured by OpenFAST, also due to the fact that the impinging flow on the turbine is purely two-dimensional, while it is not for CFD. For this reason, these aspects seem to be under-represented in the *ElastoDyn* and *BeamDyn* solutions. Although the *ElastoDyn* curve aligns with both the CFD-CSD solvers at some key frequency peaks, it does not account for the fine-scale flow-structure interactions. On the other hand, the *BeamDyn* curve provides better agreement with the CFD-CSD solvers, especially at higher frequencies near the blade's natural modes, suggesting that *BeamDyn* captures more of the structural dynamics, particularly the aeroelastic response, probably due to its nonlinearity or to the number of degrees of freedom considered. Figure 13b shows the in-plane deformation, which is primarily influenced by gravity, centrifugal, and Coriolis forces acting on the blade. The CFD-CSD solvers again demonstrate stronger low-frequency components.

Figure 13c presents the torsional deformation for the CFD-CSD/T and *BeamDyn* solvers, excluding *ElastoDyn*, which neglects the torsional DoF in the model. Additionally, also this quantity demonstrates that the CFD-CSD curves predict higher amplitudes at low frequencies. However, a good agreement between the two solvers is evident at higher frequencies, especially in the range around the first torsional eigenfrequency.

**5    Conclusions**

This study investigated the aeroelastic response of the IEA 15-MW wind turbine by employing a high-fidelity Computational Fluid Dynamics (CFD) solver that couples Large-Eddy Simulation (LES) with a Computational Structural Dynamics (CSD) solver. Two different CSD solvers are considered: the CFD-CSD/OV solver, in which the only structural quantity contributing to the definition of the angle of attack is the deformation velocity, and the CFD-CSD/T solver, in which the instantaneous torsional deformation is also considered when defining the local effective incidence. The results of the two CFD-CSD solvers are compared with those of traditional engineering solvers such as *BeamDyn* and *ElastoDyn*, both relying on Blade Element Momentum (BEM) theory. Two case studies were examined: a rotor-only configuration (RO) and one that included the tower and nacelle (TN).

In the first instance, a flow analysis uncovered important considerations regarding wake entrainment. In particular, the study found that, for the considered turbine, impinged by a laminar sheared inflow, wake recovery is only slightly hindered by the presence of the tower. The entrainment of kinetic energy driven by the tower leads to higher turbulence levels in the near wake, but then result into a slightly decreased mixing behind the turbine, differently to what has been found for the NREL 5MW wind turbine, whose wake recovery was found to be promoted by the presence of the tower. This result requires further examination, as it appears counter-intuitive and has not yet been confirmed by other studies.

In addition, the Power Spectral Density (PSD) of the power and thrust coefficients revealed that the CFD-CSD solver captures a broader range of flow-structure interactions, with a more broadband low-frequency response, compared to the BEM-based solvers. The isolated low-frequency peaks found in *BeamDyn* and *ElastoDyn* suggest that these solvers tend to over-simplify the aerodynamic fluctuations associated with phenomena such as wind shear and tower shadowing. For the large IEA 15-MW turbine, the performance drop caused by tower passage is not very pronounced and the resulting oscillations predicted by the BEM approach appear to be larger than the CFD-CSD solver.

Concerning the forces on the blade and the incidence angle, one can observe a rather good match between the CFD-CSD/OV solver and *ElastoDyn*, as well as between the CFD-CSD/T model and the *BeamDyn* solver. This is likely due to the presence – or not – of the torsional feedback, while non-linearities of the structural solver appear to have only a limited impact on the observed quantities. In agreement with previous studies, the results thus suggest that including the torsional degree of freedom in the structural solver is crucial for accurately describing the amplitude and dynamical behaviour of the aerodynamic quantities.

Moreover, it is observed that duly taking into account the torsional degree of freedom reduces the value of $C_p$. This feature is consistently observed by both CFD and BEM approaches. However, one can observe that *BeamDyn* predicts lower values of the torsional deformation and thus higher values of the aerodynamic edgewise forces with respect to the CFD-CSD/T approach, leading to a larger $C_p$ value than that predicted by LES. All in all, it can be concluded that for the considered setup, the CFD-CSD solvers tend to exhibit larger amplitudes at lower frequencies with respect to BEM ones.

The structural response of the wind turbine blade has been assessed by comparing the out-of-plane, in-plane, and torsional deformations obtained from the CFD-CSD solvers, *ElastoDyn*-based, and *BeamDyn*-based OpenFAST solver. In-plane deformation, influenced significantly by centrifugal forces, appears to be better captured by the CFD-CSD solvers, especially in the low-frequency range. Concerning the out-of plane deflection, large discrepancies are seen between the two CFD-CSD solvers, as well as between both BEM modules and the LES.

Our results underscore the importance of incorporating torsional deformation effects in the definition of the angle of attack and using high-fidelity aeroelastic models to ensure accurate predictions of wind turbine blade performance with a richer fluid dynamics. Whereas, the linearity of the structural model does not appear to have a strong effect on the aerodynamical quantities, deformations and loads. In general, the comparison of the results of the CFD-CSD solver with those of the engineering solver shows differences especially in the region behind the tower. The observed differences likely stem from the combined effects of differences in aerodynamic and structural fidelity, and cannot be uniquely attributed to one component alone.

Future work will explore the effect of turbulent fluctuations at the inlet to better investigate the impact of the atmospheric boundary layer on the aerodynamic forces, loads and deformations of the present turbine.

**A   Appendix A. Grid convergence study for LES**

A grid convergence study was conducted to evaluate the sensitivity of the LES results to spatial and temporal resolution. Two further simulations were carried out using grids of different densities: a coarser mesh and a finer mesh, having approximately 40% less and more grid points than the former in each spatial direction, respectively. This allowed for a more detailed resolution of flow structures and aerodynamic quantities. Moreover, both simulations use the same $CFL = 0.65$ as the present grid. The average time step obtained and the other key parameters of different LES runs are summarized in Table A1.

The comparison in figure A2 shows that the results obtained using the coarse and fine grids are extremely close to each other along the entire blade span. In particular, the curves of the angle of

| Parameter | Coarse Grid | Present Grid | Fine Grid |
|---|---|---|---|
| Total number of cells | $1.31 \times 10^8$ | $5.37 \times 10^8$ | $1.36 \times 10^9$ |
| Largest cell diagonal (m) | 8.1 | 5.0 | 3.5 |
| Smallest cell diagonal (m) | 3.9 | 2.5 | 1.7 |
| Actuator points per blade | 54 | 86 | 128 |
| Average time step (s) | 0.043 | 0.024 | 0.012 |
| Total number of threads | 320 | 512 | 768 |

Table A1: Comparison of the main parameters for different meshes.

attack are almost indistinguishable for the coarser and the reference grid, even in the outer portion of the blade, where stronger differences were expected due to tip effects and local three-dimensionality. Slightly larger differences are recovered between the reference and the finer grid, but only at low radius. In particular, for these two grids the maximum deviation of the incidence angle $\alpha$ between the two simulations at 80% of the span reaches a value of $\Delta\alpha_{max} \approx 0.2°$, corresponding to a relative difference of 1.6%. Whereas, the maximum deviation between the reference and the coarser grids at 80% of the blade span is $\Delta\alpha_{max} \approx 0.1°$, corresponding to a relative difference of 1.5%. Similarly, the aerodynamic forces component distributions exhibit negligible variation between the reference and finer resolutions, and less than 1% relative variations between the reference and the coarser grids, confirming the overall consistency of the LES solution examined in the Sec. 4 with respect to mesh refinement.

These results indicate that the coarse grid already accurately captures the main aerodynamic features, making the use of a finer mesh unjustified given its higher computational cost and minimal accuracy gain.

**B Appendix B. Validation of the structural model**

The structural model for the IEA 15MW wind turbine has been cross-validated with many other aeroelastic numerical codes within the framework of the International Energy Agency (IEA) Wind TCP Task 47 TURBINIA (Schepers et al., 2025). In this IEA Task, a consortium of research institutions and industrial partners benchmarked their own aeroelastic codes on the IEA 15 MW wind turbine (Cacciola et al., 2025). Since we cannot report in this paper data from all these partners, we provide here a preliminary study was conducted to validate the structural model prior to coupling it with the CFD solver. Figure B1 shows the distributions of the structural and constructive properties along the blade, which were utilized as input for the modal CSD analysis. A convergence study to determine the proper number of elements, $N_e$, (not reported here for brevity) was conducted, leading to the choice $N_e = 80$. Furthermore, the results of the present structural analysis were compared with those of five models including: the prismatic Timoshenko model without torsion (H2-PTNT); the Timoshenko model with a fully populated stiffness matrix (H2-FPM) from the study of Rinker et al. (2020); the 3D Finite Element Analysis (3D FEA) selected from Zhang et al. (2023); the ElastoDyn model; the BeamDyn model. Figure B2 shows the first 8 eigenfrequencies using the present method compared

[Figure]

Figure A1

Figure A2: Average aerodynamic quantities along the blade obtained from the coarse grid (red line), the present grid (black line) and the finer grid (green line). (top left) Incidence angle, (top right) Aerodynamic pitch moment, (bottom left) flapwise aerodynamic force, (bottom right) edgewise aerodynamic force.

with the results of these models. The computed values of the modal frequencies appear to be consistent with the other results, although some discrepancies in the higher-order modes are observed. Moreover, an analysis of the most important modes was conducted: Table B1 provides the classification of the first 8 modes, whereas, Figures B3, B4, and B5 show the modal displacements for the first spanwise, edgewise, and torsional modes, respectively.

| # | $f_n[Hz]$ | Mode |
|---|-----------|------|
| 1 | 0.5369 | 1st flapwise |
| 2 | 0.7267 | 1st edgewise |
| 3 | 1.577 | 2nd flapwise |
| 4 | 2.267 | 2nd edgewise |
| 5 | 3.113 | 3rd flapwise |
| 6 | 3.642 | 1st torsional |
| 7 | 4.571 | 3rd edgewise |
| 8 | 5.385 | 4th flapwise |

Table B1: Classification of the first 8 structural modes.

[Figure]

Figure B1: Structural properties of the blade along the span: (a) stiffness, (b) inertia, (c) density, (d) local twist angle.

[Figure]

Figure B2: A comparison of the eigenfrequencies computed by different structural models.

[Figure]

Figure B3: Mode 1 shape for all the DoFs.

[Figure]

Figure B4: Mode 2 shape for all the DoFs.

[Figure]

Figure B5: Mode 6 shape for all the DoFs.

688 *Author contribution.* CB: Investigation, Writing - Original draft, Formal analysis, Methodology, Soft-
689 ware, Validation. SC: Conceptualization, Investigation, Writing - Review & Editing, Supervision. FM:
690 Methodology, Software, Validation. GDP: Formal analysis, Writing - Review & Editing, Methodol-
691 ogy, software. SL: Conceptualization, Software, Supervision. PDP: Conceptualization, Investigation,
692 Writing - Review & Editing, Supervision.

693 *Competing interests.* The authors declare that they have no competing interests.

694 *Acknowledgments.* This study has been partially funded under the National Recovery and Resilience
695 Plan (NRRP), Mission 4 Component 2 Investment 1.3 - Call for tender No. 1561 of 11.10.2022
696 Project code PE0000021, Project title "Network 4 Energy Sustainable Transition – NEST", and
697 under the PRIN grant 20229YJP33, "Diffuser augmented Wind Turbines for URBban environments"
698 (DWTURB). Both are grants of Ministero dell'Università e della Ricerca (MUR), funded by the
699 European Union – NextGenerationEU. The cooperative work on the 15MW NREL wind turbine
700 within the IEA WIND TASK 47 - TURBINIA is also acknowledged.

**References**

702 Abdel Hafeez, M. M. and El-Badawy, A. A.: Flutter limit investigation for a horizontal axis wind tur-
703 bine blade, Journal of Vibration and Acoustics, 140, 041 014, doi:https://doi.org/10.1115/1.4039402,
704 2018.

705 Allah, V. A. and Sha ei Mayam, M. H.: Large Eddy Simulation of flow around a single and two in-line
706 horizontal-axis wind turbines, Energy, 121, 533–544, 2017.

707 Bartl, J. and Satran, L.: Blind test comparison of the performance and wake flow between two in-line
708 wind turbines exposed to different turbulent inflow conditions, Wind Energy Science, 2, 55–76, 2017.

709 Bayati, I., Belloli, M., Bernini, L., and Zasso, A.: Aerodynamic design methodology for wind tunnel
710 tests of wind turbine rotors, Journal of Wind Engineering and Industrial Aerodynamics, 167, 217–
711 227, doi:https://doi.org/10.1016/j.jweia.2017.05.004, 2017.

712 Bazilevs, Y., Hsu, M.-C., Kiendl, J., Wüchner, R., and Bletzinger, K.-U.: 3D simulation of wind
713 turbine rotors at full scale. Part II: Fluid-structure interaction modeling with composite blades,
714 International Journal for Numerical Methods in Fluids, 65, 236 – 253, doi:10.1002/fld.2454, cited
715 by: 441, 2011.

716 Bernardi, C., Posta, G. D., Palma, P. D., Leonardi, S., Bernardoni, F., Bernardini, M., and Cherubini,
717 S.: The effect of the tower's modeling on the aero-elastic response of the NREL 5 MW wind turbine,
718 Journal of Physics: Conference Series, 2505, 012 037, doi:10.1088/1742-6596/2505/1/012037, URL
719 https://dx.doi.org/10.1088/1742-6596/2505/1/012037, 2023.

Boorsma, K., Greco, L., and Bedon, G.: Rotor wake engineering models for aeroelastic applications, Journal of Physics: Conference Series, 1037, 062 013, doi:10.1088/1742-6596/1037/6/062013, URL https://doi.org/10.1088/1742-6596/1037/6/062013, 2018.

Boorsma, K., Schepers, G., Aagard Madsen, H., Pirrung, G., Sørensen, N., Bangga, G., Imiela, M., Grinderslev, C., Meyer Forsting, A., Shen, W. Z., Croce, A., Cacciola, S., Schaffarczyk, A. P., Lobo, B., Blondel, F., Gilbert, P., Boisard, R., Höning, L., Greco, L., Testa, C., Branlard, E., Jonkman, J., and Vijayakumar, G.: Progress in the validation of rotor aerodynamic codes using field data, Wind Energy Science, 8, 211–230, doi:10.5194/wes-8-211-2023, URL https://wes.copernicus.org/articles/8/211/2023/, 2023.

Boorsma, K., Schepers, J. G., Pirrung, G. R., Madsen, H. A., Sørensen, N. N., Grinderslev, C., Bangga, G., Imiela, M., Croce, A., Cacciola, S., Blondel, F., Branlard, E., and Jonkman, J.: Challenges in Rotor Aerodynamic Modeling for Non-Uniform Inflow Conditions, Journal of Physics: Conference Series, 2767, 022 006, doi:10.1088/1742-6596/2767/2/022006, URL https://doi.org/10.1088/1742-6596/2767/2/022006, 2024.

Burton, T., Jenkins, N., Sharpe, D., and Bossanyi, E.: Wind energy handbook, John Wiley & Sons, 2011.

Cacciola, S., Croce, A., Bangga, G., Pirrung, G., H., M., Sørensen, N., Grinderslev, G., Bonfils, N., Persent, E., Gilbert, I., Joulin, A., Greco, L., Aryan, N., Castorrini, A., Morici, A., Chetan, M., Jonkman, J., Branlard, E., Cherubini, S., Bernardi, C., Boorsma, K., Schepers, J. D., Bianchini, A., Pagamoci, L., Papi, F., Hach, O., Imiela, M., and Witt, D.: A Comparative Study of Different Modeling Tools and Analysis Techniques for Aeroelastic Stability Assessment, in: submitted to The Science of Making Torque from Wind, 2025.

Chen, X.: Experimental investigation on structural collapse of a large composite wind turbine blade under combined bending and torsion, Composite Structures, 160, 435–445, doi: https://doi.org/10.1016/j.renene.2022.08.113, 2017.

Chung, J. and Hulbert, G. M.: A Time Integration Algorithm for Structural Dynamics With Improved Numerical Dissipation: The Generalized-$\alpha$ Method, Journal of Applied Mechanics, 60, 371–375, doi: 10.1115/1.2900803, URL https://doi.org/10.1115/1.2900803, 1993.

Ciri, U., Petrolo, G., Salvetti, M. V., and Leonardi, S.: Large-Eddy Simulations of Two In-Line Turbines in a Wind Tunnel with Different Inflow Conditions, Energies, 10, 821, 2017.

Courant, R., Friedrichs, K., and Lewy, H.: On the partial difference equations of mathematical physics, IBM journal of Research and Development, 11, 215–234, doi:https://doi.org/10.1147/rd.112.0215, 1967.

Damgaard, M., Ibsen, L. B., Andersen, L. V., and Andersen, J. K.: Cross-wind modal properties of offshore wind turbines identified by full scale testing, Journal of Wind Engineering and Industrial Aerodynamics, 116, 94–108, doi:https://doi.org/10.1016/j.jweia.2013.03.003, 2013.

Damiani, R., Jonkman, J., and Hayman, G.: SubDyn user's guide and theory manual, Tech. rep., National Renewable Energy Lab.(NREL), Golden, CO (United States), 2015.

De Cillis, G., Semeraro, O., Leonardi, S., De Palma, P., and Cherubini, S.: Dynamic-mode-decomposition of the wake of the NREL-5MW wind turbine impinged by a laminar inflow, Renewable Energy, 199, 1–10, 2022.

Della Posta, G., Leonardi, S., and Bernardini, M.: A two-way coupling method for the study of aeroelastic effects in large wind turbines, Renewable Energy, 190, 971–992, doi: https://doi.org/10.1016/j.renene.2022.03.158, 2022.

Della Posta, G., Leonardi, S., and Bernardini, M.: Large eddy simulations of a utility-scale horizontal axis wind turbine including unsteady aerodynamics and fluid-structure interaction modelling, Wind Energy, 26, 98–125, doi:https://doi.org/10.1002/we.2789, 2023.

Dong, X., Lian, J., Wang, H., Yu, T., and Zhao, Y.: Structural vibration monitoring and operational modal analysis of offshore wind turbine structure, Ocean Engineering, 150, 280–297, doi: https://doi.org/10.1016/j.oceaneng.2017.12.052, 2018.

Gaertner, E., Rinker, J., Sethuraman, L., Zahle, F., Anderson, B., Barter, G., Abbas, N., Meng, F., Bortolotti, P., Skrzypinski, W., Scott, G., Feil, R., Bredmose, H., Dykes, K., Sheilds, M., Allen, C., and Viselli, A.: Definition of the IEA 15-Megawatt Offshore Reference Wind Turbine, Tech. rep., International Energy Agency, URL https://www.nrel.gov/docs/fy20osti/75698.pdf, 2020a.

Gaertner, E., Rinker, J., Sethuraman, L., Zahle, F., Anderson, B., Barter, G., Abbas, N., Meng, F., Bortolotti, P., Skrzypinski, W., et al.: Definition of the IEA 15-megawatt offshore reference wind turbine, 2020b.

Hansen, M.: Aerodynamics of wind turbines, Routledge, 2015.

Hansen, M. H.: Aeroelastic instability problems for wind turbines, Wind Energy: An International Journal for Progress and Applications in Wind Power Conversion Technology, 10, 551–577, doi: https://doi.org/10.1002/we.242, 2007.

Heinz, J.: Partitioned Fluid-Structure Interaction for Full Rotor Computations Using CFD, Ph.D. thesis, Denmark, 2013.

Jonkman, J.: The New Modularization Framework for the FAST Wind Turbine CAE Tool, doi: 10.2514/6.2013-202, URL https://arc.aiaa.org/doi/abs/10.2514/6.2013-202, 2013.

Jonkman, J. M., Hayman, G., Jonkman, B., Damiani, R., and Murray, R.: AeroDyn v15 user's guide and theory manual, NREL Draft Report, 46, 2015.

Krogstad, P.-A., Satran, L., and Adaramola, M. S.: Blind Test 3: calculations of the performance and wake development behind two in-line and offset model wind turbines, Journal of Fluids and Structures, 52, 65–80, 2015.

Manwell, J. F., McGowan, J. G., and Rogers, A. L.: Wind energy explained: theory, design and application, John Wiley & Sons, 2010.

Martinez-Tossas, L. A., Churchfield, M. J., Yilmaz, A. E., Sarlak, H., Johnson, P. L., Sørensen, J. N., Meyers, J., and Meneveau, C.: Comparison of four large-eddy simulation research codes and effects of model coefficient and inflow turbulence in actuator-line-based wind turbine modeling, Journal of Renewable and Sustainable Energy, 10, doi:https://doi.org/10.1063/1.5004710, 2018.

Moeller, T.: Blade cracks signal new stress problem, WindPower Monthly, 25, 1997.

Moriarty, P. J. and Hansen, A. C.: AeroDyn Theory Manual, doi:10.2172/15014831, 2005.

Orlandi, P.: Fluid flow phenomena: a numerical toolkit, vol. 55, Springer Science & Business Media, 2012.

Orlandi, P. and Leonardi, S.: DNS of turbulent channel flows with two-and three-dimensional roughness, Journal of Turbulence, p. N73, doi:https://doi.org/10.1080/14685240600827526, 2006.

Pagamonci, L., Papi, F., Balduzzi, F., Xie, S., Sadique, J., Scienza, P., and Bianchini, A.: To what extent is aeroelasticity impacting multi-megawatt wind turbine upscaling? A critical assessment, in: Journal of Physics: Conference Series, vol. 2648, p. 012005, IOP Publishing, doi:10.1088/1742-6596/2648/1/012005, 2023.

Pino Martín, M., Piomelli, U., and Candler, G. V.: Subgrid-scale models for compressible large-eddy simulations, Theoretical and Computational Fluid Dynamics, 13, 361–376, 2000.

Porte-Agel, F. and Wu, Y.-T.: Large-Eddy Simulation of Wind-Turbine Wakes: Evaluation of Turbine Parametrisations, Boundary Layer Meteorology, 138, 345–366, 2011.

Reschke, C.: Flight loads analysis with inertially coupled equations of motion, in: AIAA Atmospheric Flight Mechanics Conference and Exhibit, p. 6026, doi:https://doi.org/10.2514/6.2005-6026, 2005.

Ribeiro, A. F. P., Casalino, D., and Ferreira, C. S.: Nonlinear inviscid aerodynamics of a wind turbine rotor in surge, sway, and yaw motions using a free-wake panel method, Wind Energy Science, 8, 661–675, doi:10.5194/wes-8-661-2023, URL https://wes.copernicus.org/articles/8/661/2023/, 2023.

Rinker, J., Gaertner, E., Zahle, F., Skrzypiński, W., Abbas, N., Bredmose, H., Barter, G., and Dykes, K.: Comparison of loads from HAWC2 and OpenFAST for the IEA Wind 15 MW Reference Wind Turbine, in: Journal of Physics: Conference Series, vol. 1618, p. 052052, IOP Publishing, doi:10.1088/1742-6596/1618/5/052052, 2020.

Sabale, A. K. and Gopal, N. K. V.: Nonlinear Aeroelastic Analysis of Large Wind Turbines Under Turbulent Wind Conditions, AIAA Journal, 57, 4416–4432, doi:10.2514/1.J057404, URL https://doi.org/10.2514/1.J057404, 2019.

Saltari, F., Riso, C., Matteis, G. D., and Mastroddi, F.: Finite-element-based modeling for flight dynamics and aeroelasticity of flexible aircraft, Journal of Aircraft, 54, 2350–2366, doi: https://doi.org/10.2514/1.C034159, 2017.

Santoni, C., Ciri, U., Rotea, M., and Leonardi, S.: Development of a high fidelity CFD code for wind farm control, in: 2015 American Control Conference (ACC), pp. 1715–1720, doi: 10.1109/ACC.2015.7170980, 2015.

Santoni, C., Carrasquillo, K., Arenas-Navarro, I., and Leonardi, S.: Effect of tower and nacelle on the flow past a wind turbine, Wind Energy, 20, 1927–1939, doi:https://doi.org/10.1002/we.2130, 2017.

Santoni, C., García-Cartagena, E. J., Ciri, U., Zhan, L., Valerio Iungo, G., and Leonardi, S.: One-way mesoscale-microscale coupling for simulating a wind farm in North Texas: Assessment against SCADA and LiDAR data, Wind Energy, 23, 691–710, doi:https://doi.org/10.1002/we.2452, 2020.

Schepers, J., Boorsma, K., Madsen, H., Pirrung, G., Bangga, G., Guma, G., Lutz, T., Potentier, T., Braud, C., Guilmineau, E., Croce, A., Cacciola, S., Schaffarczyk, A. P., Lobo, B. A., Ivanell, S., Asmuth, H., Bertagnolio, F., Sørensen, N., Shen, W. Z., Grinderslev, C., Forsting, A. M., Blondel, F., Bozonnet, P., Boisard, R., Yassin, K., Hoening, L., Stoevesandt, B., Imiela, M., Greco, L., Testa, C., Magionesi, F., Vijayakumar, G., Ananthan, S., Sprague, M. A., Branlard, E., Jonkman, J., Carrion, M., Parkinson, S., and Cicirello, E.: IEA Wind TCP Task 29, Phase IV: Detailed Aerodynamics of Wind Turbines, doi:10.5281/zenodo.4813068, URL https://doi.org/10.5281/zenodo.4813068, 2021.

Schepers, J. G., Boorsma, K., Bois, R., Bangga, G., Jonkman, J., Kelley, C., Branlard, E., Gonçalves Pinto, W., Imiela, M., Hach, O., Greco, L., Testa, C., Aryan, N., Madsen, H., Croce, A., Cacciola, S., Pirrung, G., Sørensen, N., Grinderslev, C., Bernardi, C., Cherubini, S., Bianchini, A., Papi, F., Pagamonci, L., Braud, C., Höning, L., Theron, J., and Mohan, K.: Turbinia, turbulent inflow innovative aerodynamics, Tech. rep., IEA Wind TCP–Task47, 2025.

Shen, W. Z., Mikkelsen, R., Sørensen, J. N., and Bak, C.: Tip loss corrections for wind turbine computations, Wind Energy: An International Journal for Progress and Applications in Wind Power Conversion Technology, 8, 457–475, doi:https://doi.org/10.1002/we.153, 2005.

Sorensen, J. and Shen, W. Z.: Numerical Modeling of Wind Turbine Wakes, Journal of Fluids Engineering, 124, 393–399, doi:10.1115/1.1471361, URL https://doi.org/10.1115/1.1471361, 2002.

Sørensen, J. N.: Aerodynamic aspects of wind energy conversion, Annual Review of Fluid Mechanics, 43, 427–448, doi:https://doi.org/10.1146/annurev-fluid-122109-160801, 2011.

Stevens, R. J., Martinez-Tossas, L. A., and Meneveau, C.: Comparison of wind farm large eddy simulations using actuator disk and actuator line models with wind tunnel experiments, Renewable Energy, 116, 470–478, 2018.

Trigaux, F., Chatelain, P., and Winckelmans, G.: Investigation of blade flexibility effects on the loads and wake of a 15 MW wind turbine using a flexible actuator line method, Wind Energy Science, 9, 1765–1789, doi:https://doi.org/10.5194/wes-9-1765-2024, 2024.

Troldborg, N.: Actuator line modeling of wind turbine wakes, Ph.D. thesis, Technical University of Denmark, 2009.

Vermeer, L., Sørensen, J. N., and Crespo, A.: Wind turbine wake aerodynamics, Progress in aerospace sciences, 39, 467–510, doi:https://doi.org/10.1016/S0376-0421(03)00078-2, 2003.

Wang, L., Liu, X., and Kolios, A.: State of the art in the aeroelasticity of wind turbine blades: Aeroelastic modelling, Renewable and Sustainable Energy Reviews, 64, 195–210, doi: https://doi.org/10.1016/j.rser.2016.06.007, 2016a.

Wang, Q., Jonkman, J., Sprague, M., and Jonkman, B.: Beamdyn user's guide and theory manual, National Renewable Energy Laboratory, 2016b.

Yu, D. O. and Kwon, O. J.: Predicting wind turbine blade loads and aeroelastic response using a coupled CFD-CSD method, Renewable Energy, 70, 184 – 196, doi:10.1016/j.renene.2014.03.033, cited by: 111, 2014.

Zahle, F., Barlas, A., Lønbæk, K., Bortolotti, P., Zalkind, D., Wang, L., Labuschagne, C., Sethuraman, L., and Barter, G.: Definition of the IEA Wind 22-Megawatt Offshore Reference Wind Turbine, Technical University of Denmark, doi:10.11581/DTU.00000317, dTU Wind Energy Report E-0243 IEA Wind TCP Task 55, 2024.

Zhang, Y., Song, Y., Shen, C., and Chen, N.-Z.: Aerodynamic and structural analysis for blades of a 15MW floating offshore wind turbine, Ocean Engineering, 287, 115 785, doi: https://doi.org/10.1016/j.oceaneng.2023.115785, 2023.

Zheng, J., Wang, N., Wan, D., and Strijhak, S.: Numerical investigations of coupled aeroelastic performance of wind turbines by elastic actuator line model, Applied Energy, 330, 120 361, doi: https://doi.org/10.1016/j.apenergy.2022.120361, 2023.